# Transcription factor-driven coordination of cell cycle exit and lineage-specification in vivo during granulocytic differentiation

## In memoriam Professor Niels Borregaard

Kim Theilgaard-Mönch [1,2,3,4,8 ✉], Sachin Pundhir[1,2,3,5,8], Kristian Reckzeh [1,2,3], Jinyu Su [1,2,3], Marta Tapia [1,2,3], Benjamin Furtwängler [1,2,3], Johan Jendholm[1,2,3], Janus Schou Jakobsen[1,2,3], Marie Sigurd Hasemann[1,2,3], Kasper Jermiin Knudsen[1,2,3], Jack Bernard Cowland[4,6], Anna Fossum [2], Erwin Schoof [1,2,3,7], Mikkel Bruhn Schuster[1,2,3] & Bo T. Porse [1,2,3 ✉]

Differentiation of multipotent stem cells into mature cells is fundamental for development and homeostasis of mammalian tissues, and requires the coordinated induction of lineage-specific transcriptional programs and cell cycle withdrawal. To understand the underlying regulatory mechanisms of this fundamental process, we investigated how the tissue-specific transcription factors, CEBPA and CEBPE, coordinate cell cycle exit and lineage-specification in vivo during granulocytic differentiation. We demonstrate that CEBPA promotes lineage-specification by launching an enhancer-primed differentiation program and direct activation of CEBPE expression. Subsequently, CEBPE confers promoter-driven cell cycle exit by sequential repression of MYC target gene expression at the G1/S transition and E2F-meditated G2/M gene expression, as well as by the up-regulation of *Cdk1/2/4* inhibitors. Following cell cycle exit, CEBPE unleashes the CEBPA-primed differentiation program to generate mature granulocytes. These findings highlight how tissue-specific transcription factors coordinate cell cycle exit with differentiation through the use of distinct gene regulatory elements.

[1] The Finsen Laboratory, Rigshospitalet, Faculty of Health Sciences, University of Copenhagen, Copenhagen, Denmark. [2] Biotech Research and Innovation Centre, Faculty of Health Sciences, University of Copenhagen, Copenhagen, Denmark. [3] Novo Nordisk Foundation Center for Stem Cell Biology, DanStem, Faculty of Health Sciences, University of Copenhagen, Copenhagen, Denmark. [4] Department of Hematology, Rigshospitalet, Copenhagen, Denmark. [5] The Bioinformatics Centre, Department of Biology, Faculty of Natural Sciences, University of Copenhagen, Copenhagen, Denmark. [6] Department of Clinical Genetics, Rigshospitalet, Copenhagen, Denmark. [7] Department of Biotechnology and Biomedicine, Technical University of Denmark, Lyngby, Denmark. [8] These authors contributed equally: Kim Theilgaard-Mönch, Sachin Pundhir. ✉email: kim.theilgaard@finsenlab.dk; bo.porse@finsenlab.dk

The maintenance of mammalian tissues and organs is sustained by stem cells and their progeny which gradually change their transcriptional programs as they transit along their differentiation trajectory and restrict their lineage potential before entering a postmitotic state, and ultimately become fully mature tissue-specific cells[1–3]. This temporal coupling of lineage-commitment, cell cycle exit, and terminal differentiation is fundamental for the lifelong maintenance of tissue homeostasis. Consistently, genetic aberrations of key regulators of differentiation programs can lead to disease, including cancer[4–9]. Hence in the context of disease, it is essential to understand how key transcriptional regulators coordinate proliferation and differentiation during normal organ and tissue development.

In actively proliferating cells, cyclin-dependent kinases (CDK) and their cognate-activating cyclin partners are master regulators of the cell cycle machinery, which coordinate the transition from one cell cycle phase to the next. Mitogenic stimuli such as growth-promoting factors upregulate Cyclin D leading to activation of CDK4/6 during the early G1 phase which in turn phosphorylates RB in complex with E2F, conferring its release from E2F and transcription of E2F cell cycle target genes such as cyclin E/A/B, CDK2, CDK1, and MYC[10–12]. Hence the backbone of the regulatory machinery controlling cell cycle progression is formed by CDK4/6–cyclin D driven de-repression of E2F target genes by mitogenic stimuli in the early G1 phase and subsequent E2F-mediated upregulation of CDK2–cyclin E and CDK1-cyclin A/B expression in G1/S and G2/M phases, respectively[11–13]. Cell cycle exit and persistent postmitotic growth arrest during the course of tissue and organ development are controlled by mechanisms inhibiting the activities of cell cycle master regulators. These include transcription factors (TFs), which concomitantly upregulate differentiation genes and inhibitors of CDKs, leading to CDK–cyclin inhibition and sustained RB repression of E2F-mediated cell cycle gene expression[14,15]. Moreover, direct inhibition of E2F and repression of cyclin expression by TFs have been reported as a regulatory mechanism blocking E2F-mediated gene expression and proliferation[16–18]. However, the exact temporal sequence and combination of regulatory mechanisms launched by TFs to coordinate cell cycle exit and differentiation are currently poorly understood[3].

Hematopoiesis represents a paradigm of mammalian differentiation and tissue homeostasis where multipotent tissue-specific hematopoietic stem cells (HSCs) either self-renew to maintain the stem cell pool or differentiate to sustain lifelong production of all mature blood cell lineages. Among these, polymorphonuclear neutrophilic granulocytes (in the following referred to as GR) are short-lived effector cells (1–2 days) of the innate immune system that are continuously replenished by HSCs in the bone marrow (BM) through a process referred to as granulocytic differentiation[19]. Simplified, this process can be categorized into two functionally distinct phases; (i) an early phase of lineage-specification that is characterized by cellular proliferation and differentiation of HSCs via multipotent progenitors (MPPs), bipotent pre-granulocyte macrophage, and granulocyte-macrophage progenitors (preGMs and GMPs) toward progenitors restricted to the granulocytic lineage (GPs), and (ii) a terminal phase of differentiation which is characterized by growth arrest and acquisition of a highly specific cell identity while cells differentiate via a series of intermediary stages into fully mature GRs[19–24].

Recent mechanistic and genetic studies by our group and others have demonstrated that members of the CCAAT/enhancer-binding protein (CEBP) family of transcription factors, namely CEBPA and CEBPE, are indispensable for normal granulocytic differentiation[16,25,26]. CEBPA has emerged as a key regulator of myeloid lineage-specification and granulocytic differentiation and is expressed in HSCs as well as in myeloid-committed hematopoietic progenitor cells (HPCs) such as preGMs, GMPs, and their lineage-restricted granulocytic progeny[20,25,27]. Mice with conditional disruption of Cebpa in the hematopoietic system, exhibit a block at the preGM to GMP transition. As a result, these mice accumulate preGMs in their BM and lack GMPs as well as their mature progeny, similar to what is observed in humans with acute myeloid leukemia (AML)[25,28]. In contrast, CEBPE is expressed during terminal granulocytic differentiation[20,29,30]. Consistently, mice with a targeted disruption of Cebpe as well as patients suffering from specific granule deficiency (SGD) due to mutations of the CEBPE gene fail to produce mature GRs but rather exhibit an expansion of progenitors restricted to the granulocytic lineage[8,9,31,32].

Consistent with their expression during granulocytic differentiation, mechanistic studies have demonstrated that CEBPA and CEBPE can transactivate several myeloid- and GR-specific genes, including a number of early and late-appearing granule proteins and hematopoietic growth factor receptors critical for proliferation and specification of cell identity during granulocytic differentiation[33–35]. Besides acting as a transcriptional activator, CEBPA has been demonstrated to bind to E2F in vitro and in vivo via its basic region, and repress E2F-mediated transcription of target genes such as Myc[16,17]. In alignment with CEBPA, CEBPE has also been shown to bind E2F during granulocytic differentiation of myeloid cell lines and repress E2F-mediated transcription in luciferase reporter assays[36].

Taken together, these studies have identified CEBPA and CEBPE as key regulators of cellular proliferation and activators of lineage-specific gene transcription during hematopoietic differentiation along the granulocytic pathway. However, despite their substantial functional redundancy, little is known of how CEBPA and CEBPE actually coordinate their mutual activities in a temporal manner to promote granulocytic differentiation during steady-state hematopoiesis in vivo.

In this work, we applied a systematic approach to uncover how designated key transcriptional regulators (i.e., CEBPA and CEBPE) orchestrate sequential lineage-specification, cell cycle exit, and terminal differentiation at the promoter and enhancer level during hematopoietic differentiation. For this, we implemented a hierarchy of fifteen immunophenotypically defined differentiation stages, which refined the granulocytic and monocytic (GM) differentiation hierarchies at a hitherto unprecedented resolution and subjected this hierarchy to global mRNA and chromatin immunoprecipitation sequencing (RNA-seq and ChIP-seq) analyses. Here, we demonstrate an almost exclusive sequential expression of CEBPA and CEBPE during early and late granulocytic differentiation, respectively. Strikingly, our study further demonstrates that CEBPA promotes lineage-specification by direct activation of CEBPE expression and transient priming of differentiation genes at enhancers. Subsequently, CEBPE takes over and coordinates promoter-driven repression of cell cycle activators with enhancer-driven activation of a plethora of differentiation genes defining the cell identity of fully mature GR.

## Results

**High-resolution mapping of GM differentiation.** Differentiation of HSCs into fully mature GRs and monocytes represents a continuous and highly coordinated process, which is characterized by gradual changes in surface molecule expression, morphology, and proliferation. In order to study steady-state granulocytic and monocytic differentiation at a very high resolution in vivo, we developed a comprehensive flow cytometry-based sorting protocol allowing for prospective purification of murine BM populations representing successive stages of granulocytic and monocytic differentiation. In brief, immunopheno-typic HSCs and early HPCs (i.e., LIN-SCA + KIT+, referred to as

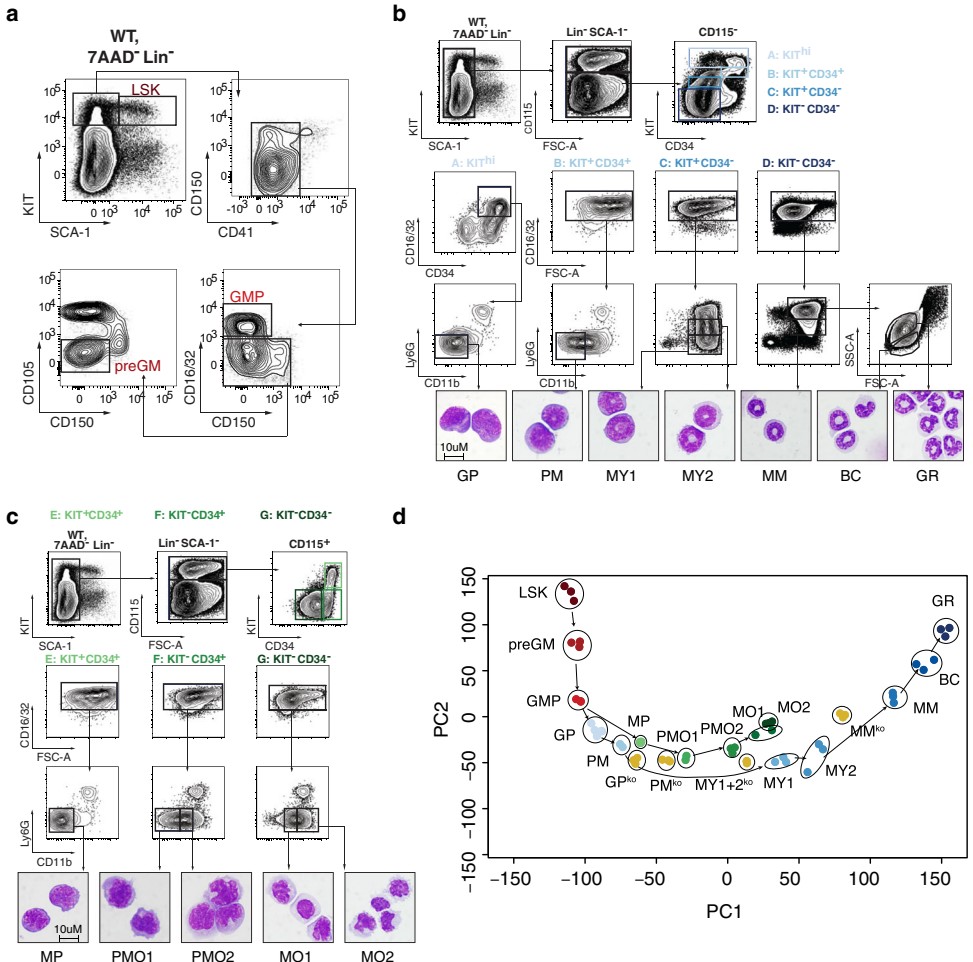

**Fig. 1 High-resolution immunophenotypic and transcriptome mapping of steady-state granulocytic and monocytic differentiation hierarchies. a–c** Flow cytometry-based gating strategy for sorting of murine bone marrow (BM) populations representing early stages of myeloid (i.e., granulocytic-monocytic) differentiation (**a**) and late stages of granulocytic (**b**) and monocytic differentiation (**c**). **d** PCA plot of murine BM populations clustered based on the similarity of gene expression profiles. Replicates of BM populations are encircled. **a** Early myeloid (i.e., granulocytic-monocytic) differentiation hierarchy: LIN-SCA + KIT + hematopoietic stem and early progenitor (LSK), pre-granulocyte-monocyte progenitors (preGM), and granulocyte-monocyte progenitors (GMP). **b** Late granulocytic differentiation hierarchy: Granulocyte progenitors (GP), promyelocytes (PM), early and late myelocytes (MY1, MY2), metamyelocytes (MM), band cells (BC), and fully mature granulocytes (GR). **c** Late monocytic differentiation hierarchy: Monocyte progenitors (MP), early and late promonocytes (PMO1, PMO2), and early and late monocytes (MO1 and MO2). Source data are provided as a Source Data file.

LSKs), as well as bipotent myeloid preGMs and GMPs, were purified as reported previously (Fig. 1a)[37]. Through combinatorial flow cytometry-based immunophenotyping, assessment of cellular morphology by microscopy, and cell cycle analyses, we further developed a protocol for analysis and purification of murine BM populations representing successive stages of late granulocytic and monocytic differentiation (Fig. 1b, c). With this protocol, monocytic differentiation was defined by five differentiation stages including lineage-restricted monocytic progenitors (MPs), early and late promonocytes (PMO1s and PMO2s), and early and late monocytes (MO1s and MO2s). Significantly, granulocytic differentiation comprised a hierarchy of seven distinct differentiation stages, including progenitors restricted to the granulocytic lineage (GPs), promyelocytes (PMs), early and late myelocytes (MY1s and MY2s), metamyelocytes (MMs), band cells (BCs), and fully mature GRs. As depicted in Fig. 1, microscopy of sorted BM populations demonstrated distinct morphological maturation patterns along the granulocytic and monocytic differentiation pathways, which validated the immunophenotypic differentiation hierarchies at the morphological level. Complementary cell cycle analyses of the granulocytic differentiation

hierarchy revealed abrupt cessation of proliferation at the MY2 to MM transition (Supplementary Fig. 1d, e), which allowed for monitoring of the regulatory mechanisms coordinating cell cycle exit and differentiation in vivo with high temporal accuracy. Overall, these findings highlight that the newly developed flow cytometry protocol represents a comprehensive tool for in vivo studies of neutrophil and monocytic differentiation at a hitherto unprecedented resolution.

**Transcriptional programs of GM differentiation.** In order to assess the gradual changes of gene expression as HSCs commit to bipotent GMPs and further differentiate into fully mature GRs and MOs, we conducted RNA-seq analyses on fifteen prospectively purified BM populations representing successive immunophenotypic stages of our branching granulocytic and monocytic differentiation hierarchy (Supplementary Fig. 2). Principal component analyses (PCA) of the data set revealed a gene expression-based hierarchy of BM populations, which not only faithfully mirrored the trajectories of the granulocytic and monocytic differentiation hierarchies but also showed a tight

clustering of replicate populations emphasizing the high quality of our flow cytometry protocol as well as the RNA-seq data (Fig. 1d). To further validate the potential heterogeneity of sorted BM populations, we analyzed publicly available single-cell RNA sequence (scRNA-seq) data from whole bone marrow samples and the bulk gene expression profiles of our sorted BM populations[22,38]. As shown in Supplementary Fig. 3a, our analysis demonstrated that single cells annotated to a specific sorted BM population based on their gene expression profiles essentially clustered together and maintained the same hierarchical order of differentiation trajectories observed in the PCA plot (Fig. 1d). Additional validation demonstrated that the gene expression profiles of sorted BM populations were similar to those of previously reported BM populations and scRNA-seq bulk signatures representing successive developmental stages of granulocytic and monocytic differentiation (Supplementary Fig. 3)[22–24].

The latter highlights that the sorted BM populations faithfully recapitulate the developmental trajectories of granulocytic and monocytic differentiation by representing successive stages of granulocytic and monocytic differentiation.

Unsupervised hierarchical clustering of the data set identified fifteen minor clusters (Supplementary Fig. 2), which based on similarity were merged into the following five major gene expression clusters (Fig. 2a) to simplify and empower subsequent functional analyses: Cluster A(GMP)—genes expressed during early granulocytic and monocytic differentiation (i.e., in LSKs, preGMPs, GMPs, GPs, PMs, and MPs); cluster B(MY)—genes transiently expressed (i.e., in MY1s and MY2s); cluster C(GR—genes terminally upregulated (i.e., in MM, BC, and GR) during late granulocytic differentiation; cluster D(GR + MO)—genes terminally upregulated during both late granulocytic and monocytic differentiation, and, finally, cluster E(MO)—genes upregulated exclusively during monocytic differentiation (i.e., in PMO1s, PMO2s, MO1s, and MO2s).

We next assessed the expression patterns of the designated key regulators of granulocytic differentiation, CEBPA and CEBPE, as well as G1/S and G2/M phase cell cycle genes and a series of primary, secondary, and tertiary granule proteins known to be expressed sequentially during granulocytic differentiation (Fig. 2a, b and Supplementary Fig. 2a). Strikingly, Cebpa/CEBPA and Cebpe/CEBPE demonstrated an almost exclusive sequential expression at the RNA and protein level (as determined by targeted mass spectrometry) including a narrow developmental window of co-expression in PMs during granulocytic differentiation (Fig. 2b and Supplementary Fig. 4a–e) (i.e., Cebpa/CEBPA expression in LSKs, preGMs, GMPs, GPs, and PMs and Cebpe/CEBPE expression in PMs, MY1s, MY2s, MMs, BCs, and GRs).

Cell cycle genes expressed during the G1/S phase were markedly downregulated at the PM to MY1 transition (cluster A(GMP)) before subsequent downregulation of G2/M genes at the MY2 to MM transition when cells exit the cell cycle (cluster B(MY)) (Fig. 2a). As expected, primary, secondary, and tertiary granule proteins demonstrated sequential expression during early and late granulocytic differentiation (primary granule proteins in GMPs, GPs, PMs cluster 5, secondary granule proteins in PMs, MYs, MMs cluster 10, tertiary granule proteins in MYs, MMs, BCs, GRs cluster 11) (Supplementary Fig. 2).

Complementary analyses of differentially expressed genes in Cebpe KO mice revealed aberrant high expression of early granulocytic differentiation, cell cycle (cluster A (GMP) + B (MY)), and monocytic differentiation genes (cluster D (GR + MO) + E (MO)). In contrast, a large number of genes expressed during late granulocytic differentiation (cluster C (GR)) were markedly downregulated in Cebpe KO mice (Fig. 2a, c). Strikingly, these animals also exhibited temporally aberrant high expression of Cebpa/CEBPA at the MY1 + MY2 and MM stages (Fig. 2b) which

despite its substantial level of expression at both RNA and protein levels (Supplementary Fig. 4d, e) failed to completely compensate for the lack of CEBPE with respect to the promotion of cell cycle exit and the expression of late granulocytic differentiation genes. Consistent with the observed expression profiles, flow cytometry analyses further demonstrated expansion of monocytic differentiation at the expense of granulocytic differentiation (Supplementary Fig. 1b, c) which suggests the lack of CEBPE concomitant with persistent CEBPA expression promotes compensatory differentiation along the monocytic pathway. Notably, the aberrant high expression of CEBPA in Cebpe KO MY1s + MY2s and MMs (i.e., ~100,000 CEBPA copies per cell) was still 2.5- to 4.0-fold lower compared to the expression of CEBPE in Cebpe WT MY1s +MY2s and MMs (i.e., ~250,000–400,000 CEBPE copies per cell). The latter suggests that cell cycle exit and granulocytic differentiation are partially regulated in a dose-dependent manner by CEBPs. Consistently, transduction of KIT + Cebpe WT and KO BM cells with retroviral vectors expressing CEBPA-WT-ER[TM] and CEBPE-WT-ER[TM], demonstrated that 4-HT induction of CEBPA and CEBPE activity promoted terminal granulocytic differentiation at comparable levels (i.e., frequencies of induced-GRs/CD115-Ly6G+ cells). In contrast, only CEBPA induced monocytic differentiation of KIT + Cebpe WT and KO BM cells (i.e., frequencies of induced-MOs/CD115 + CD11b + cells) (Supplementary Fig. 4g, h).

Overall, these findings demonstrate that Cebpa/CEBPA expression in vivo promotes early and late monocytic differentiation as well as early granulocytic differentiation including specification of HSCs (i.e., LSKs) toward lineage-restricted progenitors (i.e., GPs and PMs) and expression of early granulocytic differentiation genes (i.e., primary granule proteins). In contrast, Cebpe/CEBPE expression is indispensable for sequential downregulation of G1/S and G2/M phase genes (i.e., at the PM to MY1 and the MY2 to MM transition) when cells exit the cell cycle, and for the upregulation of a substantial number of granulocytic differentiation genes as differentiation progresses and cells become fully mature GRs. Collectively, these findings suggest that CEBPE-dependent cell cycle exit is a prerequisite to complete terminal granulocytic differentiation in vivo.

**Regulation of lineage-specification and cell cycle exit.** To explore how CEBPE coordinates the regulation of gene expression with other TFs during granulocytic differentiation, we assessed differentially expressed genes in Cebpe WT and Cebpe KO and measured CEBPE binding by ChIP-seq at proximal (promoter; <1000 bp from the transcription start site) and distal (enhancer, N = 58,562; >1000 bp from the transcription start site, bound by PU.1, CEBPA, CEBPE, MYC, or E2F1 and overlapping with previously reported sets of enhancer regions[25,39,40]) cis-regulatory elements (CREs) of the genes from each of the five clusters in MY1s + MY2s. For simplicity, measurement of CEBPE ChIP-seq binding in MMs, BCs, and GRs were not included in this analysis since more than 90% of CEBPE binding sites in these populations were also bound by CEBPE in MY1s + MY2s (See below). Strikingly, CEBPE bound and regulated expression of genes (i.e., compared to Cebpe KO) expressed during early differentiation at the promoter level (cluster A(GMP) and cluster B(MY), Fig. 3a, b). In contrast, CEBPE regulated late differentiation genes (cluster C(GR) and cluster D(GR + MO)) via binding to their enhancers (Fig. 3a, b).

Early differentiation genes (i.e., clusters A(GMP) and B(MY)) which were downregulated during cell cycle exit in Cebpe WT, but not in Cebpe KO mice, exhibited enrichment of E2F binding motifs at their promoters (Fig. 3c and Supplementary Data 1). Strikingly, CEBPE binding at promoters of these early differentiation genes was most frequently directly overlapping with

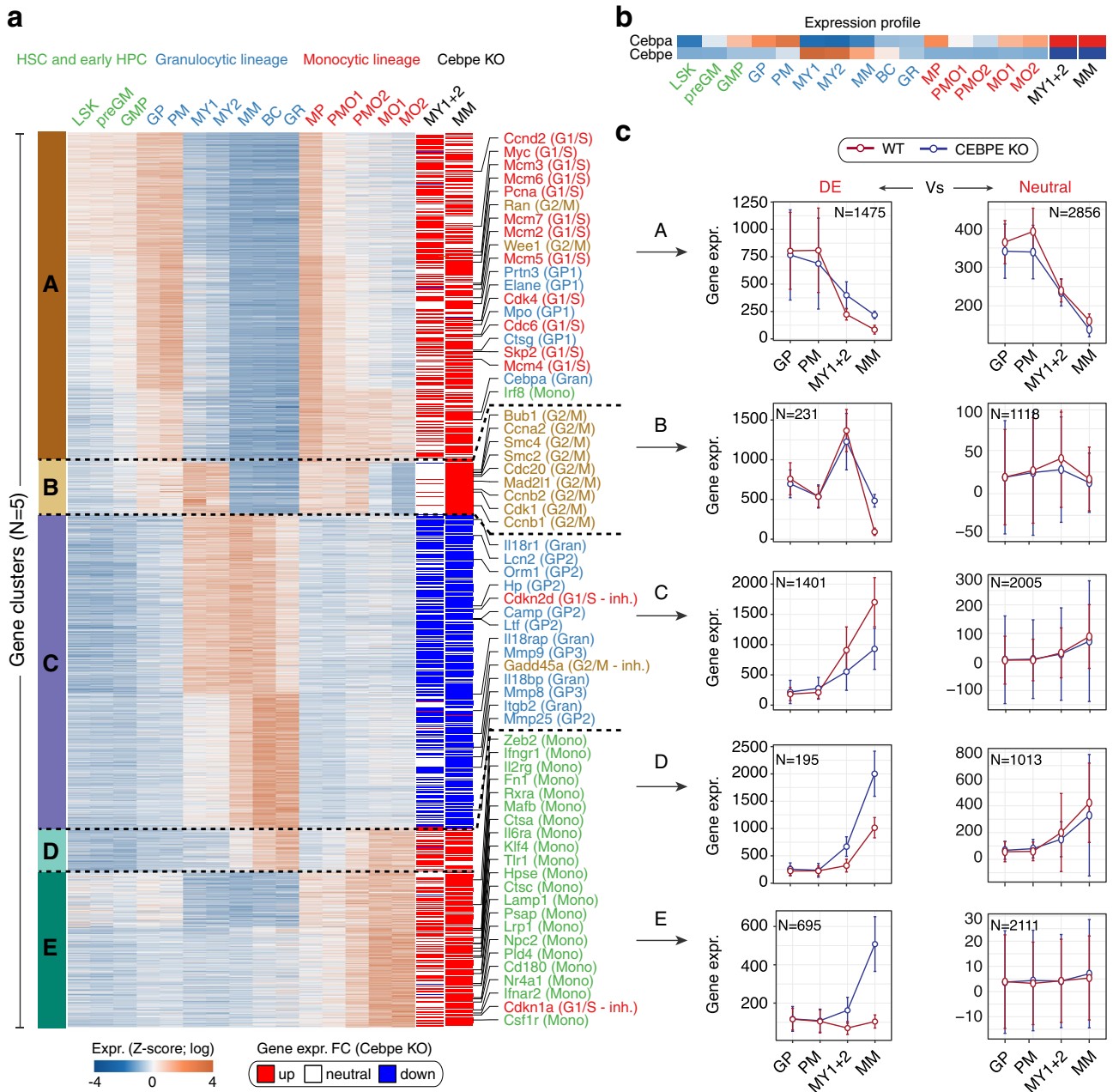

**Fig. 2 Five gene clusters stratified based on distinct gene expression profiles during early and late granulocytic and monocytic differentiation.**
**a** Unsupervised hierarchical clustering of RNA-seq gene expression profiles from sorted BM populations identified fifteen minor clusters (Supplementary Fig. 2) which based on similarity, were merged into the following five major gene expression clusters: Cluster A(GMP)—genes expressed during early granulocytic and monocytic differentiation (i.e., in LSKs, preGMs, GMPs, GPs, PMs, and MPs); cluster B(MY) and cluster C(GR)—genes transiently expressed in MY1 and MY2 and terminally upregulated in MM, BC, GR during late granulocytic differentiation, respectively; cluster D(GR + MO)—genes terminally upregulated during both late granulocytic and monocytic differentiation, and finally cluster E(MO)—genes upregulated exclusively during monocytic differentiation (i.e., in PMO1s, PMO2s, MO1s, and MO2s). Only differentially expressed (DE) genes are shown in the heatmap (all genes are shown in Supplementary Fig. 2). Gene symbols for selected genes representing (i) master regulators of G1/S and G2/M phase progression, (ii) sequentially expressed primary, secondary, and tertiary granule proteins (GP1, GP2, and GP3), and (iii) granulocytic (Gran) and monocytic (Mono) differentiation genes are depicted for each gene cluster. **b** RNA-seq gene expression profiles of *Cepba* and *Cebpe* in sorted *Cebpe* WT and *Cebpe* KO BM populations. **c** The line plots show the median expression profile of genes (median; whiskers represent the standard error) from each cluster whose expression is significantly differentially expressed (DE, left) or remains neutral (right) in BM populations of *Cebpe* KO as compared to *Cebpe* WT mice. Source data are provided as a Source Data file.

bona fide E2F1-bound regions (Fig. 3d, E2F1 ChIP-seq data from ESCs) and was independent of the presence of CEBP binding motifs (Fig. 3e). In line with this, analysis of *Cebpe* KO mice demonstrated selective upregulation of genes that were enriched for E2F binding motifs at their promoters (cluster B (MY),

Fig. 3c). This suggests that CEBPE interacts with E2F and confers repression of E2F-mediated target gene transcription at the promoter level. Subsequent functional validation utilizing EMSA and reporter assays in a granulocytic differentiation model (i.e., 32Dcl3-*CEBPE-ER*[TM] cell line)[41] confirmed that CEBPE directly

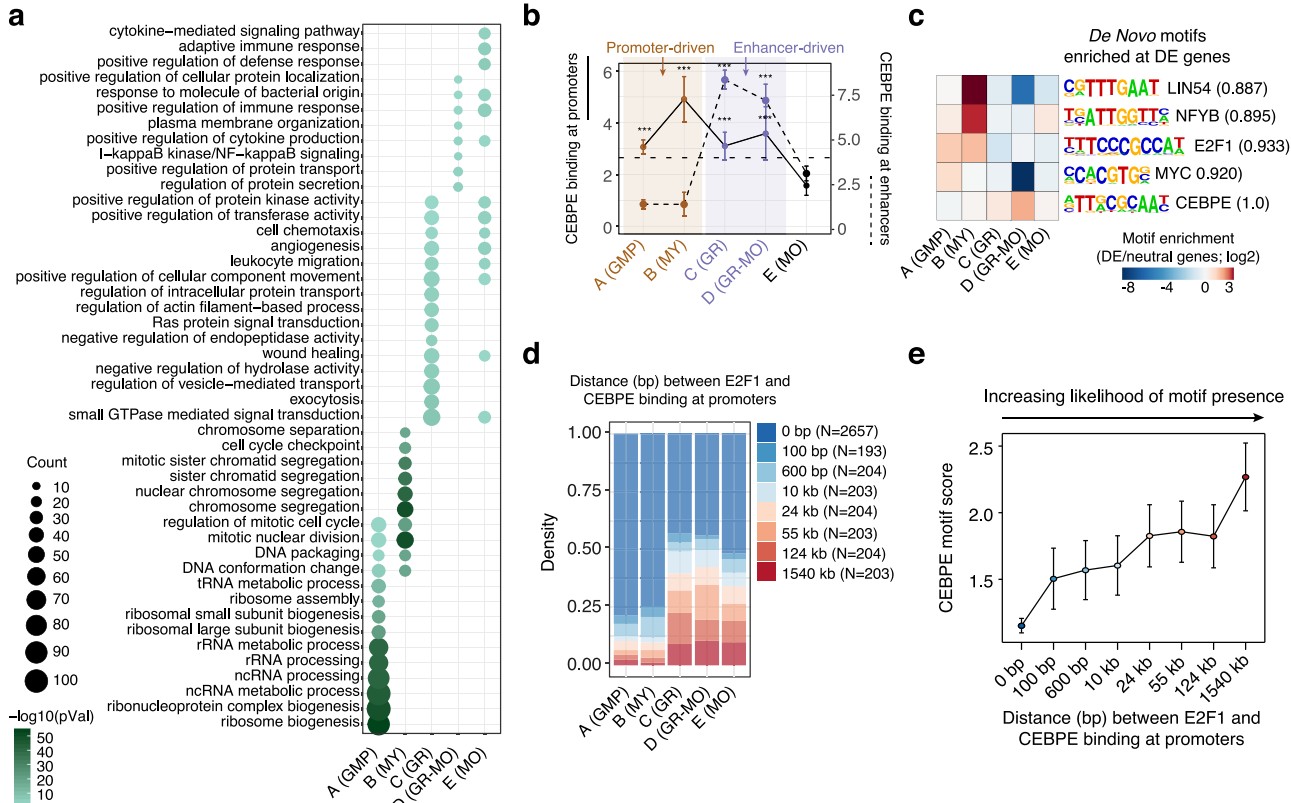

**Fig. 3 Specification of cell identity is regulated at the enhancer level and cell cycle exit at the promoter level. a** Functional gene categories of the five gene clusters depicted in Fig. 2a enriched for genes differentially expressed in *Cebpe KO* vs. *Cebpe* WT mice. **b** CEBPE binding signal (median) at promoters (solid lines) and enhancers (dashed lines) of DE genes from the five gene clusters shown in Fig. 2a (median; whiskers represent the standard error; $N = 1475$ (GMP), 231 (MY), 1401 (GR), 195 (GR-MO), and 695 (MO)). The CEBPE binding signal at enhancer(s) of each gene is normalized to the total number of enhancers linked to the respective gene. *** represent the significance level ($p < 0.0001$) at which the CEBPE binding at the promoters or enhancers of DE genes from a particular class is enriched compared to its binding at the promoters or enhancers of all DE genes ($N = 7903$). Wilcoxon test, one-sided, multiple-test corrected using Benjamini–Hochberg procedure. **c** Assessment of de novo binding motifs enriched at promoters of genes from all five gene clusters that are differentially expressed (DE) in *Cebpe KO* vs. *Cebpe* WT mice as compared to genes that were not differentially expressed (neutral). **d** CEBPE-bound gene promoters ($N = 4071$) are classified based on the base-pair (bp) distance between the closest E2F1-bound region. Shown for each gene cluster is the density distribution of each distance class (0–1540 kb). **e** CEBPE-bound gene promoters ($N = 4071$) are classified based on the distance to the closest E2F1 binding site. Shown for each distance class (0–1540 kb) is the median CEBPE motif score (mean; whiskers represent the standard error). A higher CEBPE motif score reflects CEBPE motif-assisted binding of CEBPE at a region. CEBPE binding to promoters with a motif score of 0 or with low CEBPE motif scores occurs in close proximity to E2F1 binding sites and is, therefore, more likely not assisted by a CEBPE motif but rather via indirect CEBPE binding to E2F at promoters. Source data are provided as a Source Data file.

bound to E2F complexes, repressed E2F-mediated gene expression in a dose-dependent manner and promoted growth arrest and granulocytic differentiation (Supplementary Fig. 5a–e).

In contrast to cell cycle genes, genes upregulated during late granulocytic differentiation (i.e., clusters C(GR) and D(GR + MO)) and downregulated in *Cebpe* KO mice (DE genes in Fig. 2c) were predominantly bound by CEBPE at enhancers. Significantly, motif analysis did not reveal enrichment of any other TF motifs at CEBPE-bound enhancers (Fig. 3c), suggesting that the specification of GR cell identity is primarily regulated at the enhancer level by CEBPE (Fig. 3a, b).

Overall, these findings demonstrate that regulation of differentiation is predominantly controlled at the enhancer level, while cell cycle exit is predominantly regulated at the promoter level, partially through repression of E2F-mediated transcription of key cell cycle regulators.

**CEBPE inhibits MYC-dependent G1/S phase progression**. To understand the mechanisms by which CEBPE coordinates temporal cell cycle exit in more detail, we systematically analyzed the

promoters of A(GMP) and B(MY) gene clusters, of which the former is enriched for the G1/S phase genes and later for the G2/M phase genes (Fig. 2a). TF motif analyses as well as ChIP-seq data (derived from ESCs) highlighted MYC and E2F as potential regulators of cluster A(GMP) genes (Fig. 3c and Supplementary Fig. 2f). Notably, cluster A(GMP) genes, including key G1/S phase genes demonstrated no CEBPE-dependent repression of E2F target genes (Fig. 4a and Supplementary Fig. 6b, d) in contrast to cluster B (MY) genes which indeed exhibited CEBPE-dependent E2F target gene repression (Fig. 4b and Supplementary Fig. 6c–e). In fact, genes downregulated concomitantly with the upregulation of *Cebpe*/CEBPE at the PM to MY1 transition were enriched for MYC binding at their promoters, pointing towards a direct or indirect regulation of MYC-dependent G1/S phase genes by CEBPE (Fig. 4a, c–e). The latter included several bona fide MYC G1/S target genes such as *Cdk4* and *Ccnd2* (Fig. 4f, g). *Myc* is a known target gene of E2F1, and analyses of the *Myc* promoter revealed direct binding of CEBPE at its E2F binding sites in MY1s + MY2s (Fig. 4h) when *Myc* expression ceased completely, which strongly suggests that CEBPE confers its inhibitory activity on MYC target genes by direct repression of E2F-dependent *Myc*

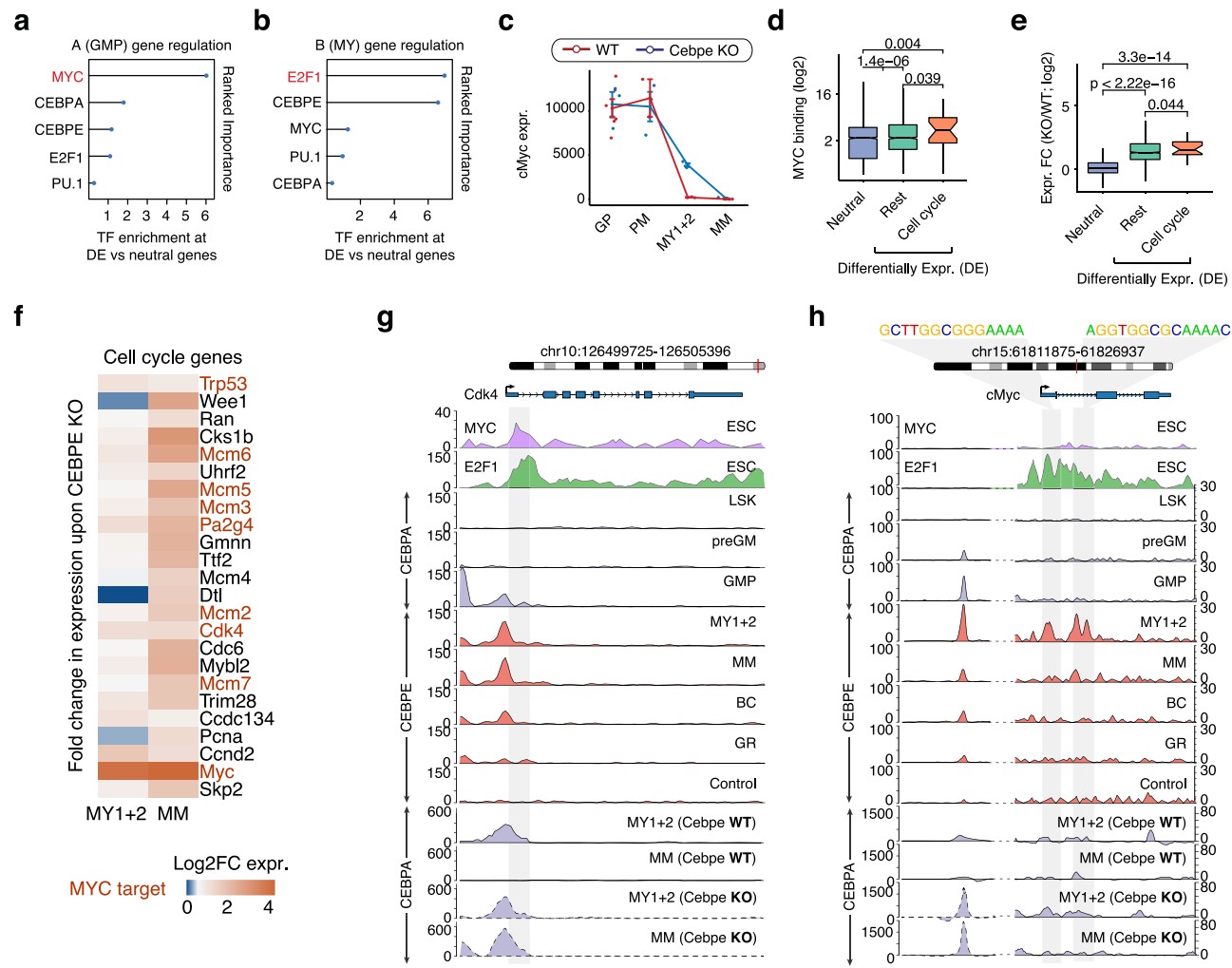

**Fig. 4 CEBPE inhibits G1/S progression by repression of MYC target gene expression. a**, **b** Ranked importance of five TFs (MYC, E2F1, CEBPE, CEBPA, and PU.1) in classifying genes from A(GMP) and B(MY) gene clusters that are differentially expressed (DE) in *Cebpe* KO vs. *Cebpe* WT mice as compared to neutral genes, respectively. Ranked importance is inferred by fitting a generalized regression model (GLM) to the binding affinities of the five TFs at gene promoters (DE vs. neutral). **c** Expression profile of *Myc* during normal granulocytic differentiation (i.e., in GPs (N = 6 WT and 3 KO), PMs (N = 2 WT and 3 KO), MY1s + My2s (N = 3 WT and 3 KO), MMs (N = 3 WT and 3 KO) biological replicates) in *Cebpe* WT and KO mice (mean; whiskers represent the standard error). **d** MYC binding levels at the promoter of genes (cluster A(GMP)) classified based on their log2-fold differential expression in *Cebpe* KO vs. *Cebpe* WT mice (DE vs. neutral). DE genes are further subdivided as key regulators of the G1/S phase vs. rest (center line, median; box limits, upper and lower quartiles; whiskers, 1.5x interquartile range) (N = 2856 neutral, 1452 rest, and 23 cell cycle) (Wilcoxon test, one-sided). **e** Same as (**d**), showing log2-fold DE genes subdivided as key G1/S phase regulators vs. rest as compared to neutral genes. **f** Aberrantly upregulated cell cycle genes (cluster A(GMP)) in MY1s + MY2s and MMs of *Cebpe* KO mice. Known MYC-target genes are highlighted in red. **g**, **h** Genome browser view of CEBPA, CEBPE, MYC, and E2F1 binding in WT and *Cebpe* KO mice (only CEBPA) to cis-regulatory elements of the G1/S phase cell cycle regulators, *Cdk4* (**g**) and *Myc* (**h**). Highlighted sequences represent E2F1 binding sites. Source data are provided as a Source Data file.

expression. CEBPE also initiated sustained promoter-driven upregulation of *Mad* (i.e., *Mxd1*) at the PM to MY1 transition, which in absence of MYC, might lead to the formation of repressive MAD/MAX dimers (instead of activating MYC/MAX dimers) and direct repression of MYC target gene expression (Supplementary Fig. 6f, g).

Taken together, these findings suggest that CEBPE represses E2F-mediated MYC expression and upregulates *Mad* expression at the PM to MY1 transition which ultimately initiates cell cycle exit by downregulation of MYC-regulated G1/S target genes.

Importantly, our data demonstrated that CEBPE also contributes to the inhibition of G1/S progression at another regulatory level by direct and sustained promoter-driven upregulation of the CDK4 and CDK2 inhibitors *Cdkn2d* (p19 INK4D) and *Cdkn1b* (p27 Kip1) at the PM to MY1 transition (cluster B(MY)) (Fig. 2a and Supplementary Fig. 6h).

**CEBPE inhibits E2F-dependent G2/M phase progression.** Whereas G1/S phase genes were downregulated at the PM to MY1 transition, genes promoting G2/M phase progression were still expressed at the MY1 and MY2 stages and markedly downregulated at the MY2 to MM transition (i.e., cluster B(MY)) concomitant with the completion of cell cycle exit. CEBPE ChIP analyses of G2/M gene promoters demonstrated frequent CEBPE binding at E2F binding sites in the absence of CEBPE motifs (Fig. 3d, e), which points toward a direct interaction of CEBPE with E2Fs at G2/M gene promoters concomitantly with their downregulation at the MY2 to MM transition (i.e., cluster B(MY)). The latter points to a direct CEBPE repression of E2F-mediated G2/M gene expression, which is supported functionally by the marked upregulation of G2/M genes in *Cebpe* KO mice at the MM stage (Fig. 2a, c) and the validation experiments described above (Supplementary Fig. 5a–e).

G2/M genes are partially regulated by E2Fs in combination with specific TFs, including the NFYA/B/C transcription factor trimer complex and variants of the MuvB complex. Specifically, expression of G2/M genes are inhibited in G0/G1 phase by the MuvB/DREAM complex, which binds to adjacent E2F4 and LIN54 (a regulatory DNA binding component of the MuvB complex) sites[42,43], and are upregulated in G2/M phase by activating E2F1-3 in combination with NFY[11]. Consistently, our TF motif analyses identified sites for E2F, NFYB, and LIN54 at bona fide G2/M gene promoters (Fig. 3c). While both *Nfyb* and *Lin54* are markedly upregulated in MY1s + MY2s, only the expression of *Lin54* is maintained throughout granulocytic differentiation, suggesting a transient activating function for *Nfyb* and a continuous repressive role for the MuvB/DREAM complex in sustaining permanent cell cycle exit (Fig. 5a). CEBPE binding to G2/M gene promoters only overlapped with regions bound by activating E2F1 and NFYB (E2F1 and NFYB ChIP-seq data from ESCs), but not with the LIN54 motif (Fig. 5b and Supplementary Fig. 7a–d) suggesting the co-regulation of G2/M genes by a CEBPE-E2F1-NFY module during cell cycle exit. To understand how CEBPE affects E2F- and NFY-dependent G2/M gene expression, we defined gene classes that were transiently highly expressed in MY1s and MY2s before cell cycle exit (cluster B(MY), Fig. 2a, $N = 1349$) based on all combinations of direct CEBPE binding and known E2F1, NFY, and LIN54 TF binding sites at their promoters. We next examined the impact of CEBPE on promoter class activity by assessment of gene expression changes in *Cebpe* KO vs. *Cebpe* WT (Fig. 5c, d). Notably, we observed the most significant differential expression of genes ($N = 76$) if their promoters exhibited binding/binding sites for all four TFs (Fig. 5c, CENL promoters). In contrast, genes whose promoters harbored binding/binding sites for CEBPE and E2F1, or alternatively individual TFs or none thereof (Fig. 5c; CE and Rest), exhibited no differential expression in *Cebpe* KO mice (Fig. 5c and Supplementary Fig. 7e). These findings were supported by the significantly higher levels of CEBPE, E2F, and NFY binding at CENL-bound promoters compared to the background (Rest), suggesting considerable levels of CEBPE, E2F, and NFY co-binding at the CENL-promoter class (Supplementary Fig. 7f–h). Importantly, CENL promoters were significantly more frequent among genes transiently highly expressed in MY1s and MY2s before cell cycle exit (cluster B(MY), Fig. 2a) as compared to genes from all other gene clusters (20% vs. 1–4%, Supplementary Fig. 7i). Gene ontology analysis confirmed that genes displaying CENL promoters are core regulators of G2/M phase progression and are strongly impacted by a loss of CEBPE (Fig. 5d and Supplementary Fig. 7j). These results suggest that CEBPE interacts with activating E2F1-3 in the proximity of NFY at promoters of G2/M genes and downregulate their expression leading to G2/M cell cycle exit, as exemplified by the *Ccnb1* and *Cdk1* genes (Fig. 5e, f).

Complementary analyses suggested that CEBPE also inhibited G2/M phase progression by promoter-driven upregulation of the CDK1 inhibitor *Gadd45a* at the MY2 to MM transition (cluster C (GR)) (Fig. 2a and Supplementary Fig. 7h).

**CEBPA primes and CEBPE activates GR cell identity genes.** Given that granulocytic differentiation genes were bound and regulated by CEBPs (Fig. 3b), we explored how *Cebpa*/CEBPA and *Cebpe*/CEBPE expression would dictate the CEBPA and CEBPE binding dynamics at their CREs. For this, we combined our previously generated CEBPA ChIP-seq analyses with the newly generated CEBPE ChIP-seq analyses of BM populations exhibiting expression of *Cebpa*/CEBPA and *Cebpe*/CEBPE (Fig. 6a–c)[25].

Intriguingly, both CEBPA and CEBPE exhibited preferential binding to enhancers as compared to promoters, and their sequential binding dynamics correlated with their expression profiles during early and late granulocytic differentiation as well as with the strength of the underlying CEBP sequence binding motif (Fig. 6d, Supplementary Fig. 8a, b and Supplementary Data 2).

Genes exclusively upregulated during late monocytic differentiation concomitant with sustained *Cebpa* expression (cluster E(MO), Fig. 2a, b) were primarily bound by CEBPA or both CEBPA and CEBPE (and not by CEBPE alone) at the enhancer level (CEBPA, CEBPA & E, Fig. 6d). In contrast, genes upregulated during granulocytic differentiation (cluster C (GR), Fig. 2a, b) were almost exclusively bound by CEPBE (CEBPE, Fig. 6d) or sequentially bound by both CEBPA and CEBPE at their enhancers during early (i.e., in LSKs, preGMs, GMPs) and late granulocytic differentiation (i.e., in MY1s + MY2s, MMs, GRs), respectively (CEBPA & E, Fig. 6d). Importantly, complementary ChIP-seq analyses of histone marks demonstrated that CEBPA is preferentially bound to primed enhancers in GMPs marked by high H3K4me1 and low H3K27ac binding, whereas CEBPE bound to active enhancers in MY1s + MY2s, MMs, GRs marked by high H3K27ac binding (Fig. 6d).

CEBPA ChIP-seq in *Cebpe* WT MY1s + MY2s and MMs revealed a clear loss of CEBPA binding at CREs concomitant with decreased *Cebpa*/CEBPA expression in these populations (Supplementary Fig. 4b, d, e). Complementary CEBPA ChIP-seq in *Cebpe* KO MY1s + MY2s and MMs exhibiting high *Cebpa*/CEBPA expression in absence of CEBPE (Supplementary Fig. 4c–e) demonstrated sustained CEBPA binding (similar to CEBPE) to these enhancers (CEBPA & E) (Fig. 6e, f). However, CEBPA failed to bind as well as activate enhancers exclusively bound by CEBPE (CEBPE, Fig. 6d) during terminal granulocytic differentiation in both *Cebpe* WT and KO MY1s + MY2s and MMs (Fig. 6e, f). Strikingly, the majority of these CEBPE-specific enhancers (71%) potentially regulate genes that are upregulated during terminal granulocytic differentiation (Fig. 6d).

A large proportion of all granulocytic differentiation genes (i.e., 41% or 1401 genes of cluster C(GR), Fig. 2a) were downregulated in *Cebpe* KO mice (Figs. 2c, 7a), and exhibited significantly higher levels of CEBPE binding at CREs compared to genes not differentially expressed in *Cebpe* KO mice (Fig. 7b, c, *Mmp9* example in 7d). Complementary Gene Ontology analyses demonstrated that these CEBPE target genes are involved in exocytosis of granules, leukocyte migration, chemotaxis, immune response, wound healing, and cell signaling (Fig. 3a).

To further validate the role of CEBPE binding at CREs we selected genes bound by CEPBE at promoters and enhancers in vivo during terminal differentiation and tested their expression using the 32Dcl3-*CEBPE-ER*$^{TM}$ cell line, which differentiates in response to 4-hydroxy-tamoxifen (4-HT) induced translocation of CEBPE-ER$^{TM}$ to the nucleus. We identified three putative promoter-regulated genes (*Hp*, *Dhrs7*, *Hlx*) and one promoter-/enhancer-regulated gene (*Dhrs7*) that were upregulated after 2 or 4 days of 4-HT induced CEBPE differentiation and whose expression were substantially reduced by CRISPR interference (CRISPRi) of their CEBPE-bound promoter (*Hp*, *Dhrs7*, *Hlx*) and enhancer (*Dhrs7*) sites (Supplementary Fig. 8c, e).

Overall, these findings demonstrate that CEBPA and CEBPE primarily bind sequentially to enhancers and prime as well as activate expression of all major functional gene categories that specify GR cell identity during early and late granulocytic differentiation, respectively.

**Reciprocal regulation of *Cebpa* and *Cebpe* expression.** Intriguingly, detailed analyses of the CEBP enhancer landscape

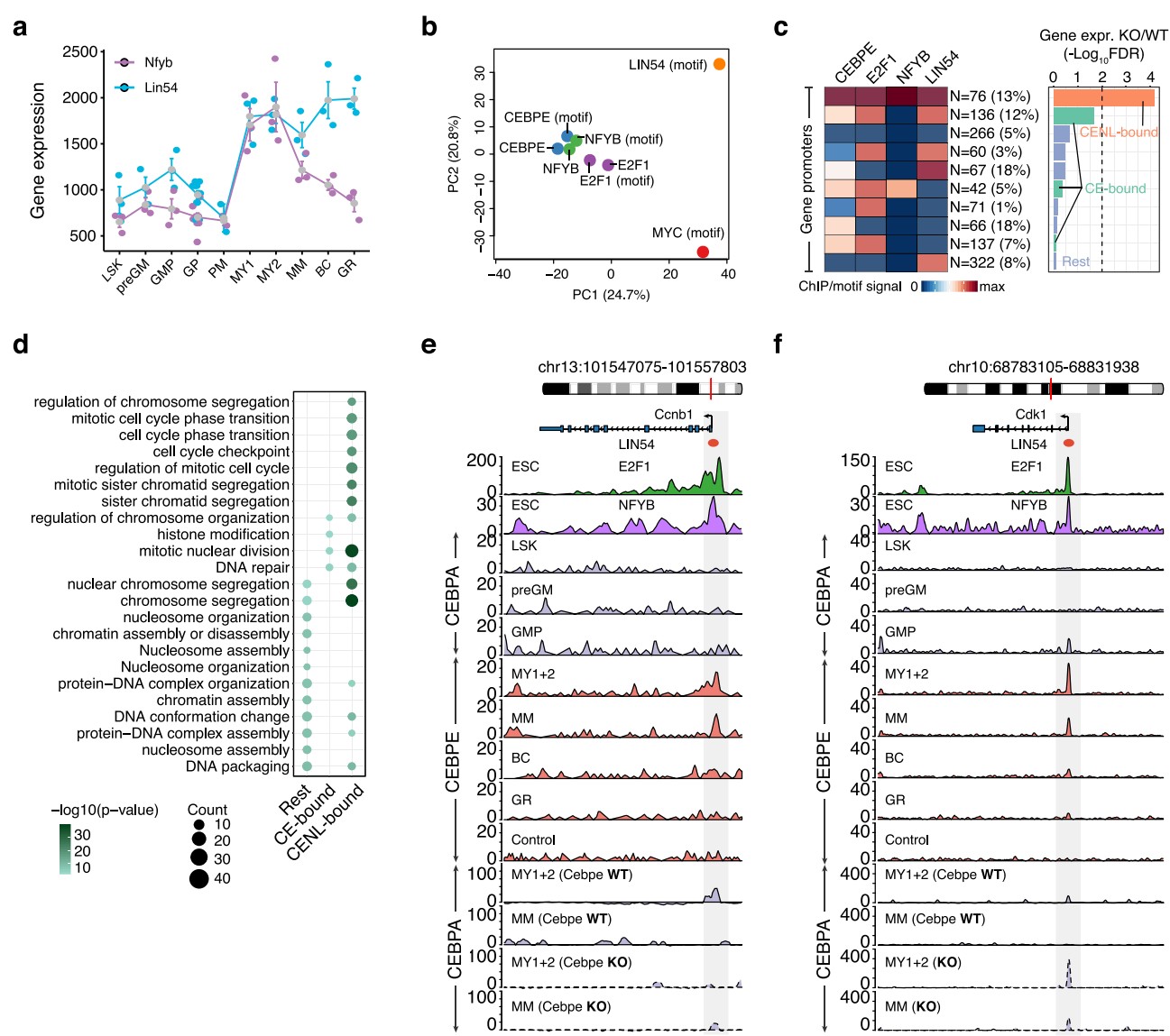

**Fig. 5 CEBPE inhibits G2/M progression by repression of E2F target gene expression. a** Expression profiles of *Nfyb* and *Lin54* during granulocytic differentiation (mean; whiskers represent the standard error) (*N* = 3 (LSK), 3 (preGM), 3 (GMP), 6 (GP), 2 (PM), 3 (MY1), 3 (MY2), 3 (MM), 3 (BC), and 3 (GR), biological replicates). **b** PCA plot showing similar binding profiles for CEBPE, NFYB, and E2F1, but not for MYC and LIN54, at promoters of G2/M genes, as inferred by the ChIP-seq binding signal and binding motif scores. **c** Cluster B(MY) genes, including G2/M phase genes, were divided into gene subclasses based on the combinatorial binding of four TFs (CEBPE = C, E2F1 = E, NFYB = N, LIN54 = L) (*N* > 30). CENL subclass (orange): Binding of all four TF. CE subclass (green): Binding of C and E with or without additional binding of either N or L. Rest (blue): Binding of other TF combinations of C, E, N, L, or none of these. C*ebpe* KO mice demonstrated marked upregulation of genes whose promoters were bound by all the four TFs (i.e., CENL-bound genes). In contrast, CE-bound and the "Rest" group of genes exhibited minimal upregulation in *Cebpe* KO vs. *Cebpe* WT mice. The percentages of genes from each TF binding subclass are depicted in brackets. **d** Functional gene categories enriched among genes from TF binding subclasses. **e, f** Genome browser view of two key G2/M phase regulators, *Ccnbb 4* (**e**) and *Cdk1* (**f**) bound by all four TFs at their promoters (CENL subclass) in WT and *Cebpe* KO mice (only CEBPA). Source data are provided as a Source Data file.

demonstrated autoregulation of CEBPA by binding to its own +37 kb enhancer at the GMP stage (Supplementary Fig. 8f)[44]. Notably, CEBPA also bound to a +6 kb enhancer of *Cebpe* at the GMP stage, which was subsequently autoregulated by CEBPE itself at the MY1 + MY2, MM, and BC stages (as evidenced by previously published CRISPRi experiments), resulting in high levels of *Cebpe*/CEBPE expression (Supplementary Fig. 8g)[45]. As stated above, CEBPE promotes cell cycle exit not only by repression of E2F-mediated *Myc* expression but also by upregulation of *Mad* (Supplementary Fig. 6g) which, in the absence of MYC can heterodimerize with MAX, potentially leading to inhibition of *Myc* target gene expression. Since the *Cebpa*

promoter harbors activating MYC/MAX binding sites[46,47] and *Cebpa*/CEBPA expression was downregulated in parallel with *Myc* and other *Myc* target genes during cell cycle exit, CEBPE potentially confers indirect MYC- and MAD-dependent repression of *Cebpa* expression (Supplementary Fig. 8f). Indeed, a reporter assay confirmed the functional importance of an MYC binding site in the *Cebpa* promoter (Supplementary Fig. 8d). The latter favors the above hypothesis that CEBPE potentially confers indirect repression of MYC target genes such as *Cebpa*.

Hence, both CEBPA and CEBPE promote autoregulation and might potentially regulate one another by reciprocal activation and feedback inhibition, respectively.

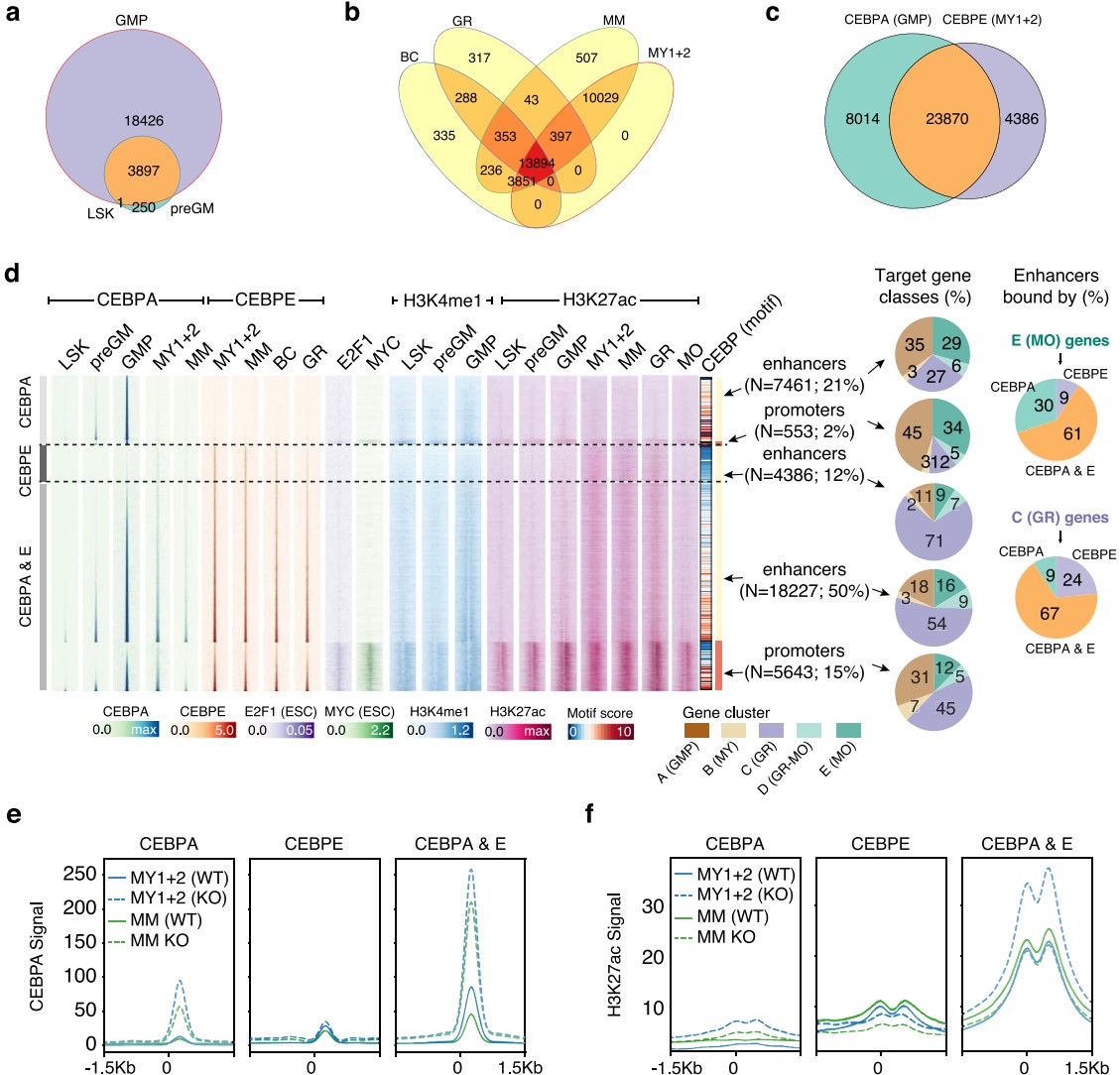

**Fig. 6 CEBPA and CEBPE binding dynamics to cis-regulatory elements during early and late granulocytic differentiation. a** Regions bound by CEBPA during early granulocytic differentiation in LSKs, preGMs, and GMPs demonstrate significant overlap and the most abundant binding of CEBPA in GMPs. **b** Regions bound by CEBPE in MY1s + MY2s, MMs, BCs, and GRs demonstrate significant overlap of CEBPE binding and the most abundant binding of CEBPE in MY1s + MY2s and MMs. **c** Regions bound by CEBPA in GMPs and by CEBPE in MY1s + MY2s demonstrate a high overlap of CEBPA and CEBPE binding during early late granulocytic differentiation. **d** Heatmap showing the levels of CEBPA, CEBPE, H3K4me1, and H3K27ac at promoters and enhancers (>1000 bp from TSS) for indicated BM populations, along with their target gene clusters. Shown alongside the CEBPE binding motif score for each region are the E2F1 and MYC binding affinities for the same regions in embryonic stem cells (ESC). Also shown are the H3K4me1 (priming marker) and H3K27ac (activation marker) levels at enhancers for indicated BM populations. **e**, **f** CEBPA binding (**e**) and H3K27ac modification (**f**) levels at regions bound by CEBPA and/or CEBPE (i.e., CEBPA & E, CEBPA, CEBPE) for the indicated populations. Source data are provided as a Source Data file.

## Discussion

Differentiation of multipotent stem cells into mature cells is fundamental for the development and homeostasis of mammalian tissues. This process requires temporal coupling of cell cycle exit and differentiation and is tightly coordinated by regulatory networks inhibiting cell cycle master regulators concomitant with activation of distinct transcriptional programs defining the identity of tissue-specific cell types. Although many cell type-specific TFs have been identified as master regulators of differentiation networks, we still have a poor understanding of the temporal sequence and combination of molecular mechanisms launched by these TF to coordinate differentiation and cell cycle exit during mammalian tissue development and maintenance in vivo[3].

In the present study, we applied a granulocytic differentiation model to explore how tissue-specific TFs coordinate cell cycle exit

and differentiation. To this end, we specifically focused our efforts on the establishment of a protocol for sorting of a unique hierarchy of ten immunophenotypically defined developmental stages of granulocytic differentiation. Notably, our sorting protocol included LSKs and preGMPs as well as a hierarchy of eight developmental stages of granulocytic differentiation (i.e., from GMPs to GRs) as compared to four, five, and six developmental stages reported for recent comprehensive sorting protocols and a recent scRNA-seq study (i.e., GMPs to mature Neus, Supplementary Fig. 3)[22–24]. Hence, our sorting protocol likely provides a granulocytic differentiation model at a hitherto unprecedented resolution. Indeed, the latter was corroborated by annotation of single-cell RNA-seq profiles to bulk RNA-seq profiles of sorted BM populations, which confirmed that the sorted BM populations faithfully recapitulate the developmental trajectories of early and late granulocytic differentiation.

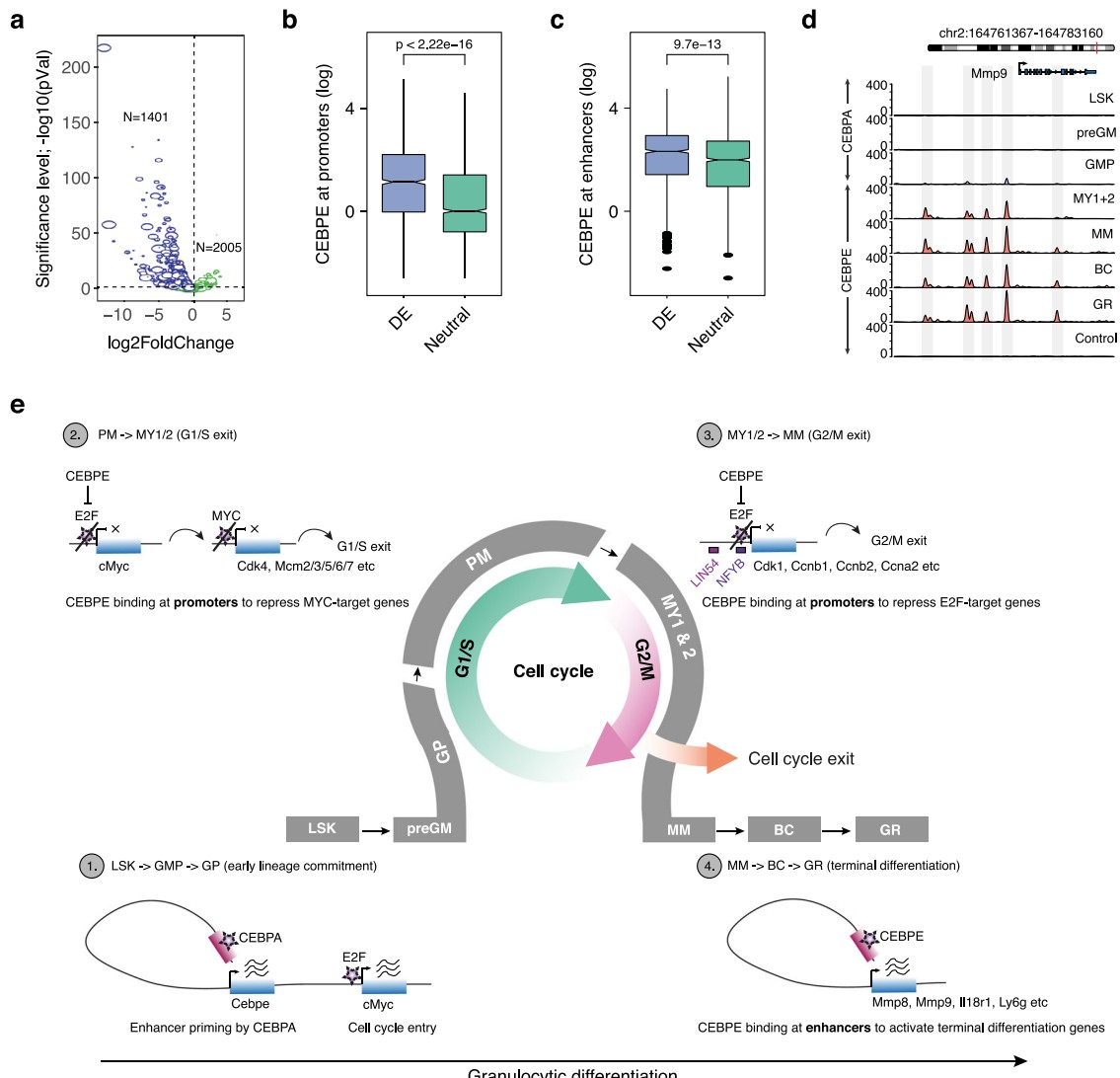

**Fig. 7 CEBPE promotes terminal granulocytic differentiation through enhancer-mediated regulation of gene expression. a** Volcano plot showing log2-fold-change in the expression of late granulocytic differentiation genes (cluster C(GR)) differentially expressed (DE) in *Cebpe* KO vs. *Cebpe* WT mice. Approximately 41% of late granulocytic differentiation genes are downregulated in *Cebpe* KO mice, whereas the residual genes exhibit similar expression. **b**, **c** Genes downregulated in *Cebpe* KO mice (*N* = 1401) demonstrate significantly higher levels of CEBPE binding, both at promoters (**b**) and enhancers (**c**), in comparison to genes whose expression remains neutral (*N* = 2005) (center line, median; box limits, upper and lower quartiles; whiskers, 1.5x interquartile range; points, outliers) (Wilcoxon test, one-sided). **d** Genome browser view of the tertiary granule protein—matrix metallopeptidase 9 (Mmp9) which is markedly upregulated during late granulocytic differentiation (cluster C (GR), Fig. 2a) and has multiple CEBPE binding sites. **e** Unified model depicting how the tissue-specific TFs, CEBPA, and CEBPE, coordinate sequential lineage-specification (step 1), growth arrest (steps 2 and 3), and lineage-determination (step 4) during early and terminal granulocytic differentiation. Source data are provided as a Source Data file.

Utilizing this differentiation model, we directed a systematic approach to assess how designated tissue-specific TFs (i.e., CEBPA and CEBPE) orchestrate cycle exit and specification of cell identity at the promoter and enhancer levels during differentiation in vivo.

Intriguingly, CEBPA and CEBPE were expressed in an almost exclusive sequential manner and preferentially bound to enhancers of target genes exhibiting CEBP binding motifs during early and late differentiation, respectively. Notably, shared CEBPA and CEBPE target genes defined a core set of GR-specific genes (i.e., 1401 genes) which were not expressed upon CEBPA binding during early differentiation, but almost exclusively upregulated by CEBPE during terminal postmitotic differentiation. Notably, enhancers of these GR-specific genes were primed (i.e., H3K4me1 marked) concomitant with CEBPA binding during early differentiation (i.e., in GMPs) and activated (i.e., H3K27ac marked)

concomitant with CEBPE binding during late granulocytic differentiation (i.e., MY1s + MY2s, MMs, GRs) (Fig. 6d). Given the ability of CEBPA to recruit the SWI/SNF chromatin remodeling complex, our findings suggest that CEBPA primes progenitor cells for granulocytic differentiation by launching a GR-specific chromatin state at the enhancer level[48,49]. Consistently, studies of muscle development in *Caenorhabditis elegans* have demonstrated that muscle-specific TFs recruit SWI/SNF leading to chromatin remodeling and ultimately sequential cell cycle exit and activation of a muscle-specific differentiation program[1,50]. In this context, our findings support a model where TFs, indispensable for tissue-specific development, direct somatic stem/progenitor cells toward lineage-specific progenitors exhibiting a full enhancer-primed differentiation program predestined for postmitotic activation and completion of differentiation. Such a model is supported by embryonic stem cell studies,

demonstrating that sequential binding of TFs to tissue-specific enhancers establishes an early tissue-specific chromatin state before they actually launch a full transcriptional differentiation program and differentiate into the various types of tissue-specific cells[51–53].

In contrast to differentiation genes, cell cycle genes were predominantly regulated at the promoter level during a narrow developmental window at the intersection of early vs. late granulocytic differentiation. The high resolution of our differentiation hierarchy demonstrated an abrupt termination of Cebpa/CEBPA expression and onset of Cebpe/CEBPE expression at the PM to MY1 transition, suggesting that this step is crucial for the initiation of terminal granulocytic differentiation. Consistent with previous in vitro studies demonstrating the binding of CEBPs to E2F[16,17,36], our in vivo ChIP-seq analyses revealed significant overlapping of CEBPE binding at bona fide E2F promoter binding sites without the presence of CEBP binding motifs. Complementary functional analyses revealed upregulation of several identified CEBPE/E2F target genes in Cebpe KO mice, CEBPE binding to E2F, and CEBPE-driven repression of E2F-mediated transcription in reporter assays. Together, these findings strongly suggest that CEPBE regulates cell cycle exit at the promoter level during in vivo granulocytic differentiation through interaction with E2F and repression of E2F-mediated transcription.

Notably, key G1/S phase genes including CDKs (Cdk4/6), Cyclins (Ccnd1/d2), and Myc were markedly downregulated concomitantly with the switch of CEBPA and CEBPE expression at the intersection of early vs. late differentiation. In contrast, G2/M phase genes were downregulated after cessation of G1/S phase gene expression in parallel with completion of cell cycle exit. This sequential progression of cell cycle exit points to distinct regulatory mechanisms for lock-down of G1/S and G2/M phase genes during terminal differentiation. To decipher potential regulatory mechanisms, we conducted elaborate analyses on cell cycle gene promoters. Analyses of G1/S phase gene promoters suggest that the direct binding of CEBPE at E2F sites of the Myc promoter leads to repression of Myc expression concomitant with the downregulation of several designated MYC target genes critical for G1/S phase progression. Importantly, CEBPE also conferred direct activation of Mad expression which, in the absence of MYC, can dimerize with MAX and directly repress MYC target gene expression during cell cycle exit and terminal differentiation. Strikingly, these findings link a tissue-type-specific TF (i.e., CEBPE) to the MYC–MAD differentiation switch at the onset of cell cycle exit[54].

Complementary promoter analyses of G2/M phase genes demonstrated binding of CEBPE at their E2F binding sites in absence of any CEBPE binding motifs. These CEBPE-bound E2F sites were adjacent to NFYB sites at promoters of G2/M phase genes, such as Ccnb1 and Cdk1, which were all downregulated upon completion of cell cycle exit. Given that activating (but not inhibiting) E2Fs are dependent on adjacent NFY binding at promoters to activate G2/M gene transcription, our findings suggest a direct CEBPE repression of E2F-mediated G2/M gene expression[11]. Indeed, the latter was supported functionally by the marked upregulation of G2/M genes in Cebpe KO mice, concomitant with sustained proliferation and a block of differentiation at the MM stage. In addition to its role as a repressor of E2F target gene expression, CEBPE also regulated cell cycle exit by sustained direct activation of CDK4, CDK2, and CDK1 inhibitor expression (Cdkn2d, Cdkn1b, and Gadd45a) concomitant with cell cycle exit and terminal differentiation. This hitherto undescribed regulatory mechanism of CEBPE is in agreement with previous reports of cell type-specific TFs promoting growth arrest

by upregulation of CDK inhibitor expression[14,15]. Previous studies have demonstrated that both CEBPA and CEBPE can repress E2F target gene expression and exhibit overlapping expression profiles in granulocytic differentiation models, suggesting that both TFs contribute to the regulation of cell cycle exit during granulocytic differentiation[17,36]. However, these findings are challenged by the high resolution of our in vivo differentiation model, which clearly demonstrates termination of Cebpa/CEBPA expression at the onset of Cebpe/CEBPE expression concomitant with initiation of cell cycle exit in vivo the PM to MY1 transition. These somewhat unexpected findings are supported by genetic studies demonstrating that CEBPA is indispensable for lineage-specification of HSCs toward GMPs, but is not required for GMPs to exit the cell cycle exit and complete terminal granulocytic differentiation[25]. Importantly, the ability of CEBPA to promote granulocytic differentiation of Cebpe KO cells in vitro while failing to do so in vivo might indeed reflect the 2.5- to 4-fold lower levels of CEBPA protein in Cebpe KO MY1s + MY2s as compared to CEBPE in Cebpe WT MY1s + MY2s. Intriguingly, ChIP-seq experiments demonstrated that CEBPE is bound to a distinct set of enhancers and promoters of granulocytic differentiation genes not bound by CEBPA, suggesting that CEBPE and CEBPA to some extent differ functionally. While these findings suggest that cell cycle exit and completion of granulocytic differentiation might partially be regulated in a dose-dependent manner by CEBPs, they do not provide any information on potential functional specificities.

Conceptually, the almost exclusive sequential expression of Cebpa/CEBPA and Cebpe/CEBPE might rely on distinct sequential cis-regulatory events. Consistently, we identified sequential CEBPA and CEBPE binding to a reported functional 6 kb enhancer of Cebpe[45]. The latter suggests that CEBPA initiates Cebpe expression, which is subsequently enhanced through CEBPE autoregulation at the onset of cell cycle exit. As stated above, CEBPE promotes cell cycle exit, not only by repression of E2F-mediated Myc expression, but also by upregulation of Mad which, in absence of MYC can heterodimerize with MAX leading to inhibition of Myc target gene expression. Given that the Cebpa promoter harbors functional MYC/MAX—MAD/MAX binding sites (Supplementary Fig. 8f) and Cebpa/CEBPA was downregulated in parallel with Myc and several Myc target genes during cell cycle exit, CEBPE potentially confers indirect MYC- and MAD-dependent feedback inhibition of Cebpa expression to establish a point-of-no-return for cell cycle exit and terminal differentiation[46,47]. Intriguingly, this switch of Cebpa and Cebpe expression might also be critical to overcome persistent CEBPA binding to promoters and enhancers of monocytic genes, once CEBPE initiates cell cycle exit and expression of granulocytic genes during terminal differentiation. Hence the sequential expression of Cebpa/CEBPA and Cebpe/CEBPE, observed in our study, represents a unidirectional fail-safe mechanism of how TFs coordinate growth arrest and differentiation.

Overall, our study provides insights into how key TFs, which are indispensable for tissue-specific differentiation, prime stem/progenitor cells for lineage-commitment, regulate cell cycle exit by distinct mechanisms, and promote a postmitotic differentiation program to generate fully mature tissue-specific cells (see a unified model in Fig. 7e). Importantly, our work also demonstrates the selective TF usage of promoters and enhancers to regulate cell cycle exit and to promote the expression of lineage-specific differentiation programs, respectively. In a broader perspective, our work contributes to the understanding of the temporal regulatory mechanisms driving differentiation, which ultimately might have an impact on the development of differentiation therapies for cancer patients[55].

## Methods

**Mice.** *Cebpe* KO mice and littermate controls were maintained on a C57BL/6 WT background and housed according to institutional guidelines at the University of Copenhagen[26]. Experiments were performed with 10–12 weeks old male and female *Cebpe* KO mice and littermate controls. All experiments were approved by the Danish Animal Research Ethical Committee (license no. 2012-15-2935-0001).

**Immunophenotypic characterization and purification of murine BM populations.** Flow cytometry-based analyses and cell sorting of LSKs, preGMPs, and GMPs were carried out as previously described[25,37,56]. Briefly, murine BM cells were collected from tibiae, femur, and ilia of 8–10 weeks old mice. Subsequently, BM cells were KIT-enriched using anti-CD117-MoAb microbeads (Miltenyi Biotec, Bergisch Gladbach, DE). Whole BM cells or KIT-enriched BM cells (planned for sorting) were subsequently stained with the following cocktail of fluorochrome-conjugated antibodies: CD150-APC (1:200, clone TCF15-12F12.2; BioLegend, San Diego, CA, USA), CD41-FITC (1:200, clone MWReg30), CD105-PE or CD105-PE-Cy7 (1:200, clone Mj7/18), CD115-PE (1:200, clone AFS98), Sca1-PerCP-Cy5 (1:200, clone D7), FcgRII/III-A700 (1:50, clone 93), Kit-A780 (1:200, clone 2B8) and a lineage cocktail consisting of the following PE-Cy5 conjugated antibodies: Ter119-PE-Cy5 (1:400, clone Ter119, BioLegend), Gr1-PE-Cy5 (1:400, clone RB6-8C5), B220-PE-Cy5 (1:400, clone RA3-6B2), CD3e-PE-Cy5 (clone 145-2C1) and Mac1-PE-Cy5 (1:800, clone M1/70, BioLegend).

For analyses and sorting of BM populations representing sequential developmental stages of late granulocytic (GP, PM, MY1, MY2, MM, BC, GR) and late monocytic (MP, PMO1, PMO2, MO1, and MO2) differentiation hierarchies, BM cells were subjected to red blood cell lysis with PharmLyse (BD Biosciences, San Jose, CA, USA). BM cells were then stained using the following fluorochrome-conjugated antibodies: CD34-FITC (clone RAM34, 1:25), CD115-PE (clone AFS98, BioLegend, 1:100), Ter119- PE-Cy5 (clone Ter119, BioLegend, 1:400) B220-PE-Cy5 (clone RA3-6B2, 1:400), CD3e-PE-Cy5 (clone 145-2C1, 1:400), NK1-PE-Cy5 (clone PK136, BioLegend, 1:200), Sca1-PerCP-Cy5.5 (clone D7, 1:200), CD11b-PE-Cy7 (M1/70, 1:200), KIT-APC (clone 2B8, 1:200), FcgRII/III-A700 (clone 93, 1:50), and Ly6G-APC-Cy7 (clone 1A8, BioLegend, 1:50). Antibodies were provided by eBiosciences (eBiosciences/ Thermo Fisher Scientific, Waltham, MA, USA) unless otherwise stated.

To assess the proliferation rates for individual BM populations, BM samples were surface-stained as described above and subjected to intracellular DNA staining with DAPI (4′,6-diamidino-2-phenylindole, dihydrochloride, Invitrogen Molecular Probes, Eugene, OR, USA) as described previously by our laboratory[21].

Flow cytometry analyses of cells were carried out on BD LSR II or BD ARIA III flow cytometers (BD Biosciences, San Jose, CA, USA) according to the gating strategy depicted in Supplementary Fig. 1a, d. Viable BM populations were sorted on a BD ARIA I or III flow cytometer by the exclusion of non-viable 7AAD + cells (1 µg/mL, 7-amino-actinomycin D, Invitrogen, Carlsbad, CA, USA). FlowJo analysis software (Version 10.1, TreeStar Inc., San Carlos, CA, USA) was used for subsequent analysis of flow cytometry data. Gates defining positive and negative populations for specific markers were set according to fluorescence-minus-one (FMO) controls stained with isotype-matched control antibodies.

The morphologies of sorted BM populations were assessed by microscopy of Wright-Giemsa stained cytospins as described previously[21].

**mRNA sequencing analyses (mRNA-seq).** Total RNA was purified from sorted BM populations using the RNeasy Micro Kit (Qiagen, Aarhus, DK) as described by the manufacturer. RNA was subjected to double-stranded cDNA synthesis using the Ovation RNA-Seq System V2 (NuGEN, San Carlos, CA, USA), and cDNA libraries were prepared from sheared cDNA fragments (150–550 bp) using the Ovation Ultralow System V2 (NuGEN). The indexed cDNA libraries were pooled in equimolar ratios and subjected to 75-cycles of sequencing at Exiqon (Exiqon now at Qiagen) on an Illumina NextSeq 500 Sequencing System (Illumina, San Diego, CA, USA).

**Chromatin immunoprecipitation sequencing (ChIP-seq).** CEBPA, H3K4me1, and H3K27ac ChIP-seq analyses of sorted BM populations (i.e., LSKs, PreGMPs, and GMPs) have been described previously by our group[25]. CEBPE, CEBPA, and H3K27ac ChIP-seq analyses of sorted BM populations (i.e., MY1 + MY2, MM, BC, and GR) were conducted, essentially as described previously by our group[25,57,58]. In brief, we used 300,000–800,000 sorted MY1 + MY2s (pooled MY1s + MY2s), MMs, BCs, and GRs from *Cebpe* WT mice for indicated CEBPE and CEBPA ChIP-seq experiments, and 100,000 sorted cells for H3K27ac ChIP-seq experiments. 1,000,000 pooled MYs and MMs purified from *Cebpe* KO mice were used as a negative control for CEBPE ChIP-seq analyses. Chromatin was incubated with an antibody targeting CEBPE (1:1000, clone H-75, Santa Cruz Biotechnology, Dallas, TX, USA, discontinued), CEBPA (1.5:1000, clone 14AA, Santa Cruz Biotechnology, discontinued), and H3K27ac (1:10,000, ab4729, Abcam, Cambridge, UK). Antibody-bound chromatin was captured with a mixture of Protein-A/G sepharose beads (1:25) or a mixture of Dynabeads™ Protein-A/G (1:100) (Invitrogen/ Thermo Fisher Scientific), washed, de-crosslinked, and precipitated. If the amount of precipitated DNA was higher than 2 ng, DNA was directly amplified. If the amount of DNA was below 2 ng, it was mixed with fragmented *E. Coli* DNA to

yield a tot of 2 ng before amplification using the NEBNext ChIP-seq Library Prep Kit for Illumina (discontinued) or NEBNext Ultra™ II DNA Library Prep Kit for Illumina (New England Biolabs, Ipswich, MA, USA) as described by the manufacturer. DNA libraries were sequenced on an Illumina HiSeq 2500 or NextSeq 500 Sequencing System.

**Luciferase reporter assay.** Q2bn fibroblasts (kindly provided by Claus Nerlov, MRC Weatherall Institute of Molecular Medicine, University of Oxford, John Radcliffe Hospital, Headington, UK) were cultured at 37 °C and 5% CO$_2$ in DMEM supplemented with 8% FBS and 2% chicken serum (InVitrogen/Thermo Fisher Scientific). Q2bn fibroblasts were transiently transfected by calcium-phosphate precipitation, grown for 24 h, and assayed for reporter gene expression as described previously[16]. The following plasmids were used for transient transfection experiments: p*E2Fx6*-TATA-LUC reporter, pCMV-*E2F1*, pCMV-*DP1* expression vectors, and pCMVneoBam control vector (all generously provided by Kristian Helin, BRIC, Univ. of Copenhagen, DK). The pcDNA3-*CEBPE* and pcDNA3-*CEBPE*-*BRM5* expression vectors were generated by insertion of full-length human *CEBPE* or *CEBPE*-BRM5 into the pcDNA3 vector (pcDNA3, InVitrogen/Thermo Fisher Scientific). Empty pcDNA3 vector was used to adjust the total amount of DNA per dish and the beta-galactosidase reporter plasmid pRSV-βGAL, was used to normalize Luciferase activity.

Human embryonic kidney (HEK) 293 cells (ACC 305, DMSZ-German Collection of Microorganims and Cell Cultures GmBH, Braunschweig, DE) were cultured in DMEM supplemented with 10% FCS at 37 °C 5% CO$_2$. HEK293 cells were plated in 96-well tissue culture and transiently transfected with the following vectors: 0.2 ng of *Renilla* control vector (pRL-CMV, Promega, Madison, WI, USA), 20 ng of empty pGL4 vector or pGL4 vector containing a *Cebpa* promoter 57 base-pair fragment with or without its MYC-binding E-box (pGL4:23, Promega), and 20 ng of either empty or MYC containing pcDNA3 vector (Thermo Fisher Scientific, Addgene, MA, USA). Vectors were transfected using a 3:1 µl/µg ratio of TransIT-2020 reagent as described by the manufacturer (MIR 5404, Mirus Bio Madison, WI, USA). After 2 days of incubation, dual-luciferase assays were performed according to the manufacturer's instructions (Promega) using a standard luminometer.

Annealed oligonucleotides containing a 57 basepair fragment of the *Cebpa* promoter sequence with its putative MYC-binding E-box (CGCGCA) or deletion thereof, were cloned between the NheI and HindIII restriction sites of the pGL4.23 vector (Promega). *Cebpa* promoter sequences for WT and mutated E-box:
*Cebpa*_Myc_WT_s,
CTAGGTGGGCGGCGGCGACAGCGGCGCCACGCGCAGGCTGGAGGCCGC
CGAGGCTCGGCCA;
*Cebpa*_Myc_WT_as,
AGCTTGGCGAGCCTCGGCGGCCTCCAGCCTGCGCGTGGCGCCGCTGTCG
CCGCCGCCCAC;
*Cebpa*_Myc_MUT_s,
CTAGGTGGGCGGCGGCGACAGCGGCGCCAGGCTGGAGGCCGCCGAG
GCTCGGCCA
*Cebpa*_Myc_MUT_as,
AGCTTGGCGAGCCTCGGCGGCCTCCAGCCTGGCGCCGCTGTCGCCGCCG
CCCAC.

**Electrophoretic mobility shift assay (EMSA).** Q2bn fibroblast were transfected with pCMV-E2F1, pCMV-DP1, and pcDNA3-*CEBPE* expression vectors by calcium-phosphate precipitation as indicated. After 24 h, nuclear extracts were prepared according to a protocol originally described by ref. [59]. EMSAs were performed as described previously by our group[16] using a CEBP probe derived from the FAVBP4 (AP2) promoter[60], an E2F probe derived from the DHFR promoter[61], and anti-CEBPE Ab (H-75, Santa Cruz Biotechnology).

**32Dcl3 cell lines and BM cells.** 32Dcl3 cells (kindly provided by Alan D. Friedman, Dept of Oncology, Johns Hopkins University, Baltimore, MD, USA) were maintained at 37 °C and 5% CO$_2$ in Iscove´s modified medium (IMDM, InVitrogen/Thermo Fisher Scientific) supplemented with 10% heat-inactivated calf serum, 1 ng/ml murine IL-3 (StemCell Technologies, Vancouver, BC, Canada) and 100 units/ml penicillin/100 µg/ml streptomycin (InVitrogen/Thermo Fisher Scientific). Puromycin (2 µg/ml, Sigma-Aldrich, St Louis, MO, USA) was added to the medium of 32Dcl3 cell lines transduced with a pBabePuro retroviral vector.

The *CEBPE-ER*^TM cDNA construct was prepared by linking the full-length human *CEBPE* cDNA as a BamH1/AscI fragment in a frame to an AscI/EcoR1 fragment of the tamoxifen-responsive estrogen receptor hormone-binding domain (murine ER^TM; amino acids 281–599)[62]. The one mutation in the non-DNA binding side of the CEBPE basic region (BRM5: Y208A) was generated using the QuickChange system (Stratagene, La Jolla, CA) and confirmed by sequencing. The *CEBPE-ER*^TM, *CEBPE-BRM5-ER*^TM, and *ER*^TM cDNAs were inserted as BamH1/ EcoR1 fragments into the poly-linker of the pBabePuro retroviral vector (Nolan lab homepage: http://www.stanford.edu/group/nolan/index.html). The pBabePuro-*CEBPE-ER*^TM, pBabePuro- *CEBPE-BRM5-ER*^TM, and pBabePuro-*ER*^TM vectors were transfected into the ecotropic packaging cell line Phoenix-ECO (CRL-3214, ATCC, Manassas, VA, USA), by calcium-phosphate precipitation. After 24 h,

32Dcl3 cells were co-cultured with transfected Phoenix cells for another 48 h in 32Dcl3 medium plus 4 µg/ml polybrene (Sigma-Aldrich). Subsequently, 32Dcl3 cells were selected in puromycin (2 µg/ml, Sigma-Aldrich), and subclones were generated by the transfer of single cells into 96-well dishes using an automated Quickcell transfer device (Stoelting, Wood Dale, IL, USA).

Translocation of the fusion and control proteins from the cytosol to the nucleus in 32Dcl3-*CEBPE-ER*$^{TM}$ and 32Dcl3-*CEBPE-BRM5-ER*$^{TM}$ cell lines was induced by the addition of 200 nM of 4-hydroxy-tamoxifen (4-HT, Sigma-Aldrich) to the medium. 32Dcl3-*ER*$^{TM}$ cells induced by 4-HT served as control. To assess the ability of CEBPE to induce differentiation and cell cycle exit, 32Dcl3 cell lines were subjected to 4-HT induction and stained with fluorochrome-conjugated antibodies CD11b-PE (clone M1/70) or GR1-FITC (clone RB6-8C5) or propidium iodide (Sigma-Aldrich). Flow cytometry analysis was carried out on the BD LSR II flow cytometer (BD Biosciences). Subsequent analyses of flow cytometry data were conducted using FlowJo analysis software.

KIT-positive BM cells from *Cebpe* WT and KO mice were MACS-enriched using CD117 Microbeads and LD columns according to the manufacturer's recommendations (Miltenyi). KIT-enriched BM cells were maintained in in X-VIVO complete media at 37 °C, 5% CO$_2$, 95% humidity, and retrovirally transduced with pBabePuro-*CEBPE-ER*$^{TM}$, pBabePuro-*CEBPA-ER*$^{TM}$ generously provided by Gerhard Behre, Clinic for Internal Medicine I, Dessau Medical Centre, Dessau, DE) and empty pBabePuro vectors. For this, 24-well plates were RetroNectin–coated (Takara Bio Inc., Kusatsu, JPN), blocked with 2% BSA-blocked (StemCell Technologies), and spinoculated with retrovirus (2000 × g, 50 min, 32 °C, frozen supernatants produced as for 32Dcl3 transduction). After spinoculation, the retroviral supernatant was discarded, and 500.000 KIT-enriched BM cells per well were plated in X-VIVO Complete media by centrifugation (300 × g, 1 min, RT). cultured for 2 days, and subjected to puromycin selection (2 µg/ml, Sigma-Aldrich) for 1 day. Subsequently, transduced KIT-enriched cells were treated with 4-HT for 3 days to assess differentiation after 4-HT induction of CEBPA and CEBPE activity by flow cytometry analyses as described above. Induced granulocytes (iGRs) and induced monocytes (iMOs) were defined as CD115-Ly6G+ and CD115 + CD11b + cells, respectively. X-VIVO complete medium: X-VIVO 15 with gentamicin (Lonza, Walkersville, MD, USA), supplemented with 10% bovine serum albumin (BSA) (StemCell Technologies), 0.1 mM β-mercaptoethanol (Sigma-Aldrich), 1% ʟ-glutamine (Gibco/Thermo Fisher Scientific), 1% Pen/Strep (Gibco/Thermo Fisher Scientific), and the following cytokines: hIL-6 (50 ng/ml), mSCF (50 ng/ml), mIL-3 (10 ng/ml), and GM-CSF (10 ng/ml) (PeproTech, Hamburg, DE).

**CRISPR interference.** Putative gene promoters and enhancers of genes bound by CEBPE and expressed during terminal granulocytic differentiation (at their promoters/enhancers) were selected for CRIPSRi experiments, which were done essentially as described previously[63,64]. 32Dcl3-*CEBPE-ER*$^{TM}$ cells were transduced with lentiviral dCas9-KRAB-MeCP2 (Addgene) and treated with blasticidin (40 µg/mL) to select for cells stably expressing the dCas9-KRAB-MeCP2 protein. Sequences for sgRNAs targeting putative promoters and enhancers bound by CEBPE were designed utilizing the Broad Institute website: *Hlx* promoter sgRNA, GAGCTTTCGAGTCAGACCCC; *Dhrs7* enhancer sgRNA, TCAGAACTTTGA GGCCAACC; *Dhrs7* promoter, GCCAAGCTGAGCCTTGACCA; *Hp* promotor sgRNA, TGCAAACACAGAAATGGAGG. To generate sgRNA constructs, sgRNA oligos were inserted into pL-CRISPR-SFFV-Puro-P2A-EGFP (generously provided by Kristian Helin, BRIC, Univ. of Copenhagen, DK). sgRNAs expressing constructs were co-transfected with pAX8 and VSVG to HEK293FT cells using a standard calcium-phosphate protocol. Viral supernatants were collected 48 h after HEK293FT transfection and used for lentiviral transduction of 32Dcl3-*CEBPE-ER*$^{TM}$ cells with indicated sgRNAs or scramble sgRNA as control. Forty-eight hours after transduction, cells were cultured with 4-HT to induce CEBPE activity. On day 2 or 4 of 4-HT, 1 × 10$^5$ GFP-positive 32Dcl3-*CEBPE-ER*$^{TM}$ cells were sorted and subjected to real-time RT-PCR to assess gene expression levels as described previously[65]. The following primers were used: *Hp* forward primer, TTCTACAGACTACGGGCCGA, *Hp* reverse primer, CGACTGTGTTCACCCA TTGC; *Hlx* forward primer, TGTCTGCGGAATTTGACCCA, *Hlx* reverse primer, AGATGCGAAGAACTGTCCCG; *Dhrs7* forward primer, TGCAGCTCTTGC GCTTTTTG, *Dhrs7* reverse primer, CAGCTCCCATTCTGGGCGT; *B2m* forward primer, ACGTAACACAGTTCCACCCG, *B2m* reverse primer, CAGTCTCAGTG GGGGTGAAT. Relative gene expression levels for *Hp, Hlx*, and *Dhrs7* were calculated by normalization to the *B2m* housekeeping gene.

**Targeted mass spectrometry (MS) analysis.** BM populations were sorted into a 384-well Eppendorf LoBind PCR plate (Eppendorf AG, Hamburg, DE) and prepared as previously described[66]. Five hundred cells were sorted into wells containing 1 µl of lysis buffer (50 mM Triethylammonium bicarbonate (TEAB) pH 8.5, 20% 2,2,2-Trifluoroethanol (TFE), 10 mM tris(2-carboxyethyl) phosphine (TCEP), and 40 mM Chloroacetamide (CAA)). Directly after sorting, plates were briefly spun, snap-frozen on dry ice, boiled in a PCR device at 95 °C (Veriti 384-Well Thermal Cycler, Applied Biosystems, Waltham, MA, USA) for 5 min, and snap-frozen again on dry ice, and stored at −80 °C. Subsequently, 384-well plates with BM populations were thawed, and absolute quantified peptides (SpikeTides™ TQL, JPT Peptide Technologies GmbH, Berlin, DE) were dispensed into wells using the

I.DOT One instrument (Dispendix GmbH, Stuttgart, DE). CEBPA peptide 1 = VGAPALRPLVIK, peptide 2 = VLELTSDNDR, CEBPE peptide 1 = GGQQPLEFSGGR, peptide 2 = VLEYMAENER. In addition, three to four dilution series for each synthetic peptide were dispensed to confirm the linear response of peak areas and to determine the limit of quantification (LOQ) for each peptide. These dilution curves were prepared in the same way as the absolute quantification experiments and thus contained the same background matrix.

Digestion was performed by adding 10 ng of Trypsin (T6567, Sigma-Aldrich), dissolved in 1 µl of 100 mM TEAB pH 8.5 containing Benzonase (E1014, Sigma-Aldrich) diluted 1:5000 (vol/vol) to digest any DNA that would interfere with downstream processing. Plates were kept at 37 °C overnight to complete the digestion. Digestion was stopped by adding 1 µl of 2% Trifluoroacetic acid (TFA, Sigma-Aldrich) to each well. Samples were directly loaded onto conditioned EvoTips (Evosep Biosystems, Odense, DK), by transferring the content of eight wells (4000 cells) into 40 µl of buffer A (0.1% Formic acid) that was placed on top of the EvoTips. Subsequently, EvoTips were prepared according to the manufacturer's protocol. Samples on EvoTips were acquired using the 20 SPD Whisper method of the Evosep One System coupled to an Orbitrap Eclipse Tribrid Mass Spectrometer with FAIMS Pro Interface (Thermo Fisher Scientific) running Tune 3.4 and Xcalibur 4.3. FAIMS was set to −40 CV. A parallel reaction monitoring (PRM) method was run with the following parameters. Each cycle consisted of an MS1 scan with Orbitrap resolution of 60k, 400–1600 m/z scan range, 30% RF lens, 300% AGC, and 50 ms maximum injection time. Following the MS1 the eight targeted peptide ions (four endogenous, four heavy-labeled synthetics) were measured in a retention time scheduled manner via MS2 with 1.2 m/z isolation window, 30% HCD collision energy, Orbitrap resolution of 500k or 240k, and 500% AGC target, maximum injection time set to auto, 300–2000 m/z scan range and 30% RF lens.

Raw files were analyzed with Skyline 21.1.0.146 (https://analyticalsciencejournals.onlinelibrary.wiley.com/doi/10.1002/mas.21540) as described previously[67]. Peak areas and ratios to synthetic peptides were exported for subsequent analysis in Python 3.8. Three to four dilution series for each synthetic peptide in the same matrix was performed and analyzed to confirm the linear response of peak areas and to determine the limit of quantification (LOQ) for each peptide. For the absolute quantification of endogenous peptides in each sample, peak area ratios of an endogenous peptide to synthetic heavy-labeled spike-in of known amounts was used.

**RNA-seq data analyses**

*Read mapping and gene expression profiling.* Paired-end reads derived from mRNA sequencing of sorted BM populations of *Cebpe* WT mice (i.e., GP, PM, MY1, MY2, MM, BC, GR, MP, PMO1, PMO2, MO1, and MO2) and BM populations of *Cebpe* KO mice (i.e., GP$^{KO}$, PM$^{KO}$, MY1 + MY2$^{KO}$, and MM$^{KO}$) were mapped to the mouse genome assembly (mm9) using STAR (version 020201)[68]. Raw read counts mapping to Gencode-defined gene annotations (Ensembl 65) were determined using the featureCounts function of the Rsubread package (version 1.24.2)[69,70]. Complementary previously published mRNA-seq data of sorted WT LSKs, preGMs, and GMPs (GEO ID: GSE89767)[25] were included in the final RNA-seq data set, which allowed temporal gene expression profiling of 33,924 genes during the course of early and late granulocytic and monocytic differentiation. Batch effects of gene expression values were corrected using the removeBatchEffect function of the Limma package[71]. The PRCOMP function in R was used to generate the PCA plot based on the gene expression profile of all 33,924 genes expressed during granulocytic and monocytic differentiation. Genes ($N = 22,215$) having at least 1 mapped read across at least two samples, and standard deviation across samples of more than 0 were selected for subsequent analyses.

*Identification of gene clusters and differentially expressed genes.* The optimal number of gene expression clusters was determined to be 15 by applying kmeans function in R and by minimizing the 'sum of square' distance among clusters. Heatmaps showing gene expression profiles of gene clusters during differentiation were generated using the complexHeatmap tool[72]. Genes differentially expressed in *Cebpe* KO vs. *Cebpe* WT GPs, PMs, MY1s, MY2s, and MMs were identified using the DESeq2 package (Up: log2FC > 0 and adjusted p value <0.05; Down: log2FC < 0 and adjusted p value <0.05; Neutral: otherwise)[73]. Subsets of the 15 gene clusters were merged into the following five major gene clusters represented in Fig. 2 based on similarity, to empower subsequent functional bioinformatics analyses: Clusters 3, 4, and 5 were merged into cluster A(GMP)—genes expressed during early granulocytic and monocytic differentiation (i.e., in LSKs, preGMs, GMPs, GPs, PMs, and MPs); cluster 6 matched cluster B(MY)—genes transiently expressed in MY1s and MY2s; clusters 10 and 11 were merged into cluster C(GR)—genes terminally upregulated in MMs, BCs, and GRs during late granulocytic differentiation; cluster 12 matched cluster D(GR + MO)—genes terminally upregulated during both late granulocytic and monocytic differentiation; clusters 14 and 15 were merged into cluster E(MO)—genes upregulated exclusively during monocytic differentiation in PMO1s, PMO2s, MO1s, and MO2s.

Genes from clusters A(GMP), B(MY), D(GR + MO), and E(MO) that are upregulated in MY1s + MY2s and MMs of *Cebpe* KO vs. *Cebpe* WT mice were defined as differentially expressed (DE) or neutral. Similarly, genes from cluster

C(GR) that were downregulated in MY1s + MY2s and MMs in *Cebpe* KO vs. *Cebpe* WT mice were defined as differentially expressed (DE) or neutral.

*Cell cycle genes and Myc target genes*. A list of 102 cell cycle genes was generated by combining 68 key cell cycle genes reported by ref. [74] with a manually curated list of 34 genes. The MYC target gene list was previously reported by ref. [75].

*Functional characterization of genes*. Gene ontology analysis was performed using clusterProfiler[76] to identify the top ten gene ontology terms of "biological processes" (FDR < 0.05) for the five gene clusters in Fig. 2.

*CEBPA-mediated regulation of the +6KB enhancer of Cebpe*. A gene expression matrix containing the expression profiles of 32Dcl3 cells before and after CRISPR-mediated functional disruption of the +6KB *Cebpe* enhancer was kindly provided by Dr. Pavithra Shyamsunder[77]. Changes of *Cebpa* and *Cebpe* expression following the loss of the +6KB *Cebpe* enhancer activity were determined using EdgeR[78].

**Single-cell data analysis**. Integrated analysis of single-cell data from whole bone marrow (The Tabula Muris Consortium—version 3) and Kwock et al. (GSE151630) was performed using Seurat v4.0.1 (default parameters)[22,38,79]. To annotate single cells with bulk RNA-seq profiles of sorted BM populations of the current study, and studies by Kwock et al., Evrard et al., and Xie et al., SingleR (v1.8.0) was used[22–24,80]. Single cells were annotated to different phases of cell cycle (G1, S and G2/M) using Seurat v4.0.1 (default parameters).

**ChIP-seq data analysis**

*Read mapping*. Raw reads derived from CEBPA (MY1s + MY2, MMs), CEBPE, H3K27ac (MY1s + MY2s, MMs, BCs, and GRs) ChIP-seq experiments and our previously published PU.1, CEBPA, H3K4me1, H3K27ac (LSKs, preGMs, and GMPs) ChIP-seq datasets[25] were mapped to mouse (mm9) genome assembly using Bowtie2[81]. We used uniquely mapped and PCR duplicates (exact copies) collapsed as one read and extended to their fragment length by determining the read extension size using MACS2 (predicted parameter)[82]. Raw read counts were normalized to TPM using deeptools (bamCoverage)[83].

*Peak calling*. Genomic regions enriched for PU.1 and CEBPA binding in LSKs, preGMs, and GMPs, and for CEBPE in MY1s + MY2s, MMs, BCs, and GRs, were determined by peak calling using MACS2[82]. To increase the specificity of the enriched regions, we used a control sample prepared with an IgG control antibody (PU.1 and CEBPA) or *Cebpe* KO cells (CEBPE). Irreproducible Discovery Rate (IDR)[84] was used at a false discovery rate of 0.05 to filter out irreproducible regions between the two replicates. Genomic regions enriched for MYC, E2F1, and NFYB binding were inferred by using previously published ChIP-seq data derived from ESC (GEO ID: GSE11431, GSE56839)[40,85]. Since 99% of CEBPA peaks (22,323 out of 22,582) were detected in GMPs (Fig. 6a), we measured CEBPA binding based on its signal intensity in this BM population. The signal intensity of CEBPE was measured in MY1s + MY2s which exhibited 93% of all CEBPE peaks (28,171 out of 30,250) (Fig. 6b).

*Enhancer identification and target genes*. Potential enhancer regions (N = 58,562) were defined as (i) regions distal to promoters (>1000 bp from TSS), bound by one or more of the following TFs: PU.1, CEBPA, CEBPE, MYC, or E2F1, and overlapping with open chromatin regions defined in a compendium of mouse immune cells[39], or (ii) as regions previously identified as enhancer regions of myeloid BM populations[25]. Each enhancer region was set to a fixed width of 500 bp and was linked to the closest gene to define its potential cis-regulatory target.

*Ranked importance of TF in cell cycle regulation*. To rank TFs (PU.1, CEBPA, CEBPE, MYC, and E2F1) on the basis of their relative importance in regulating the expression of cell cycle genes that are differentially expressed in *Cebpe* KO vs. *Cebpe* WT mice as compared to those genes that remain neutral, we applied a generalized linear model using the train and varImp functions of the Caret package (type: classification) in R (R core team, Vienna, Austria; https://www.r-project.org/index.html). Specifically, we identified DE genes and neutral genes from clusters A(GMP) and B(MY) (Fig. 2a), and measured the binding levels of the five TFs at their promoters. We considered a promoter to be bound by PU.1, CEBPA, or CEBPE, if its normalized ChIP-seq signal (TPM) was above its median signal at the promoters of all genes that were differentially expressed in *Cebpe* KO vs. *Cebpe* WT cells, leading to a binary vector of 1 (bound) or 0 (not-bound) for each of the three TFs. Similarly, previously annotated MYC (N = 3422; GSM288356) and E2F1 (N = 20,696; GSM288349) bound regions in ESC were used to identify whether they overlapped with the promoters of genes from clusters A(GMP) and B(MY)[40].

*Genome browser plots*. To visualize the position of CEBPA, CEBPE, MYC, E2F1, NFYB, and LIN54 binding at the promoter of genes, we used Gviz to plot their ChIP-seq binding signals[86]. For comparison across different cell populations, ChIP-seq signals of TFs were plotted to the same scale. Heatmaps showing the

binding signal of CEBPE, NFYB, and E2F1 at promoters were plotted using the ngs.plot package[87].

*Distance between CEBPE and E2F1 binding analysis*. We identified gene promoters (N = 4071) that were bound by CEBPE (i.e., in MY1s + MY2s) and measured the distance of CEBPE binding to the closest E2F1 binding site. Gene promoters for which the distance between the CEBPE and E2F1 binding sites was 0 (i.e., overlapping) were considered as promoters where CEBPE potentially interacts with E2F to regulate E2F-dependent gene expression.

*Test for similarity of canonical TF binding motifs*. The Similarities of the underlying DNA sequence of gene promoters to canonical CEBPE, MYC, E2F1, NFYB, and LIN54 binding motifs were defined by Jaspar (NFYB, LIN54, and CEBPE) or Hocomoco (E2F1 and MYC) and measured using findMotifsGenome.pl script in the Homer package[88–90].

*DNA sequence motif analysis*. Sequences corresponding to promoters of genes from the five gene clusters (Fig. 2a) were analyzed for enrichment of Transcription Factor (TF) binding sites using HOMER[90]. Known TF-binding motifs were downloaded from MEME, Jaspar, Uniprobe, Hocomoco, and refs. [88,89,91–93]. Motif enrichment was computed for promoters of genes differentially expressed from all five clusters in *Cebpe* KO vs. *Cebpe* WT mice and compared to their enrichment in neutral genes (DE vs. neutral genes in Fig. 2). Motif enrichment was calculated using the cumulative binomial distribution. One hundred motifs were searched for a range of motif lengths (7–14 bp), and, after filtering for redundant motifs, the top 50 motifs resulting from each search were combined to obtain a final set of motifs. Only motifs enriched at the promoter of DE relative to neutral genes from each of the five clusters are shown in Fig. 3c.

*Definition of CENL, CE, and Rest gene subclasses in cluster B(MY)*. Known E2F1 and NFYB binding sites (ChIP-seq) in ESCs were obtained from previous studies (GEO ID: GSE11431 and GSE56839)[40,85], and were overlapped with gene promoters from cluster B(MY) (N = 1349; Fig. 2) using bedtools[94]. Next, subclasses of cluster B(MY) genes (N > 30) were defined based on combinatorial binding sites of four TFs (CEBPE = C, E2F1 = E, NFYB = N, LIN54 = L). CENL subclass: Binding of all four TFs. CE subclass: Binding of C and E with or without additional binding of either N or L. Rest: Binding of other TF combinations of C, E, N, L, or none thereof. Binding sites for each of the four TFs on gene promoters were defined based on ChIP-seq data (E2F1 and NFYB in ESCs, CEBPE in MY1s + MY2s) and binding motif analyses (NFYB and LIN54).

**Statistics**. No statistical method was used to predetermine sample size. Median levels and standard error were calculated for replicates of each BM population analyzed by RNA-seq. ChIP-seq signals were normalized to tags per million (TPM) and the mean of the two replicates was used for subsequent analyses. The Wilcoxon test was used to test for statistical significance between groups unless otherwise specified in the figure legends. All the statistical analyses for RNA-seq and ChIP-seq experiments was performed in R version 3.4.4 and exact p-values were calculated and reported in the figures.

**Reporting summary**. Further information on research design is available in the Nature Research Reporting Summary linked to this article.

## Data availability

The datasets generated in this study are available at GEO, ID GSE159430. The previously published datasets used in this study are available at GEO, ID GSE89767, GSE11431, GSE56839, GSE151630, GSE109467, GSE137538, and https://tabula-muris.ds.czbiohub.org (Tabula Muris single-cell data). Source data are provided as a Source Data file. Source data are provided with this paper.

## Code availability

Codes for the generation of manuscript figures are available at https://github.com/porseLab/Cebpe[95].

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

## Acknowledgements

This work was supported by the Novo Nordisk Foundation (B.T.P. and K.T.-M., Novo Nordisk Foundation Center for Stem Cell Biology, DanStem; Grant Number NNF17CC0027852). B.F. is the recipient of a Copenhagen Bioscience Ph.D. stipend by the Novo Nordisk Foundation, NNF19SA0035442). K.T.-M. is supported by a clinical research fellowship (grant no. 100191) and by grants from the Danish Council for Strategic Research (grant no. 133100153, K.T.-M.), the Danish Cancer Society (grant no. R72-A4572-13-S2, K.T.-M.).

## Author contributions

Conceptualization: K.T.-M., S.P., and B.T.P.; Methodology and investigation: K.T.-M., S.P., K.R., J.S., M.T., B.F., M.B.S., J.J., J.S.J., M.S.H., K.J.K, J.B.C., A.F., and E.S.; Software: S.P.; Formal analysis: S.P. and K.T.-M.; Writing: K.T.-M, S.P., and B.T.P. Visualization: S.P. and K.T.-M; Supervision: K.T.-M., and B.T.P. The authors wish it to be known that, in their opinion, the first two authors should be regarded as joint first authors. Co-first authors can prioritize their names when adding this paper's reference to their résumés.

## Competing interests

The authors declare no competing interests.
