## [Peer Review File · Nature Communications]

Transcription factor-driven coordination of cell cycle exit and lineage-specification in vivo during granulocytic differentiationREVIEWER COMMENTS

Reviewer #1 (Remarks to the Author):

Summary:

In this paper, Theilgaard-Mönch et al. investigated how the tissue-specific transcription factors, CEBPA and CEBPE, coordinate cell cycle exit and lineage-specification during granulocytic differentiation. First, they attempt to identify high-resolution populations of cells along granulocytic and monocytic trajectories with differential expression of cell surface proteins using flow cytometry, supplemented with microscopy. Thus, they claim that they have developed a method for isolating populations in a map of steady-state granulocytic and monocytic differentiation. Sorting these populations and performing bulk RNA-seq allows for identification of trajectories for both normal and Cebpe-mutant populations. From this, the authors can conclude that there are specific transcriptional programs of granulocytic and monocytic differentiation, and that Cebpe is correlated with myeloid commitment. Next, ChIP-seq, along with motif analysis of genomic regions of interest is used to profile the interactions of Cebpa and Cebpe in the populations of interest. They claim that Specification of cell identity is regulated at the enhancer level and cell cycle exit at the promoter level. Promoters of G1/S phase genes are found to be associated with Cebpe inhibition of Myc with induction of Cdk inhibitors. G2/M phase genes were found to be regulated by Cebpe repression of E2F target genes. The ChIP-seq data of both Cebpa and Cebpe was used to show that sequential binding of these factors occurs at target genes involved in the specification of granulocytic cells. Lastly, the authors claim that Cebpa may be regulated by Myc/Max binding near its promoter. The major strength of this paper is the combinatorial binding of Cebpe, E2f1, and Nfyb paired with the control of cell cycle gene regulatory networks. However, there are a lot of weaknesses that detract from initial enthusiasm. First, the authors fail to reference and integrate prior studies concerning the identification of cell states in development of granulocytes/monocytes. Second, there is no effort to validate that the populations being studied are homogenous, which undermines the claim that the authors have successfully captured developmental timepoints, or that Cebpa and Cebpe expression is mutually exclusive. Third, there is little functional validation for the regulatory interactions identified. Thus, each are associative and not proven to be causal at this point. Please refer to the lists of major and minor weaknesses below:

Major weaknesses:

- Author's don't connect their findings to previously published populations of neutrophil and monocytic progenitors
- What are the overlapping populations relative to previous studies?
- Monocytic and granulocytic progenitor populations have been described in detail with scRNA-seq experiments, combined with cell surface protein experiments to identify subpopulations in these trajectories. The authors make no attempt to reference these prior studies or integrate the labels of previously identified populations to their new scheme.
- It would be fairly straightforward to integrate the labels provided for new populations by doing a correlation of RNA-seq vectors between the pseudobulks and bulk RNA-seq of the populations the authors identify. Of course, if the populations studied actually correspond to more than one cluster from single cell studies then this would fail.
- Expression of cell cycle genes at different stages of neutrophil progenitor populations has previously been explored with scRNA-seq and should be explored/referenced
- Please consider:

-This manuscript addressed that Cebpe and E2f2 contribute to neutrophil commitment (Xie, X., Shi, Q., Wu, P. et al. Single-cell transcriptome profiling reveals neutrophil heterogeneity in homeostasis and infection. *Nat Immunol* 21, 1119–1133 (2020)).

-This manuscript addressed Cebpa in GM lineage (Oh P, Lobry C, Gao J, Tikhonova A, Loizou E, Manet J, van Handel B, Ibrahim S, Greve J, Mikkola H, Artavanis-Tsakonas S, Aifantis I. In vivo mapping of notch pathway activity in normal and stress hematopoiesis. *Cell Stem Cell*. 2013 Aug 1)

-This manuscript addressed bipotent progenitors; Cebpe and Gfi1 driving granulocyte specification;

Cebpa in co-regulation

(Olsson A, Venkatasubramanian M, Chaudhri VK, Aronow BJ, Salomonis N, Singh H, Grimes HL. Single-cell analysis of mixed-lineage states leading to a binary cell fate choice. *Nature*. 2016 Sep 29)

-This manuscript addressed Myc/Mxd1, Irf8/Gfi1, and Cebpa/Cebpe in granulocyte subclusters and lineage decisions

(Muench, D.E., Olsson, A., Ferchen, K. et al. Mouse models of neutropenia reveal progenitor-stage-specific defects. *Nature* 582, 109–114 (2020).)

-This manuscript addressed Cebpa regulating entry into all myeloid fates

(Giladi, A., Paul, F., Herzog, Y. et al. Single-cell characterization of haematopoietic progenitors and their trajectories in homeostasis and perturbed haematopoiesis. *Nat Cell Biol* 20, 836–846 (2018).)

-This manuscript attempts to devise a flow scheme to separate populations of cells undergoing neutrophil granulocyte differentiation.

(Kwok, Immanuel, et al. "Combinatorial single-cell analyses of granulocyte-monocyte progenitor heterogeneity reveals an early uni-potent neutrophil progenitor." *Immunity* 53.2 (2020): 303-318.)

Also, please consider:

Kim, Min-Hyeok, et al. "A late-lineage murine neutrophil precursor population exhibits dynamic changes during demand-adapted granulopoiesis." *Scientific reports* 7.1 (2017): 1-15.

Zhu, Yanfang Peipei, et al. "Identification of an early unipotent neutrophil progenitor with pro-tumoral activity in mouse and human bone marrow." *Cell reports* 24.9 (2018): 2329-2341.

Evrard, Maximilien, et al. "Developmental analysis of bone marrow neutrophils reveals populations specialized in expansion, trafficking, and effector functions." *Immunity* 48.2 (2018): 364-379.

- It is not clear whether there is heterogeneity within the identified populations, for which single cell assays are required to validate...

- Again, need to respond to work that was previously published in this area

- The separation of populations is based on only 8 markers (Lin, Kit, Sca, CD115, CD16/32, CD34, Ly6G, CD11b)

- There is nowhere near enough data to justify that these populations are transcriptionally distinct and pure relative to single-cell heterogeneity

- Other groups are using assays that capture signals from >20 markers (CYTOF and CITE-seq). Can the authors comment on the purpose of using such tools if 8 markers is sufficient to isolate these populations of interest?

- The authors claim that unipotent granulocytic (GP) and monocytic (MP) progenitor populations exist, yet they don't validate these predictions with colony forming assays or lineage tracing experiments, or reference previous literature to validate that these populations are unipotent progenitors for the specified trajectories.

- The authors claim that Cebpa and Cebpe show mutually exclusive expression profiles, but this is clearly incorrect. Figure 2B is certainly insufficient to make this claim. First of all, these are bulk RNA-seq samples, so it is not clear if single cells simultaneously express Cebpa and Cebpe. Previously published scRNA-seq data from multiple papers shows that there are many cells that co-express these genes.

- The effect of CEBPE from enhancers and promoters at this point is only an association of the binding of CEBPE to the change in gene expression at different stages. No causal role for CEBPE can be made without functional validation of genomic loci on target gene expression.

- For example, the authors could mutate CEBPE binding sites in promoters and enhancers and test

for changes in target gene expression

- Alternatively, CRISPRi assays could be performed to silence the chromatin regions for functional validation

Minor weaknesses:

- Gene clusters identified in 2A are overlapping in areas

- Eg. what is the difference between: LSK vs preGM? GP vs PM? MY1 vs MY2? MO1 vs MO2?

- If these are unique populations, they should certainly have unique gene expression patterns...

- The authors state:

"Overall, these findings demonstrate that Cebpa expression correlates with myeloid commitment of HSCs (i.e. LSKs) toward lineage-restricted progenitors (i.e. GPs/PMs) and expression of early granulocytic differentiation genes (i.e. primary granule proteins)."

- However, the word commitment should be replaced with specification (moving from a multipotent state towards a single lineage, but not fully committed). Commitment should be a word reserved for not being able to produce other lineages. However, the authors do not show that this cluster is a commitment stage, and previous literature would suggest that it is myeloid specification.

- The authors proposed enhancers for genes lack a clear genomics + informatics approach to link enhancers to those genes. The assessment of genomic loci as enhancers >1,000 bp from the transcription start site is ambiguous

- Would the authors consider a loci 1,000,000 bp from a TSS to be an enhancer of the gene?

- Promoter capture Hi-C or co-accessibility of genomic regions across a single-cell experiment would provide better definition of enhancer-gene connections (again, if the authors had read prior literature, they would see that some of these datasets already exist)

- Information about the 3D chromatin shape is lacking. Is it the case that there are interactions provoked by CEBPE bound enhancer that help to explain the changes in gene expression?

- Given the large number of motifs in the databases used ("Known TF-binding motifs were downloaded from MEME, Jaspar, Uniprobe, Hocomoco and Jolma et al."), the authors should include the motif scores for all of the tested motifs, at least in the supplemental information.

- Ranked importance of TFs seems heavily biased towards the available ChIP-seq data. The authors should comment on efforts to control for the importance of other TFs.

- There is a suggested direct interaction of CEBPE with E2Fs at G2/M promoters on the basis of absence of CEBPE motifs at promoter regions

- Can co-immunoprecipitation validate the complex of CEBPE and E2F in the indicated populations?

- Can 3D chromatin conformation confirm chromatin looping of these enhancer-promoter interactions? Alternatively, can the authors comment on how the regulation of this gene via this enhancer is occurring?

- Can the authors comment on the structural differences of CEBPA and CEBPE that may confer differential cell cycle activity (interaction with E2F complex)?

- The authors should supply a supplementary file with each of the CEBPE motif positions that they find interesting in describing the target genes of interest, such that they can provide genomic coordinates for the regulatory activity of CEBPE.

- Is there a reason the authors don't plot ChIP-seq profiles for CEBPA and CEBPE simultaneously for the same populations (Figure 5E/F)?

- Do CEBPA and CEBPE ever bind to the same location in the same cell population?

- I would really like to see the dynamics of CEBPA and CEBPE binding across all of the populations

- Can the authors functionally validate the MYC/MAX activating sites of the Cebpa promoter?
- Would mutation at these MYC/MAX sites inhibit Cebpa promoter activity? (This could easily be answered with a Cebpa reporter, cloning in the enhancer upstream)

- Why are CEBPA and CEBPE written in all caps sometimes? Aren't all of the data generated in mice, and should take on the convention of Cebpa and Cebpb with the first letter alone capitalized?

- Shouldn't the title use identity instead of identify
- Eg. Transcription factor-driven coordination of cell cycle exit and specification of cell identity during granulocytic differentiation

Reviewer #2 (Remarks to the Author):

General comments

In this study, Theilgaard-Monch, Pundhir, and Porse et al addressed a fundamental question about the coordination of cell cycle exit and cell type specification by analyzing the role of CEBPA and CEBPE transcription factors in granulocytic differentiation *in vivo*. By detailed transcriptome analyses and some ChIP-seq analyses, the authors demonstrate that CEBPA directly induces Cebpe gene expression, and CEBPE indirectly binds to promoter-proximal regions through chromatin-bound E2F to inhibit E2F-mediated expression of cell cycle genes including Myc. On the other hand, CEBPA and CEBPE directly bind to promoter-distal enhancer regions to induce early and late granulocytic genes, respectively. The authors also suggest the possible mechanism by which the gene expression of Cebpa is replaced by that of Cebpe through the Myc-Mad-Max system. Overall, the data nicely describe the mechanism of how CEBPA-induced CEBPE causes both cell cycle exit and terminal differentiation towards granulocytes, which significantly contributes to the progress in the research field. On the other hand, some of the conclusions, especially the concept that CEBPA transiently primes, while CEBPE activates or unleashes, the terminal granulocytic differentiation program at the promoter and enhancer level, are not fully supported by the data and require additional experiments.

Specific comments

Major points

1. Summary: One of the significant points of this study is that the authors dealt with cells freshly isolated from mice to address the "in vivo" mechanism. This would be better to be stated in the Summary.

2. The order of Fig numbers does not match between the text and Figures, making it difficult to read the paper.

3. GMPs are analyzed in Fig. 1D and the authors state in lines 181-182 and 194 that the Figs 2A and B include GMPs. However, Figs 2A, 2B and S6A do not display the data of GMPs. This is inconsistent and confusing especially because the cluster A is named "GMP". Showing the GMP data should be important to understand the switch from CEBPA to CEBPE.

4. Figs. 2B and S6A: Please show the expression of CEBP proteins as well. It would be even more informative if the authors could estimate their molecular numbers per cell to address the mechanism by which CEBPA and CEBPE play distinct roles (see Comment 7c).

5. The authors state in lines 221-223 "Collectively, these findings suggest that CEBPE-dependent cell cycle exit is a prerequisite to complete terminal granulocytic differentiation". Here, the fundamental question would be whether cell cycle arrest is required for the induction of granulocytic genes, which is not addressed or discussed. If available, CEBPE mutants that cannot bind to E2F but can bind to DNA and activate transcription could be transduced into CEBPE KO

cells.

6. Fig. S6D: This Figure is very important, so should be shown as a main Figure. It would be informative to show the overlap between CEBPA and CEBPE binding sites by Venn diagrams.

7. The conclusion about the sequential actions of CEBPA and CEBPE, stated in Summary (“CEBPA promotes lineage commitment by launching an enhancer-primed differentiation program” and “CEBPE unleashes the CEBPA-primed differentiation program”) and elsewhere, is important and quite interesting but requires further validation and discussion.

--a. Does CEBPA prime but not activate the enhancers targeted by both CEBPA and CEBPE? This can be tested by ChIP-seq of CEBPA KO cells for H3K4me1 and H3K27ac, and the authors might already have such data in the previous study by Pundhir et al, Cell Rep 2018.

--b. Is CEBPE required for activating the enhancers that had been primed by CEBPA? This can be tested by ChIP-seq of CEBPE KO cells for H3K27ac.

--c. I feel that it will be natural for readers to wonder the mechanism by which CEBPA and CEBPE belonging to the same family can play cooperative but distinct roles. Is it due to the timing of their expression, molecular copy numbers (see Comment 4), or their distinct intrinsic activities?

Although I understand that a single paper cannot solve everything, this highly relevant issue should be addressed or at least fully discussed. Can ectopic expression of CEBPE and CEBPA rescue granulocytic differentiation of CEBPA KO and CEBPE KO cells, respectively? When CEBPE is lost, CEBPA expression is significantly upregulated, but cannot compensate for the loss of CEBPE to control gene expression; does CEBPA bind to the CEBPE target sites in these cells without activating enhancers and promoters (can be tested by ChIP-seq of CEBPA in CEBPE KO and WT cells)? These additional experiments might help addressing the issue.

Minor points

8. Fig. 2C: The print “Gene expr..” is not fully shown.

9. Fig. 3A: The print of GO terms is not fully shown (the right portions are missing).

10. According to the Materials and Methods section, the authors retrieved ChIP-seq data of histone modifications from previous publications to define the enhancer regions. This is better to be stated in the main text.

Reviewer #3 (Remarks to the Author):

In this manuscript the authors develop a sorting strategy to isolate cells at various stages of neutrophil and monocyte differentiation and then characterize the gene expression profiles and chromatin occupancy profiles for CEBPE. They use this newly generated data and compare it to previously generated data for CEBPA binding to propose the mechanism of coordinate regulation of gene expression by these two important transcription factors. Based on these data they also propose that CEBPA and CEBPE are inducing different cellular phenotypes at different stages of differentiation. Overall, the newly described approach to isolating cells at various stages of myelomonocytic differentiation is of interest as are the gene expression and ChIP-seq data generated from these cells. The major concern is that the overwhelming majority of the data is correlative and while fairly compelling it seems more functional characterization would help significantly.

Specifically, it would be very helpful to better characterize the 32D cell system they show in supplemental data. They demonstrate that expression of CEBPE in these cells induces differentiation and cell cycle exit, but it wasn't clear to me if they have shown that CEBPE is binding to chromatin in these cells in a way that is consistent with the hypothesis generated from primary cells. More clarity (and experiments) here would be very helpful.

Also, there is an implication that CEBPE and CEBPA are expressed at specific stages of myelod

development to drive specific phenotypes. Does this mean that CEBPA and CEBPE induce different gene expression programs at different stages of development? Is there some way that the authors could demonstrate this functional difference in the 32D cell system? More insight in regard to how they two transcription factors differ functionally would also add much needed clarity to the **field**.

1 **Point-to-point response to the referees' comments:**

2

3 **Table listing changes in the figures**

Revised manuscript	Changes compared to the original manuscript
Figure 1	No changes
Figure 2	Panel A and B: added gene expression values for GMP
Figure 3	No changes
Figure 4	Panel C: now also showing individual data points; Panel G and H: added CEBPA tracks for MY1+2 and MM (CEBPE WT and KO populations)
Figure 5	Panel A: added gene expression values for GMP and now also showing individual data points; Panel E and F: added CEBPA tracks for MY1+2 and MM (CEBPE WT and KO) populations)
Figure 6	New figure. Panel A, B and D: previous Supplementary Figure S6. New panel C, E, and F
Figure 7	No changes (Previous Figure 6)
Supplementary Figure 1	Panel B, C and E: now also showing individual data points
Supplementary Figure 2	Panel A: added gene expression values for GMP
Supplementary Figure 3	New figure
Supplementary Figure 4	Added panel B, previous Supplementary Figure 3); Panel A, C and E: now also showing individual data points
Supplementary Figure 5	Panel A: added gene expression values for GMP; Panel A and G: now also showing individual data points; previously Supplementary Figure 4)
Supplementary Figure 6	No changes , previous Supplementary Figure 5)
Supplementary Figure 7	Panel C, D and E: new; Panel F and G: added CEBPA tracks for MY1+2 and MM (WT and CEBPE KO) populations, previous Supplementary Figure 6)

**Reviewer #1 comments:**

**Summary:**

In this paper, Theilgaard-Mönch et al. investigated how the tissue-specific transcription factors,
CEBPA and CEBPE, coordinate cell cycle exit and lineage-specification during granulocytic
differentiation. First, they attempt to identify high-resolution populations of cells along
granulocytic and monocytic trajectories with differential expression of cell surface proteins
using flow cytometry, supplemented with microscopy. Thus, they claim that they have
developed a method for isolating populations in a map of steady-state granulocytic and
monocytic differentiation. Sorting these populations and performing bulk RNA-seq allows for
identification of trajectories for both normal and Cebpe-mutant populations. From this, the
authors can conclude that there are specific transcriptional programs of granulocytic and
monocytic differentiation, and that Cebpe is correlated with myeloid commitment. Next, ChIP-
seq, along with motif analysis of genomic regions of interest is used to profile the interactions
of Cebpa and Cebpe in the populations of interest. They claim that Specification of cell identity
is regulated at the enhancer level and cell cycle exit at the promoter level. Promoters of G1/S
phase genes are found to be associated with Cebpe inhibition of Myc with induction of Cdk
inhibitors. G2/M phase genes were found to be regulated by Cebpe repression of E2F target
genes. The ChIP-seq data of both Cebpa and Cebpe was used to show that sequential binding
of these factors occurs at target genes involved in the specification of granulocytic cells. Lastly,
the authors claim that Cebpa may be regulated by Myc/Max binding near its promoter. The
major strength of this paper is the combinatorial binding of Cebpe, E2f1, and Nfyb paired with
the control of cell cycle gene regulatory networks. However, there are a lot of weaknesses that
detract from initial enthusiasm First, the authors fail to reference and integrate prior studies
concerning the identification of cell states in development of granulocytes/monocytes. Second,
there is no effort to validate that the populations being studied are homogenous, which
undermines the claim that the authors have successfully captured developmental timepoints,
or that Cebpa and Cebpe expression is mutually exclusive. Third, there is little functional
validation for the regulatory interactions identified. Thus, each are associative and not proven
to be causal at this point. Please refer to the lists of major and minor weaknesses below:

**Major weaknesses:**

**Referee #1 - comment #1:**

- - Author's don't connect their findings to previously published populations of neutrophil and
- monocytic progenitors
- - What are the overlapping populations relative to previous studies?
- - Monocytic and granulocytic progenitor populations have been described in detail with scRNA-
- seq experiments, combined with cell surface protein experiments to identify subpopulations in

these trajectories. The authors make no attempt to reference these prior studies or integrate
the labels of previously identified populations to their new scheme.

**Response referee #1 - comment #1:**

We are very thankful for this important point made by the referee and have now referenced
and discussed two recent comprehensive protocols for sorting of murine BM populations
representing successive stages of granulocytic differentiation (Kwok et al., Immunity, 2020;
Evrad et al., Immunity, 2018). In addition, we have referenced a seminal study applying
scRNA-seq to identify the transcriptional landscape of granulocytic differentiation including
distinct molecular signatures for BM populations representing successive developmental
stages (Xie et al., Nature Immunology, 2020).

While both sorting protocols and ours have some similarities they also differ with respect to the
immunophenotypes of BM populations due the application of several different markers as well
as gating strategies for shared markers. Moreover, the sorting protocols differ with respect to
the total numbers of sorted developmental stages. Notably, our protocol included sorting of
LSKs and preGMPs as well as a hierarchy (i.e. from GMP to mature Neu/GR) of 8
developmental stages as compared 4 and 6 developmental stages sorted by the two published
protocols (Kwok et al., Immunity, 2020: 6 stages GMP to mature Neu, Evrad et al., Immunity,
2018: 4 stages GMPs to mature Neu). Consistently, the scRNA-seq study also reported a
lower number of “molecular signature” based BM populations representing successive
developmental stages as compared to our sorting protocol (i.e. 5 stages GMPs to mature Neu
referred to as G0-G4 in the BM, Xie et al., Nature Immunology, 2020).

Hence, our protocol likely provides a slightly higher resolution, particularly with respect to
resolution of BM population representing successive stages of cell cycle exit during neutrophil
differentiation (i.e. 5 stages GP>PM>MY1>M2>MM), which are not discriminated at the same
high resolution by the two published sorting protocols and the scRNA-seq (Evrard et al.,
Immunity, 2018: 2 stages GMP>pre-NEU; Kwok et al., Immunity, 2020: 3 stages pro-
Neu1>pro-Neu2>pre-Neu; Xie et al., Nature Immunology, 2020: 3 stags G0>G1>G2).

We have very thoroughly considered and validated the proposal by the referee to “integrate
the labels of previously identified populations to their new scheme” and compare our sorting
protocol with that of Kwok et al. (Immunity, 2020) and Evrad et al., (Immunity, 2018).
However, we would like to emphasize that the 3 sorting protocols should be compared with
caution as they all utilize different marker combinations and gating strategies and generate
different numbers of developmental stages, which obviously does not allow a direct
comparison of BM populations. We therefore like to argue that a direct comparison would be a
merely speculative estimate without any additional comprehensive validation, which would not
add to the impact and be beyond the scope of this study.

In the revised MS we have now referenced these 3 seminal studies in the introduction and in
more detail in the discussion (ref. 22-24, line 92, line 443-455).

Referee #1 - comment #2:

- It would be fairly straightforward to integrate the labels provided for new populations by doing
a correlation of RNA-seq vectors between the pseudobulks and bulk RNA-seq of the
populations the authors identify. Of course, if the populations studied actually correspond to
more than one cluster from single cell studies then this would fail.

- Expression of cell cycle genes at different stages of neutrophil progenitor populations has
previously been explored with scRNA-seq and should be explored/referenced

Response Referee #1 comment #2:

We thank the referee for raising this important point, and for the highly valuable referral to
publicly available single cell RNA-seq data (Kwock et al., 2020, Immunity, The Tabula Muris
consortium - <https://tabula-muris.ds.czbiohub.org>, scRNA data from bulk bone marrow).

In the revised MS we analyzed these single cell RNA-seq data to validate the heterogeneity of
our sorted cell populations. Similar to the analysis performed by Kwock et al. (Figure 1G), we
integrated the 10X scRNA bone marrow data (The Tabula Muris consortium, scRNA data from
bulk bone marrow) with SmartSeq2 scRNA bone marrow data by Kwock et al. Next, we
annotated each of the single cells using gene expression profiles of each of our sorted BM
populations to define the potential heterogeneity. As shown in **Supplementary Figure S3A** our
population-based annotation approach demonstrated that single cells annotated to a specific
sorted BM population essentially clustered together and maintained the same hierarchical
order of monocytic and granulocytic differentiation trajectories observed in the PCA plot
mapping the sorted BM population based on similarity of gene expression profiles (**Figure 1D**).
These novel analyses strongly suggest that our sorted BM populations are highly
homogeneous populations representing successive stages of granulocytic and monocytic
differentiation.

As suggested by the referee we have also annotated each of the single cells using G1, S,
G2M gene expression scores from Seurat v4.0.1 along monocytic and granulocytic
differentiation trajectories (**Supplementary Figure S3A and S3F**). Notably, all proliferating BM
populations contained cells in either G1, S, or G2M phase (i.e. GP, PM, MY1; PM, PMO1)
whereas quiescent BM populations are exclusively in G1 (MM, BC, GR; MO1, MO2).
Strikingly, our protocol also allowed sorting of BM populations exclusively in G2M (MY2 and
PMO2), which allowed us to discriminate between genes regulated during early G1/S exit (i.e.

DEGs at the PM to MY1 transition) and late G2/M exit (i.e. DEGs at the MY2 to MM transition)
during granulocytic differentiation (Supplementary Figure S3A and S3F).

We have reported and discussed these novel complementary data in the revised MS (line
175-183, 444-455).

- Please consider:

Response: We have referenced and discussed the highlighted publications in the revised MS.

-This manuscript addressed that Cebpe and E2f2 contribute to neutrophil commitment
(Xie, X., Shi, Q., Wu, P. et al. Single-cell transcriptome profiling reveals neutrophil
heterogeneity in homeostasis and infection. Nat Immunol 21, 1119–1133 (2020)).

-This manuscript addressed Cebpa in GM lineage
(Oh P, Lobry C, Gao J, Tikhonova A, Loizou E, Manet J, van Handel B, Ibrahim S, Greve
142 J, Mikkola H, Artavanis-Tsakonas S, Aifantis I. In vivo mapping of notch pathway activity
in normal and stress hematopoiesis. Cell Stem Cell. 2013 Aug 1)

-This manuscript addressed bipotent progenitors; Cebpe and Gfi1 driving granulocyte
specification; Cebpa in co-regulation
(Olsson A, Venkatasubramanian M, Chaudhri VK, Aronow BJ, Salomonis N, Singh H,
Grimes HL. Single-cell analysis of mixed-lineage states leading to a binary cell fate
choice. Nature. 2016 Sep 29)

-This manuscript addressed Myc/Mxd1, Irf8/Gfi1, and Cebpa/Cebpe in granulocyte
subclusters and lineage decisions
(Muench, D.E., Olsson, A., Ferchen, K. et al. Mouse models of neutropenia reveal
progenitor-stage-specific defects. Nature 582, 109–114 (2020).)

-This manuscript addressed Cebpa regulating entry into all myeloid fates
(Giladi, A., Paul, F., Herzog, Y. et al. Single-cell characterization of haematopoietic
progenitors and their trajectories in homeostasis and perturbed haematopoiesis. Nat Cell
Biol 20, 836–846 (2018).)

-This manuscript attempts to devise a flow scheme to separate populations of cells
undergoing neutrophil granulocyte differentiation.
(Kwok, Immanuel, et al. "Combinatorial single-cell analyses of granulocyte-monocyte
progenitor heterogeneity reveals an early uni-potent neutrophil progenitor." Immunity
53.2 (2020): 303-318.)

Also, please consider:

Kim, Min-Hyeok, et al. "A late-lineage murine neutrophil precursor population exhibits dynamic changes during demand-adapted granulopoiesis." *Scientific reports* 7.1 (2017): 1-15.

Zhu, Yanfang Peipei, et al. "Identification of an early unipotent neutrophil progenitor with pro-tumoral activity in mouse and human bone marrow." *Cell reports* 24.9 (2018): 2329-2341.

Evrard, Maximilien, et al. "Developmental analysis of bone marrow neutrophils reveals populations specialized in expansion, trafficking, and effector functions." *Immunity* 48.2 (2018): 364-379.

Referee #1 - comment #3:

- It is not clear whether there is heterogeneity within the identified populations, for which single cell assays are required to validate.
- There is nowhere near enough data to justify that these populations are transcriptionally distinct and pure relative to single-cell heterogeneity.

Response Referee #1 - comment #3:

We thank the referee for raising the important issue of population heterogeneity. As stated in detail in our response to the referee's comment #2, our novel complementary integrative analyses of publicly available scRNA-seq data and our BM population RNA-seq data demonstrate that our sorted BM populations are (i) transcriptionally distinct based their very low single cell heterogeneity, and (ii) represent successive stages of granulocytic and monocytic differentiation (Supplementary Figure S3A, Figure 1D).

Referee #1 - comment #4:

- The separation of populations is based on only 8 markers (Lin, Kit, Sca, CD115, CD16/32, CD34, Ly6G, CD11b)
- Again, need to respond to work that was previously published in this area
- Other groups are using assays that capture signals from >20 markers (CYTOF and CITE-seq). Can the authors comment on the purpose of using such tools if 8 markers is sufficient to isolate these populations of interest?

Response Referee #1 - comment #4:

As stated correctly by the referee more markers than those 8 selected as well as the 2 additional parameters (i.e. SSC and FSC) applied in our sorting protocol could further improve resolution of sorted BM populations.

We like to emphasize that we validated marker combinations and gating strategies based on cellular morphology (by an experienced hematologist – KTM) and DNA cell cycle profiles (i.e. DNA stain) of BM populations in order to develop a highly reproducible and robust sorting protocol allowing to sort sufficient quantities of live BM populations for complementary downstream applications such as ChIP-seq, clonogenic/functional assays, and quantitative MS protein analyses, etc. We therefore like to argue that our simple sorting protocol based on 8+2 markers provides a valuable tool to researchers that do not have access to methodologies such as scChIP-seq, scProteomics/ scPhospho-proteomics, CYTOF/CITE-seq, or even more commonly available scRNA-seq facilities. Given the distinct changes of G1/S and G2/M cell cycle gene expression as BM cells exit cycle while they differentiate via GP>PM>MY1>M2>MM, our novel sorting protocol also provides a unique tool for functional studies of cell cycle exit in the context of differentiation *in vivo* at a relative high resolution.

As pointed out by the referee, we have now cited two recent comprehensive protocols for sorting of murine BM populations representing successive stages of granulocytic differentiation (ref 22 & 23). While both protocols and ours have some similarities they also differ with respect to their distinct immunophenotypes and total numbers of sorted developmental stages (Kwok et al., Immunity, 2020: 6 stages GMP to mature Neu; Evrad et al., Immunity, 2018: 4 stages GMPs to mature Neu). We like to point out that our protocol allows for sorting of a total of 8 developmental stages (i.e. GMP to mature Neu/GR) as compared to the total of 4 and 6 developmental stages sorted by the two published protocols. Hence, our protocol likely provides a slightly higher resolution, particularly with respect to resolution of BM population representing successive stages of cell cycle exit during neutrophil differentiation (i.e. 5 stages GP>PM>MY1>M2>MM), which are not discriminated at the same high resolution by the two published sorting protocols (Evrad et al., Immunity, 2018: 2 stages pre-NEU>immature NEU, Kwok et al., Immunity, 2020: 3 stages pro-Neu1>pro-Neu2>pre-Neu). However, we would also like to emphasize that the 3 protocols should be compared with caution as they all use slightly different marker combinations and gating strategies, which does not allow a direct comparison of developmental stages for the 3 protocols, as they have different immunophenotypes.

We did not apply CYTOF/CITE-seq for our study as we aimed to develop a robust protocol for purification live BM populations, applicable for multiple downstream methodologies in parallel.

Referee #1 - comment #5:

- The authors claim that unipotent granulocytic (GP) and monocytic (MP) progenitor
populations exist, yet they don't validate these predictions with colony forming assays or
lineage tracing experiments, or reference previous literature to validate that these populations
are unipotent progenitors for the specified trajectories.

**Response referee #1 - comment #5:**

**With respect to the clonogenic potential of GPs and MPs we refer to our previous publication**
**by Pundhir et al. (2018, Cell Reports, Suppl. Figure S1D), which highlights that**
**immnuophenotypic identical GPs/CFU-Gs (LIN-, SCA1-, KIT+,FCgRII/III+, MCSFR-) and**
**MPs/CFU-Ms (LIN-, SCA1-, KIT+,FCgRII/III+, MCSFR-) were highly enriched in progenitors**
**with lineage-restricted granulocytic (approx. 80%) or monocytic (approx. 80%) differentiation**
**potential, respectively.**

**Referee #1 - comment #6:**

- The authors claim that *Cebpa* and *Cebpe* show mutually exclusive expression profiles, but
this is clearly incorrect. Figure 2B is certainly insufficient to make this claim. First of all, these
are bulk RNA-seq samples, so it is not clear if single cells simultaneously express *Cebpa* and
*Cebpe*. Previously published scRNA-seq data from multiple papers shows that there are many
cells that co-express these genes.

**Response referee #1 - comment 6:**

**We are very thankful for this very important comment by the referee! In the revised MS we**
**have now added complementary quantitative MS analysis of CEBPA and CEBPE protein**
**expression for relevant sorted BM populations to validate our RNA-seq data at the protein level**
**(Supplementary Figure S3D-E). In addition, we have mapped *Cebpa* and *Cebpe* expression in**
**the newly analyzed scRNA-seq hierarchy in Supplementary Figure S3B-C. Notably these**
**complementary data demonstrate NO mutually exclusive expression** but rather an almost
**exclusive sequential expression of CEBPA and CEBPE during granulocytic differentiation**
**including a co-expression in the PM population. The latter is consistent with our novel**
**complementary CEBPA ChIP-seq data on populations with very low CEBPA protein/RNA**
**(MY1+2, MM) and very high CEBPE protein/RNA expression (Figure 6D).**

**Based on our MS protein analyses and the expression trajectories of *Cebpa* and *Cebpe***

**mapped onto the novel scRNA-seq hierarchy (Supplementary Figure S3B-C), we now**

**demonstrated that the PM population exhibit co-expression of CEBPA/*Cebpa* and CEBPE/**

***Cebpe* both at the protein and RNA levels (incl. scRNA level). Importantly, both RNA and**

**protein profiles also highlight that upregulation of CEBPE/*Cebpe* is correlated with a marked**

downregulation of CEBPA/*Cebpa* during a very narrow developmental window when cells exit
cell cycle and initiate terminal granulocytic differentiation.

In the revised manuscript we have therefore changed our statement “mutually exclusive
expression” to “almost exclusive sequential expression” accordingly in the text (line 196-201).

Referee #1 - comment #7:

- The effect of CEBPE from enhancers and promoters at this point is only an association of the
binding of CEBPE to the change in gene expression at different stages. No causal role for
CEBPE can be made without functional validation of genomic loci on target gene expression.

- For example, the authors could mutate CEBPE binding sites in promoters and enhancers and
test for changes in target gene expression

- Alternatively, CRISPRi assays could be performed to silence the chromatin regions for
functional validation

Response referee #1 - comment #7:

We have now validated the role of CEBPE binding at promoters and enhancers using a
granulocytic differentiation model of 32D cells expressing 4-hydroxy tamoxifen (4-HT) inducible
*CEBPE-WT-ER*. We successfully identified 3 genes (*Hp*, *Dhrs7*, *Hlx*), that were upregulated
after 4-HT induction of CEBPE activity during differentiation, and whose expression was
abrogated by CRISPRi targeting of their CEBP promotor site (Supplementary Figure S7C and
S7E). In addition, we identified 1 gene (*Dhrs7*), that was upregulated after 4-HT induction
during differentiation, and whose expression was significantly reduced by CRISPRi targeting of
its putative CEBP enhancer binding site Supplementary Figure S7C and S7E. Based on our *in*
*vivo* differentiation data we also tested additional putative CEBPE regulated enhancers of *in*
*vivo* granulocytic differentiation genes, which however, were not upregulated during 4-HT
induced granulocytic differentiation of our 32D *CEBPE-WT-ER*TM cell line. These novel
CRISPRi data demonstrate that CEBPE binding at promoters and enhancers of genes
contributes to their upregulation during granulocytic differentiation.

We have reported and discussed these novel complementary CRISPRi promotor/enhancer
experiments in detail in line 402-412.

Minor weaknesses:

Referee #1 – minor comment #1:

- - Gene clusters identified in 2A are overlapping in areas
- Eg. what is the difference between: LSK vs preGM? GP vs PM? MY1 vs MY2? MO1 vs
MO2?
- If these are unique populations, they should certainly have unique gene expression
patterns...

Response referee #1 – minor comment #1:

We apologize for this misunderstanding. The gradual change of the unique gene expression
pattern of all populations is rather reflected in the detailed unsupervised hierarchical cluster
analyses demonstrating 15 unique minor clusters, in the **Supplementary Figure S2**, which
demonstrates the gradual changes of gene expression in individual populations in more detail
than the 5 merged major clusters represented in Figure 2A. We like to emphasize that the
populations such LSK and preGMP, GP and PM, MY1 and MY2, or MO1 and MO2 represent
rather close developmental stages, which however still exhibit distinct differential gene
expression among the 15 minor clusters (**Supplementary Figure S2**). Unfortunately, these
minor differences in gene expression between the closest populations cannot be visualized
optimally following merging of the 15 minor clusters into the 5 major clusters (**Figure 2A**) used
for simplification of our comprehensive bioinformatics analyses. Notably, the 5 major clusters
in **Figure 2A** only show expression profiles of those genes that are deregulated upon CEBPE
KO.

We also like to emphasize that the 15 populations can be considered distinct developmental
stages based on their distinct immunophenotype / morphology, but most importantly in their
hierarchical mapping based on similarity of gene expression profiles in the PCA plot in **Figure**
**1D**. Indeed, the latter was corroborated by our new integrative analyses of the bulk RNA-seq
from our BM population data and scRNA-seq data, which demonstrated that single cells
annotated to a specific sorted BM population essentially clustered together and maintained the
same hierarchical order of monocytic and granulocytic differentiation trajectories observed in
the PCA plot (**Supplementary Figure S3A and Figure 1D**).

In the revised manuscript have reported these novel integrated analyses of scRNA-seq/bulk
RNA-seq BM population in detail in the results section (line 175-183).

Referee #1 – minor comment #2:

- The authors state:

"Overall, these findings demonstrate that *Cebpa* expression correlates with myeloid
commitment of HSCs (i.e. LSKs) toward lineage-restricted progenitors (i.e. GPs/PMs) and
expression of early granulocytic differentiation genes (i.e. primary granule proteins)."

- However, the word commitment should be replaced with specification (moving from a
multipotent state towards a single lineage, but not fully committed). Commitment should be a
word reserved for not being able to produce other lineages. However, the authors do not show
that this cluster is a commitment stage, and previous literature would suggest that it is myeloid
specification.

**Response referee #1 – minor comment #2:**

**We agree with the referee and have changed the commitment with specification throughout**
**the MS where indicated.**

**Referee #1 – minor comment #3:**

- The authors proposed enhancers for genes lack a clear genomics + informatics approach to
link enhancers to those genes. The assessment of genomic loci as enhancers >1,000 bp from
the transcription start site is ambiguous

- Would the authors consider a loci 1,000,000 bp from a TSS to be an enhancer of the gene?

- Promoter capture Hi-C or co-accessibility of genomic regions across a single-cell experiment
would provide better definition of enhancer-gene connections (again, if the authors had read
prior literature, they would see that some of these datasets already exist)

**Response referee #1 – minor comment #3:**

**The reviewer has rightly pointed out a well-known problem of studies on transcriptional**
**regulation, namely linking enhancers to their target genes. While we agree that enhancers do**
**not always target the closest gene, we still decided to use the “closest gene approach” for two**
**main reasons:**

**1. A recent study benchmarked different approaches for identifying experimentally**
**validated sets of enhancer-promoter interactions, and found that the closest gene**
**approach is among the best approaches to use with a high precision and recall of ~75%**
**(Figure 1C in Nasser et al., 2021, Nature). Notably, the performance, at least on the**
**tested set of genes, is higher as compared to promoter capture HiC.**

**2. We also explored a recently published dataset (Zhang et al. 2020, Cell Reports) in**
**which the authors have identified enhancer-promoter interactions in different**
**populations of hematopoiesis using tagHiC. However, due to sparsity of the contact**
**matrices, we couldn't find regulatory enhancers for many of the genes analyzed in our**

study, for example *Cebpa*, *Cebpe*, *Mmp9* and *Dhrs7* (Figure below). This limits the usage of such a dataset for our study as we miss most of the enhancer candidates. In fact, we experimentally validate one of these enhancer candidates to target the closest gene (*Dhrs7*, Supplementary Figure S7C and S7E).

Lastly, we also checked for the distribution of distance between enhancers and linked promoters using the closest gene approach (see figure below). We found that >95% of enhancer-promoter pairs are actually <200kb away from each other, with a median distance of ~40kb, which is the predominant length of fine-scale loops (10-50kb) observed for chromatin interactions between enhancer and promoter (Hsieh et al. Mol Cell, 2020).

distribution of distance between enhancer and target promoter defined using closest-gene approach

Referee #1 – minor comment #4:

- Information about the 3D chromatin shape is lacking. Is it the case that there are interactions provoked by CEBPE bound enhancer that help to explain the changes in gene expression?

Response referee #1 – minor comment #4:

We agree with the referee that 3D chromatin structure would be helpful to further improve our
understanding of how cell cycle exit and differentiation is regulated. However, we like to argue
that such 3D chromatin experiments would be a study in itself and are beyond the scope of the
current study.

Referee #1 – minor comment #5:

- Given the large number of motifs in the databases used ("Known TF-binding motifs were
downloaded from MEME, Jaspar, Uniprobe, Hocomoco and Jolma et al."), the authors should
include the motif scores for all of the tested motifs, at least in the supplemental information.

Response referee #1 – minor comment #5:

We agree with the reviewer that such information should be provided. We have now compiled
the motif scores for the tested motifs (MYC, E2F1, CEBPE, NFYB and LIN54, shown in Figure
3C, Figure 3E, and Figure 5B) at the gene promoter regions in Supplementary Data S1.

Referee #1 – minor comment #6:

- Ranked importance of TFs seems heavily biased towards the available ChIP-seq data. The
authors should comment on efforts to control for the importance of other TFs.

Response referee #1 – minor comment #6:

We acknowledge the reviewer's comment and would like to stress that we have conducted a
"ranked importance" analysis on TFs that were enriched in the cluster A(GMP) or B(MY) (i.e.
MYC and E2F1 as shown in Figure 3C) or the TFs that are known to be key regulators of early
and late granulocytic differentiation (CEBPA, CEBPE and PU.1) (Figure 3C and Figure 4A-B).
NFYB and LIN54 are not included in our "ranked importance" analysis because both these TFs
are specifically enriched at promoters of cluster B(PM) genes, but not enriched at promoters of
cluster A(GMP) genes.

Referee #1 – minor comment #7:

- There is a suggested direct interaction of CEBPE with E2Fs at G2/M promoters on the basis
of absence of CEBPE motifs at promoter regions

- Can co-immunoprecipitation validate the complex of CEBPE and E2F in the indicated
populations

Response referee #1 – minor comment #7:

Co-ChIP for CEBPE and multiple E2F family members on BM is technically very challenging
(i.e. not possible in our Lab) due to the limited number of cells available for sorted BM
populations and due to the fact that several activating and inhibitory E2F family members are
expressed differentially in multiple BM populations during early and late granulocytic
differentiation.

Hence, we decided to directly assess whether CEBPE binding to E2F is responsible for
repression E2F dependent transcription. For this, we conducted EMSA assays as described
previously by our group (Porse BT et al., Cell, 2001) utilizing the same experimental setup as
for the reporter assays, which demonstrated that CEBPE repressed E2F-dependent
transcription in a dose dependent manner (Supplementary Figure S4A). Our complementary
EMSA assays showed that CEBPE bound both to a CEBP probe as well as to E2F probes
bound by E2F/DP1 complexes (Supplementary Figure S4B). These findings support that
CEBPE represses E2F-mediated transcription through direct binding of E2F/DP1 complexes at
E2F promotor sites, rather than by sequestration E2F and preventing its binding to E2F
promotor sites. Importantly, the latter is consistent with our *in vivo* ChIP-seq data showing that
CEBPE binds at E2F binding sites of promoters in *Cebpe* +/+ mice leading to downregulation
of E2F target genes in MY1+MY2 populations concomitant with cell cycle exit, followed by
terminal granulocytic differentiation. Importantly, *Cebpe* -/- mice exhibited NO downregulation
of E2F target genes in MY1+MY2 populations and NO cell cycle exit followed by terminal
granulocytic differentiation.

These novel EMSA experiments are described in detail in line 267-271.

Referee #1 – minor comment #8:

- Can 3D chromatin conformation confirm chromatin looping of these enhancer-promoter
interactions? Alternatively, can the authors comment on how the regulation of this gene via this
enhancer is occurring?

Response referee #1 – minor comment #8:

Again, we agree with the referee that 3D chromatin structure would be helpful to further
improve our understanding of how cell cycle exit and differentiation is regulated. However, we
like to argue that such 3D chromatin experiments would be a study in itself and are beyond the
scope of the current study.

Referee #1 – minor comment #9

- Can the authors comment on the structural differences of CEBPA and CEBPE that may confer differential cell cycle activity (interaction with E2F complex)?

Response referee #1 – minor comment #9:

In the revised MS, we have conducted a series of new experiments, which collectively suggest that cell cycle exit and completion of granulocytic differentiation are partially regulated by CEBPs in a dose-dependent manner.

First, we demonstrated that similar to CEBPA, CEBPE can bind to DNA-bound E2F. Next, we tested whether high ectopic expression of CEBPE and CEBPA can promote granulocytic and monocytic differentiation. For this, KIT⁺ cells from *Cebpe* ^{+/+} as well as *Cebpe* ^{-/-} mice were retrovirally transduced with pBabe-Puro vectors expressing CEBPA-WT-ER and CEBPE-WT-ER. After puromycin selection cells were treated with 4-HT to induce nuclear translocation and transcriptional activity of CEBPA and CEBPE.

As demonstrated in (Supplementary Figure S3G and S3H) CEBPA induced both monocytic (CD115⁺/11b⁺ cells) and granulocytic differentiation (CD115⁻-Ly6G⁺ cells) in KIT⁺ *Cebpe* ^{+/+} whereas CEBPE only induced granulocytic differentiation (CD115⁻-11b⁺ cells) as compared to the empty control vector. Importantly, CEBPA also induced monocytic (CD115⁺/11b⁺ cells) and to a lesser also granulocytic differentiation (CD115⁻-Ly6G⁺ cells) in KIT⁺ *Cebpe* ^{-/-} as compared KIT⁺ *Cebpe* ^{+/+} cells. Again, CEBPE only induced granulocytic differentiation (CD115⁻-Ly6G⁺ cells) in KIT⁺ *Cebpe* ^{-/-} cells.

These findings suggest that that high levels/activity of nuclear CEBPA to some degree promotes *in vitro* granulocytic differentiation, but fail to do so *in vivo*, as CEPBA protein copy numbers in *Cebpe* ^{-/-} MY1+MY2 are 2.5- to 4-fold lower (i.e. approx. 100,000) compared to CEPBE protein copy numbers in *Cebpe* ^{+/+} MY1+MY2 (i.e. approx. 250,000-400,000) (Supplementary Figure S3E).

Overall, these novel experiments suggest that cell cycle exit and completion of granulocytic differentiation are partially regulated by CEBPs in a dose-dependent manner and not conferred by structural differences between CEBPA/CEBPE.

These novel experiments are reported and discussed in the revised MS in line 222-235, 531-539.

Referee #1 – minor comment #10:

- The authors should supply a supplementary file with each of the CEBPE motif positions that
they find interesting in describing the target genes of interest, such that they can provide
genomic coordinates for the regulatory activity of CEBPE.

Response referee #1 – minor comment #10:

We thank the reviewer for this very important and useful suggestion. In the revised MS, we
now provide information on the genomic coordinates of CEBPA/E binding, their potential target
genes (based on proximity), and the underlying CEBPE motif score in **Supplementary Data S2**.

Referee #1 – minor comment #11:

- Is there a reason the authors don't plot ChIP-seq profiles for CEBPA and CEBPE
simultaneously for the same populations (Figure 5E/F)?
- Do CEBPA and CEBPE ever bind to the same location in the same cell population?
- I would really like to see the dynamics of CEBPA and CEBPE binding across all of the
populations

Response referee #1 – minor comment #11:

To address these important comments, we performed additional CEBPA ChIP-seq
experiments on MY1s+MY2s/MMs, to compare CEBPA with CEBPE binding at CEBPA/E
genomic regions in MY1s+MY2s/MMs (**Figure 6D**). **Figure 6D** highlights that CEBPA and
CEBPE both bind to a common set (CEBPA & E) as well as exclusive sets (CEBPA or
CEBPE) of genomic regions. Notably, the common set (CEBPA & E) genomic regions were
bound by CEBPA during early differentiation in GMPs demonstrated decreased CEBPA
binding as cells differentiate into MY1s+MY2s/MMs and CEBPE binding takes over. These
sequential binding levels of CEBPA and CEBPE correlate with sequential expression of
CEBPA and CEBPE at the RNA/protein level during early and late granulocytic differentiation,
respectively. Importantly additional ChIP-seq analyses demonstrated that CEBPA binds to
primed enhancers in GMPs marked by H3K4me1 and CEBPE binds to active enhancers in
MY1+MY2/MM/GR marked by H3K27ac. The later suggests that CEBPA primes enhancers of
GR-specific genes during early granulocytic differentiation (i.e. in GMPs) before CEBPE takes
over and promotes cell cycle exit and upregulation of GR-specific genes during terminal
granulocytic differentiation (i.e. in MY1s+MY2s/MM/BC/GR).

In the revised manuscript we have discussed these novel extended ChIP-Seq analyses in line
375-394)

Referee #1 – minor comment #12:

- - Can the authors functionally validate the MYC/MAX activating sites of the *Cebpa* promoter?
- Would mutation at these MYC/MAX sites inhibit *Cebpa* promoter activity? (This could easily
be answered with a *Cebpa* reporter, cloning in the promoter upstream)

Response referee #1 – minor comment #12:

We thank the referee for this highly relevant comment and have now functionally validated the
previously reported MYC/MAX binding site of the *Cebpa* promoter in reporter assays
(Supplementary Figure S7F). Co-transfection HEK-293 cells with a *Myc* expression vector and
reporter constructs harboring a minimal *Cebpa* promoter with either wt or mutated MYC
binding sites demonstrated significantly higher activity of wt vs mutated reporter
(Supplementary Figure S7D). The latter favors the proposed hypothesis that CEBPE promotes
feedback inhibition of MYC-dependent *Cebpa* expression through (i) direct repression of E2F-
mediated *Myc* expression, but also (ii) through up-regulation of *Mad* (Supplementary Figure
S5G) which, in the absence of MYC can heterodimerize with MAX, and repress MYC target
genes such as *Cebpa*.

In the revised MS these novel experiments are reported and discussed in line 426-429 and line
547-551.

Referee #1 – minor comment #13:

- - Why are CEBPA and CEBPE written in all caps sometimes? Aren't all of the data generated
in mice, and should take on the convention of *Cebpa* and *Cebpb* with the first letter alone
capitalized?

Response referee #1 – minor comment #13:

To our best of our knowledge the convention for mouse protein and gene symbols is correct in
our MS (https://en.wikipedia.org/wiki/Gene_nomenclature):

Gene and protein symbol conventions ("sonic hedgehog" gene)		
Species	Gene symbol	Protein symbol
Homo sapiens	SHH	SHH
Mus musculus, Rattus norvegicus	Shh	SHH
Gallus gallus	SHH	SHH
Anolis carolinensis	shh	SHH
Xenopus laevis , X. tropicalis	shh	Shh

Danio rerio	shh	Shh
------------	-----

Referee #1 – minor comment #14:

- Shouldn't the title use identity instead of identify

- Eg. Transcription factor-driven coordination of cell cycle exit and specification of cell identity
during granulocytic differentiation

**Response referee #1 – minor comment #14:**

**We agree with the referee and have changed the title accordingly.**

**Reviewer #2 comments (total 12):**

General comments

In this study, Theilgaard-Monch, Pundhir, and Porse et al addressed a fundamental question
about the coordination of cell cycle exit and cell type specification by analyzing the role of
CEBPA and CEBPE transcription factors in granulocytic differentiation *in vivo*. By detailed
transcriptome analyses and some ChIP-seq analyses, the authors demonstrate that CEBPA
directly induces *Cebpe* gene expression, and CEBPE indirectly binds to promoter-proximal
regions through chromatin-bound E2F to inhibit E2F-mediated expression of cell cycle genes
including *Myc*. On the other hand, CEBPA and CEBPE directly bind to promoter-distal
enhancer regions to induce early and late granulocytic genes, respectively. The authors also
suggest the possible mechanism by which the gene expression of *Cebpa* is replaced by that of
*Cebpe* through the *Myc*-Mad-Max system. Overall, the data nicely describe the mechanism of
how CEBPA-induced CEBPE causes both cell cycle exit and terminal differentiation towards
granulocytes, which significantly contributes to the progress in the research field. On the other
hand, some of the conclusions, especially the concept that CEBPA transiently primes, while
CEBPE activates or unleashes, the terminal granulocytic differentiation program at the
promoter and enhancer level, are not fully supported by the data and require additional
experiments.

Specific comments

Major points

1. Summary: One of the significant points of this study is that the authors dealt with cells
freshly isolated from mice to address the “*in vivo*” mechanism. This would be better to be
stated in the Summary.

**Response referee #2 – comment #1:**

**We have now addressed the important aspect of investigating the “*in vivo*” mechanisms of cell
cycle exit and differentiation in our study in both title and abstract.**

2. The order of Fig numbers does not match between the text and Figures, making it difficult to
read the paper.

**Response referee #2 – comment #2:**

**We apologize for this inconvenience and have now matched the figure numbers consecutively
in the revised MS.**

3. GMPs are analyzed in Fig. 1D and the authors state in lines 181-182 and 194 that the Figs
2A and B include GMPs. However, Figs 2A, 2B and S6A do not display the data of GMPs. This
is inconsistent and confusing especially because the cluster A is named “GMP”. Showing the
GMP data should be important to understand the switch from CEBPA to CEBPE.

**Response referee #2 – comment #3:**

We have now added the GMP data in all relevant figures in the revised MS (Supplementary
Figure S2A, Figure 2A, Figure 2B, former Supplementary Figure S6A - now Supplementary
Figure S3D, Supplementary Figure S5A, Figure 5A).

4. Figs. 2B and S6A: Please show the expression of CEBP proteins as well. It would be even
more informative if the authors could estimate their molecular numbers per cell to address the
mechanism by which CEBPA and CEBPE play distinct roles (see Comment 7c).

**Response referee #2 – comment #4:**

We are very thankful for this very important suggestion by the referee! In the revised MS we
have now added complementary quantitative MS analysis of CEBPA and CEBPE protein
expression for relevant sorted BM populations to validate our RNAseq data at the protein level
(Supplementary Figure S3D and S3E). In addition, we have mapped *Cebpa* and *Cebpe*
expression in the newly analyzed scRNA-seq hierarchy in Supplementary Figure S3B-C.
Notably these complementary data demonstrate NO mutually exclusive expression but rather
and almost exclusive sequential expression of CEBPA and CEBPE during granulocytic
differentiation including a co-expression in the PM population. The latter is consistent with our
novel complementary CEBPA ChIP-seq data on populations with very low CEBPA
protein/RNA (MY1+2, MM) and very high CEBPE protein/RNA expression (Figure 6D).
Based on our MS protein analyses and the expression trajectories of *Cebpa* and *Cebpe*
mapped onto the novel scRNA-seq hierarchy (Supplementary Figure S3B-C), we now
demonstrated that the PM population exhibit co-expression of CEBPA/*Cebpa* and
CEBPE/*Cebpe* both at the protein and RNA levels (incl. scRNA level). Importantly, both RNA
and protein profiles also highlight that upregulation of CEBPE/*Cebpe* is correlated with a
marked downregulation of CEBPA/*Cebpa* during a very narrow developmental window when
cells exit cell cycle and initiate terminal granulocytic differentiation.

In the revised manuscript we have changed our statement “mutually exclusive expression” to
“almost exclusive sequential expression” of CEBPA/*Cebpa* and CEBPE/*Cebpe* accordingly
(line 196-201).

5. The authors state in lines 221-223 “Collectively, these findings suggest that CEBPE-
dependent cell cycle exit is a prerequisite to complete terminal granulocytic differentiation”.
Here, the fundamental question would be whether cell cycle arrest is required for the induction
of granulocytic genes, which is not addressed or discussed. If available, CEBPE mutants that
cannot bind to E2F but can bind to DNA and activate transcription could be transduced into
CEBPE KO cells.

**Response referee #2 – comment #5:**

As highlighted by our gene expression profiles, cell cycle analyses, and our novel MS and
ChIP-seq analyses the persistent presence of CEBPA protein in MY1/2 of *Cebpe* *-/-* mice
(MY1+2 in *Cebpe* *+/+* mice have high CEBPE and essentially lack CEBPA), is not sufficient to
compensate for the loss of CEBPE to induce cell cycle exit and subsequent upregulation of
genes critical for completion of terminal granulocytic differentiation after cell cycle exit.
Our statement in the MS “Collectively, these findings suggest that CEBPE-dependent cell
cycle exit is a prerequisite to complete terminal granulocytic differentiation” is merely based on
the sequence of events assessed during terminal granulocytic differentiation where cell cycle
exit precedes upregulation of terminal granulocytic genes and completion of differentiation.
Notably, **Supplementary Figure S4C-S4E** shows that CEBPE with a single point mutation (i.e.
CEBPE-BRM5-ER) that abrogates binding to E2F, as compared to CEBPE-WT, cannot induce
cell cycle exit and differentiation in 32D cells.

6. Fig. S6D: This Figure is very important, so should be shown as a main Figure. It would be
informative to show the overlap between CEBPA and CEBPE binding sites by Venn diagrams.

**Response referee #2 – comment #6:**

We agree with the referee and have now added this **Supplementary Figure S6D** as **Figure 6D**
in the revised manuscript. We have also shown the overlap of CEBPA and CEBPE binding
sites in a new Venn diagram in **Figure 6C**.

7. The conclusion about the sequential actions of CEBPA and CEBPE, stated in Summary
(“CEBPA promotes lineage commitment by launching an enhancer-primed differentiation
program” and “CEBPE unleashes the CEBPA-primed differentiation program”) and elsewhere,
is important and quite interesting but requires further validation and discussion.

—7a. Does CEBPA prime but not activate the enhancers targeted by both CEBPA and
CEBPE? This can be tested by ChIP-seq of CEBPA KO cells for H3K4me1 and
H3K27ac, and the authors might already have such data in the previous study by
Pundhir et al, Cell Rep 2018.

—7b. Is CEBPE required for activating the enhancers that had been primed by CEBPA?
This can be tested by ChIP-seq of CEBPE KO cells for H3K27ac.

Response referee #2 – comment #7a + #7b:

We thank the reviewer for the important comments and highly relevant suggestions. To understand the priming/activating role of CEBPA and CEBPE, we performed novel H3K27ac ChIP-seq experiments in CEBPE WT MY1+MY2, MM and integrated these novel ChIP-seq data with our previously published ChIP-seq data on H3K27ac of GR/MO and H3K4me1 ChIP-seq data on LSK, preGM, and GMP (Pundhir et al., Cell Reports, 2018). Below we summarize and discuss these novel results shown in **Figure 6D-F**:

1. CEBPA and CEBPE bound sequentially to a common set of enhancers during early (i.e. in GMPs) and late (i.e. in MY1s+MY2s/MMs/GRs) granulocytic differentiation, respectively. Notably enhancers were indeed primed (i.e. H3K4me1 high - H3K27ac low) concomitant with CEBPA binding in GMPs and activated (i.e. H3K27ac high) concomitant with CEBPE binding in MY1s+MY2s/MMs/GRs cells (**Figure 6D**). However, this CEBPE-dependent activation of enhancers is not exclusive to CEBPE. Specifically, we observed that CEBPA can also activate these enhancers (H3K27ac levels increase or are maintained) in CEBPE KO MY1+2 and MM cells (**Figure 6E-F**).
2. Using our new ChIP-seq data, we also identified a subset of enhancers only bound by CEBPE (i.e. not bound by CEBPA).
3. In striking contrast to the common set of enhancers bound by both CEBPA & CEBPE, the CEBPE-bound subset of enhancers was not bound by CEBPA and exhibited no activation (i.e. H3K27ac marked) neither in CEBPA expressing WT GMPs and nor in CEBPE KO MY1s+MY2s/MMs. This strongly indicates that CEBPA does neither prime nor activate this subset of CEBPE-bound enhancers *in vivo* (**Figure 6D-F**). Interestingly, a large fraction of these CEBPE-bound enhancers (71%) potentially regulate the expression of genes that are upregulated during terminal granulocytic differentiation (**Figure 6D**).

In the revised MS we have now reported and discussed these novel findings in the results section and the discussion (line 375-394, 464-480)

—7c. I feel that it will be natural for readers to wonder the mechanism by which CEBPA and CEBPE belonging to the same family can play cooperative but distinct roles. Is it due to the timing of their expression, molecular copy numbers (see Comment 4), or their distinct intrinsic activities? Although I understand that a single paper cannot solve

everything, this highly relevant issue should be addressed or at least fully discussed.
Can ectopic expression of CEBPE and CEBPA rescue granulocytic differentiation of
CEBPA KO and CEBPE KO cells, respectively? When CEBPE is lost, CEBPA
expression is significantly upregulated, but cannot compensate for the loss of CEBPE to
control gene expression; does CEBPA bind to the CEBPE target sites in these cells
without activating enhancers and promoters (can be tested by ChIP-seq of CEBPA in
CEBPE KO and WT cells)? These additional experiments might help addressing the
issue.

Response referee #2 – comment #7c:

To test whether high ectopic expression of CEBPE and CEBPA can promote granulocytic and
monocytic differentiation, KIT⁺ cells from *Cebpe* ^{+/+} as well as *Cebpe* ^{-/-} mice were retrovirally
transduced with pBabe-Puro vectors expressing CEBPA-WT-ER and CEBPE-WT-ER. After
puromycin selection cells were treated with 4-HT to induce nuclear translocation and
transcriptional activity of CEBPA and CEBPE.

As demonstrated in (Supplementary Figure S3G and S3H) CEBPA induced both monocytic
(CD115⁺/11b⁺ cells) and granulocytic differentiation (CD115⁻Ly6G⁺ cells) in KIT⁺ *Cebpe* ^{+/+}
whereas CEBPE only induced granulocytic differentiation (CD115⁻11b⁺ cells) as compared to
the empty control vector. Importantly, CEBPA also induced monocytic (CD115⁺/11b⁺ cells)
and to a lesser also granulocytic differentiation (CD115⁻Ly6G⁺ cells) in KIT⁺ *Cebpe* ^{-/-} as
compared to KIT⁺ *Cebpe* ^{+/+} cells. Again, CEBPE only induced granulocytic differentiation
(CD115⁻Ly6G⁺ cells) in KIT⁺ *Cebpe* ^{-/-} cells.

These findings suggest that high levels/activity of nuclear CEBPA to some degree
promotes *in vitro* granulocytic differentiation, but fail to do so *in vivo*, as CEBPA protein copy
numbers in *Cebpe* ^{-/-} MY1+MY2 are 2.5- to 4-fold lower (i.e. approx. 100,000) compared to
CEBPE protein copy numbers in *Cebpe* ^{+/+} MY1+MY2 (i.e. approx. 250,000-400,000)
(Supplementary Figure S3E).

Hence, these findings suggest that cell cycle exit and completion of granulocytic differentiation
are partially regulated by CEBPs in a dose-dependent manner.

We also performed novel complementary CEBPA ChIP-seq analyses of MY1+MY2 purified
from *Cebpe* ^{+/+} and *Cebpe* ^{-/-} (Figure 6D-F). Comparison of CEBPA ChIP-seq vs. CEBPE
ChIP-seq demonstrated that CEBPE and CEBPA binds both common and distinct sets of
genomic regions suggesting that CEBPE might specifically regulate genes expressed during
granulocytic differentiation. However, this differential binding of CEBPA and CEBPE to
genomic regions might partially be dose-dependent, given that CEBPA protein copy numbers
in *Cebpe* ^{-/-} MY1+MY2 are 2.5- to 4-fold lower (i.e. approx. 100,000) compared to CEBPE
protein copy numbers in *Cebpe* ^{+/+} MY1+MY2 (i.e. approx. 250,000-400,000). Overall, these
novel experiments suggest that cell cycle exit and completion of granulocytic differentiation are

partially regulated by CEBPs in a dose-dependent manner and not conferred by structural
differences between CEBPA/CEBPE.

These novel experiments are reported and discussed in the MS in line 222-235, 531-539.

Minor points

8. Fig. 2C: The print “Gene expr..” is not fully shown.

Response referee #2 – comment #8:

We checked the MS version processed by the submission software, however we couldn't
reproduce the problem. We therefore submit a separate Figure 2C with the revised MS (in
addition to the Figure 2C inserted in the MS).

9. Fig. 3A: The print of GO terms is not fully shown (the right portions are missing).

Response referee #2 – comment #9:

We checked the MS version processed by the submission software, however we couldn't
reproduce the problem. We therefore submit a separate Figure 3A with the revised MS (in
addition to the Figure 3A inserted in the MS).

10. According to the Materials and Methods section, the authors retrieved ChIP-seq data of
histone modifications from previous publications to define the enhancer regions. This is better
to be stated in the main text.

Response referee #2 – comment #10:

We have now stated the use of published ChIP-seq from previous publications in the main text
of the revised MS (ref. 25 - Pundhir et al., 2018, Cell Reports, ref. 40 - Chen et al., 2008, Cell;
line 248-250).

**Reviewer #3 comments (total 3):**

In this manuscript the authors develop a sorting strategy to isolate cells at various stages of
neutrophil and monocyte differentiation and then characterize the gene expression profiles and
chromatin occupancy profiles for CEBPE. They use this newly generated data and compare it
to previously generated data for CEBPA binding to propose the mechanism of coordinate
regulation of gene expression by these two important transcription factors. Based on these
data they also propose that CEBPA and CEBPE are inducing different cellular phenotypes at
different stages of differentiation. Overall, the newly described approach to isolating cells at
various stages of myelomonocytic differentiation is of interest as are the gene expression and
ChIP-seq data generated from these cells. The major concern is that the overwhelming
majority of the data is correlative and while fairly compelling it seems more functional
characterization would help significantly.

Referee #3 – comment #1:

Specifically, it would be very helpful to better characterize the 32D cell system they show in
supplemental data. They demonstrate that expression of CEBPE in these cells induces
differentiation and cell cycle exist, but it wasn't clear to me if they have shown that CEBPE is
binding to chromatin in these cells in a way that is consistent with the hypothesis generated
from primary cells. More clarity (and experiments) here would be very helpful.

**Response referee #3 – comment #1:**

**We acknowledge this important point by the referee but we also have several experimental**
**concerns with the suggested novel experiments! First, we like to emphasize that the aim of**
**our study was to investigate the sequence and potential regulatory mechanisms of *in vivo***
**granulocytic differentiation at high resolution (i.e. in 10 developmental stages LSK>GR), which**
**is unlikely to be recapitulated properly by any *in vitro* differentiation model. Although our *in vitro***
**32DCI3 CEBPE-WT-ERTM differentiation model will have some similarities of *in vivo***
**differentiation it will also fail to recapitulate many aspects of high resolution *in vivo* granulocytic**
**differentiation. Indeed, the latter is corroborated by some previous and novel experiments**
**reported in the revised MS. More specifically, 4-HT treated 32DCI3 CEBPE-WT-ERTM cells**
**cease proliferation and differentiate as heterogenous bulk population (based surface marker**
**expression) within 2 and 4 days, respectively as compared to the 10 days of *in vivo***
**differentiation of GMPs into mature GRs. We therefore like to argue that complementary**
**transcriptomics, ChIP-seq, and protein analyses of our 32DCI3 CEBPE-WT-ERTM**

differentiation model will only partially recapitulate our current *in vivo* data, and therefore not
be of substantial value to validate our comprehensive *in vivo* data.
With respect to CEBPE binding to chromatin, we conducted novel EMSA assays to directly
assess whether CEBPE binding to E2F is responsible for repression E2F dependent
transcription. More specifically, we utilized the same experimental setup for EMSA assays
(Porse BT et al., Cell, 2001) as for the reporter assays, which demonstrated that CEBPE
repressed E2F-dependent transcription in a dose dependent manner (Supplementary Figure
S4A). Our complementary EMSA assays showed that CEBPE bound both to a CEBP probe as
well as to E2F probes bound by E2F/DP1 complexes (new Supplementary Figure S4B). These
findings support that CEBPE represses E2F-mediated transcription through direct binding of
E2F/DP1 complexes at E2F promotor sites, rather than by sequestration E2F and preventing
its binding to E2F promotor sites. Importantly, the latter is consistent with our *in vivo* ChIP-seq
data showing that CEBPE binds to E2F binding sites of promoters in *Cebpe* *+/+* mice leading
to downregulation of E2F target genes in MY1+MY2 populations concomitant with cell cycle
exit, followed by terminal granulocytic differentiation. Importantly, *Cebpe* *-/-* mice exhibited NO
downregulation of E2F target genes in MY1+MY2 populations and NO cell cycle exit followed
by terminal granulocytic differentiation.

These novel EMSA experiments are described in detail in line 267-271.

Referee #3 – comment #2:

Also, there is an implication that CEBPE and CEBPA are expressed at specific stages of
myeloid development to drive specific phenotypes. Does this mean that CEBPA and CEBPE
induce different gene expression programs at different stages of development?

Response referee #3 – comment #2:

We are very thankful for this important point made by the referee and addressed this by
several complementary experiments and bioinformatics analyses. Our novel MS analyses
(Supplementary Figure S3E) of CEBPA and CEBPE protein expression, novel ChIP-seq data
for CEBPA in MY1+MY2 and MM, as well as H3K4me1 and H3K27ac ChIP-seq data (Figure
6D-F), and the mapping of *Cebpa* and *Cebpe* expression in distinct developmental trajectories
of a murine BM scRNA-seq hierarchy (Supplementary Figure S3A-S6C) strongly supports that
CEBPE and CEBPA promote distinct albeit to some extent overlapping gene expression
programs during early granulocytic development (exclusively CEBPA), as well as monocytic
(exclusively CEBPA) and late granulocytic differentiation (almost exclusively CEBPE).

In the revised manuscript we have reported these novel experiments in line 175-183, 196-201,
375-394, and discussed our findings in line 444-455, 464-467, 531-539.

Referee #3 – comment #3:

Is there some way that the authors could demonstrate this functional difference in the 32D cell
system? More insight in regard to how they two transcription factors differ functionally would
also add much needed clarity to the field.

Response referee #3 – comment #3:

We thank the reviewer for this comment and while we do appreciate the potential of the 32D
system to assess the contribution of CEBPE and CEBPA to granulocytic differentiation we
deemed it more relevant to address this in the context of *in vivo* hematopoiesis.

Therefore, to test whether high ectopic expression of CEBPE and CEBPA can promote
granulocytic and monocytic differentiation, KIT⁺ cells from *Cebpe* ^{+/+} as well as *Cebpe* ^{-/-}
mice were retrovirally transduced with pBabe-Puro vectors expressing CEBPA-WT-ER or
CEBPE-WT-ER. After puromycin selection cells were treated to with 4-HT to induce nuclear
translocation and transcriptional activity of CEBPA and CEBPE.

As demonstrated in (Supplementary Figure S3G and S3H) CEBPA induced both monocytic
(CD115⁺/11b⁺ cells) and granulocytic differentiation (CD115⁻-Ly6G⁺ cells) in KIT⁺ *Cebpe* ^{+/+}
whereas CEBPE only induced granulocytic differentiation (CD115⁻-11b⁺ cells) as compared to
the empty control vector. Importantly, CEBPA also induced monocytic (CD115⁺/11b⁺ cells)
and to a lesser also granulocytic differentiation (CD115⁻-Ly6G⁺ cells) in KIT⁺ *Cebpe* ^{-/-} as
compared KIT⁺ *Cebpe* ^{+/+} cells. Again, CEBPE only induced granulocytic differentiation
(CD115⁻-L6G⁺ cells) in KIT⁺ *Cebpe* ^{-/-} cells.

These findings suggest that that high levels/activity of nuclear CEBPA to some degree
promotes *in vitro* granulocytic differentiation, but fail to do so *in vivo*, as CEPBA protein copy
numbers in *Cebpe* ^{-/-} MY1+MY2 are 2.5- to 4-fold lower (i.e. approx.100,000) compared to
CEPBE protein copy numbers in *Cebpe* ^{+/+} MY1+MY2 (i.e. approx. 250,000-400,000)

(Supplementary Figure S3E).

Hence, these findings suggest that cell cycle exit and completion of granulocytic differentiation
are partially regulated by CEBPs in a dose-dependent manner.

We also performed novel complementary CEBPA ChIP-seq analyses of MY1+MY2 purified
from *Cebpe* ^{+/+} and *Cebpe* ^{-/-}. Comparison of CEBPA ChIP vs. CEBPE ChIP-seq
demonstrated that CEBPE and CEBPA binds both common and distinct sets of genomic
regions suggesting that CEBPE might specifically regulate genes expressed during
granulocytic differentiation. However, this differential binding of CEBPA and CEBPE to
genomic regions might partially be dose-dependent, given that CEPBA protein copy numbers

in *Cebpe* *-/-* MY1+MY2 are 2.5- to 4-fold lower compared to CEPBE protein copy numbers in
*Cebpe* *+/+* MY1+MY2 (Supplementary Figure S3E).

To conclude, CEBPA and CEBPE appear to have the same functional competences but our
data suggest that the failure of CEBPA to rescue granulocytic differentiation in the *Cebpe* *-/-* is
due to the lower levels of CEBPA as compared to CEBPE in the WT context.

These novel experiments are reported and discussed in the revised MS in line 222-235, 531-
539.

REVIEWER COMMENTS

Reviewer #1 (Remarks to the Author):

Overall, the authors responded to the most major and minor weaknesses identified in the manuscript. However, they did not provide evidence at a single-cell level that the populations they are isolating from the bone marrow are transcriptionally homogenous. Removing statements that describe these populations as homogenous would make the claims consistent with the data presented and would result in a valuable publication to the field. Notably, there are remaining problems with the first five major weaknesses.

Major weaknesses:

Comment #1

- Author's don't connect their findings to previously published populations of neutrophil and monocytic progenitors
- What are the overlapping populations relative to previous studies?
- Monocytic and granulocytic progenitor populations have been described in detail with scRNA-seq experiments, combined with cell surface protein experiments to identify subpopulations in these trajectories. The authors make no attempt to reference these prior studies or integrate the labels of previously identified populations to their new scheme.

The authors made an effort to map their identified bulk expression profiles to single cell RNA-seq experiments that include myeloid differentiation. However, little to no effort was made to try to understand the overlap of current isolation protocols. They claim that differences in the markers used and the gating strategies prevent this understanding. If all markers are included in the same panel, then it would be possible to see the overlap of the identified populations.

Comment #2

- It would be fairly straightforward to integrate the labels provided for new populations by doing a correlation of RNA-seq vectors between the pseudobulks and bulk RNA-seq of the populations the authors identify. Of course, if the populations studied actually correspond to more than one cluster from single cell studies then this would fail.
- Expression of cell cycle genes at different stages of neutrophil progenitor populations has previously been explored with scRNA-seq and should be explored/referenced

The authors took the bulk expression profiles from their sorted populations and mapped these to scRNA-seq for cells undergoing myeloid differentiation. This is fine, but these assays do not strongly suggest that the sorted BM populations are highly homogenous, as the authors state. It is not possible to tell if the sorted populations are homogenous because they are captured in bulk. Because it is in bulk, the data that the authors have collected are averages of cell populations. It does not consider whether there are multiple populations or smaller contaminating populations that exist inside the sorted gates. Based on previously published CITE-seq data, the populations that the authors have sorted are not transcriptionally homogenous.

Comment #3

- It is not clear whether there is heterogeneity within the identified populations, for which single cell assays are required to validate.
- There is nowhere near enough data to justify that these populations are transcriptionally distinct and pure relative to single-cell heterogeneity

The authors response to this comment is absolutely not sufficient. Single cell assays are required to show that the single-cells in the proposed sorted gates are homogenous. Bulk assays will

average together the signals from multiple populations. Those averages will easily map to distinct positions in cellular differentiation trajectories. This does not mean that the sorted populations are homogenous. It simply means that the averaged signal from the bulk assay represents a single point in the high dimensional expression space. This emphasizes the need for the authors to look at more markers within the panel they are using for sorting.

Comment #4

- The separation of populations is based on only 8 markers (Lin, Kit, Sca, CD115, CD16/32, CD34, Ly6G, CD11b)
- Again, need to respond to work that was previously published in this area
- Other groups are using assays that capture signals from >20 markers (CYTOF and CITE-seq). Can the authors comment on the purpose of using such tools if 8 markers is sufficient to isolate these populations of interest?

Here, the authors state that including more markers could further improve the resolution of sorted BM populations, yet above, they said that they had isolated pure populations with their panel. Both statements can't be true, if you can improve the resolution with more antibodies, then the populations you are identifying are not pure. The authors should remove any statements from the manuscript claiming that the BM populations are homogenous.

Comment #5

- The authors claim that unipotent granulocytic (GP) and monocytic (MP) progenitor populations exist, yet they don't validate these predictions with colony forming assays or lineage tracing experiments, or reference previous literature to validate that these populations are unipotent progenitors for the specified trajectories.

To respond to this point, the authors reference a prior study in which they observed ~80% of lineage restricted progenitors in their gates that are unipotent granulocytic and monocytic. It is confusing that the authors do not try to repeat this with their proposed new "homogenous" populations, as surely they should improve the accuracy to which the unipotent progenitors give rise to lineage restricted colonies. Again, either the authors should adjust the language of the text such that they are not claiming that their populations are homogenous, or they should redo this assay and show that they can improve the accuracy.

Comment #6

- The authors claim that Cebpa and Cebpe show mutually exclusive expression profiles, but this is wrong. Figure 2B is certainly insufficient to claim this. First of all, these are bulk RNA-seq samples, so it is not clear if single cells simultaneously express Cebpa and Cebpe. Previously published scRNA-seq data from multiple papers shows that there are many cells that co-express these genes. This claim is wrong and no data presented by the authors provides conclusive evidence to the contrary.

The authors further investigated this point and discovered that a population of cells do in fact co-express Cebpa and Cebpe, referred to as the "PM" population. It is good that they changed the language of the manuscript so that false statements are not being made.

Comment #7

- The effect of CEBPE from enhancers and promoters at this point is only an association of the binding of CEBPE to the change in gene expression at different stages. No causal role for CEBPE can be made without functional validation of genomic loci on target gene expression.
- For example, the authors could mutate CEBPE binding sites in promoters and enhancers and test for changes in target gene expression
- Alternatively, CRISPRi assays could be performed to silence the chromatin regions for functional validation

The authors validated some examples of targets of CEBPE effectively.

Minor Comment #1

- Gene clusters identified in 2A are overlapping in areas
- Eg. what is the difference between: LSK vs preGM? GP vs PM? MY1 vs MY2? MO1 vs MO2?
- If these are unique populations, they should certainly have unique gene expression patterns...

The authors clarified that there are unique gene expression patterns in the given clusters. It is clear from the PCA analysis in Figure 1D that the bulk transcriptomes reflect unique expression patterns. However, it is still not clear if the sorted populations are homogenous.

Minor Comment #2

- The authors state:

"Overall, these findings demonstrate that Cebpa expression correlates with myeloid commitment of HSCs (i.e. LSKs) toward lineage-restricted progenitors (i.e. GPs/PMs) and expression of early granulocytic differentiation genes (i.e. primary granule proteins)."

- However, the word commitment should be replaced with specification (moving from a multipotent state towards a single lineage, but not fully committed). Commitment should be a word reserved for not being able to produce other lineages. However, the authors do not show that this cluster is a commitment stage, and previous literature would suggest that it is myeloid specification.

The authors agreed and changed the manuscript to fix this issue.

Minor Comment #3

- The authors proposed enhancers for genes lack a clear genomics + informatics approach to link enhancers to those genes. The assessment of genomic loci as enhancers >1,000 bp from the transcription start site is ambiguous
- Would the authors consider a loci 1,000,000 bp from a TSS to be an enhancer of the gene?
- Promoter capture Hi-C or co-accessibility of genomic regions across a single-cell experiment would provide better definition of enhancer-gene connections (again, if the authors had read prior literature, they would see that some of these datasets already exist)

The authors clarified this in the text effectively.

Minor Comment #4

- Information about the 3D chromatin shape is lacking. Is it the case that there are interactions provoked by CEBPE bound enhancer that help to explain the changes in gene expression?

The authors claim that studying the 3D chromatin shape is outside the scope of the paper, which is acceptable after they provided more support for the functional role of Cebpe later in the text.

Minor Comment #5

- Given the large number of motifs in the databases used ("Known TF-binding motifs were downloaded from MEME, Jaspar, Uniprobe, Hocomoco and Jolma et al."), the authors should include the motif scores for all of the tested motifs, at least in the supplemental information.

The authors provided the requested information.

Minor Comment #6

- Ranked importance of TFs seems heavily biased towards the available ChIP-seq data. The authors should comment on efforts to control for the importance of other TFs.

The authors clarified this.

Minor Comment #7

- Can co-immunoprecipitation validate the complex of CEBPE and E2F in the indicated populations?

The authors provided EMSA experiments and updated the manuscript to provide validation for this mechanism.

Minor Comment #8

- Can 3D chromatin conformation confirm chromatin looping of these enhancer-promoter interactions? Alternatively, can the authors comment on how the regulation of this gene via this enhancer is occurring?

Again, the 3D chromatin structure is proposed to be outside the scope of the paper.

Minor Comment #9

- Can the authors comment on the structural differences of CEBPA and CEBPE that may confer differential cell cycle activity (interaction with E2F complex)?

The authors showed in vitro that CEBPA can induce granulocytic differentiation and argue that the structural differences do not confer the differential activity observed.

Minor Comment #10

- The authors should supply a supplementary file with each of the CEBPE motif positions that they find interesting in describing the target genes of interest, such that they can provide genomic coordinates for the regulatory activity of CEBPE.

The authors provided this information.

Minor Comment #11

- Is there a reason the authors don't plot ChIP-seq profiles for CEBPA and CEBPE simultaneously for the same populations (Figure 5E/F)?
- Do CEBPA and CEBPE ever bind to the same location in the same cell population?
- I would really like to see the dynamics of CEBPA and CEBPE binding across all of the populations

The authors provide additional ChIP-seq data which helps to understand the specification event.

Minor Comment #12

- Can the authors functionally validate the MYC/MAX activating sites of the Cebpa promoter?
- Would mutation at these MYC/MAX sites inhibit Cebpa promoter activity? (This could easily be answered with a Cebpa reporter, cloning in the enhancer upstream)

The authors added this experiment to the paper.

Minor Comment #13

- Why are CEBPA and CEBPE written in all caps sometimes? Aren't all of the data generated in mice, and should take on the convention of Cebpa and Cebpb with the first letter alone capitalized?

The authors clarified this misunderstanding.

Minor Comment #14

- Shouldn't the title use identity instead of identify
- Eg. Transcription factor-driven coordination of cell cycle exit and specification of cell identity during granulocytic differentiation

The authors changed this.

Reviewer #2 (Remarks to the Author):

The authors have addressed all of my comments and suggestions. Particularly, I am very happy to see that the authors performed a quantitative MS analysis of CEBPA and CEBPE proteins to directly compare their expression levels. Overall, the quality of the paper has been significantly improved.

Reviewer #3 (Remarks to the Author):

The authors have performed a number of new experiments that significantly improve the manuscript and start to address some of the functional predictions derived from their RNA-seq and ChIP-seq experiments. While the study remains largely descriptive, the new functional experiments and extended analyses produce a study that will be of interest to the field and is performed about as well as possible given current technical limitations.

**Point-to-point response to the referees' comments:**

**Referee #1 (Remarks to the Author):**

Overall, the authors responded to the most major and minor weaknesses identified in the manuscript.
However, they did not provide evidence at a single-cell level that the populations they are isolating
from the bone marrow are transcriptionally homogenous. **Removing statements that describe these**
**populations as homogenous would make the claims consistent with the data presented and would**
**result in a valuable publication to the field.** Notably, there are remaining problems with the first five
major weaknesses.

**We thank the reviewer for the encouraging comments and thorough assessment of our work. We**
**realize that we have not addressed the remaining problems regarding our sorting protocols in an**
**adequate manner in our first revision/comments. Admittedly, the focus of our work has not been to**
**provide (or claim) a superior sorting strategy, but merely to develop a robust protocol that allowed us**
**to sort cells representing snapshots of mainly granulocytic differentiation for our downstream**
**analyses, which constitutes the center of our work. However, we do hope that we have now clarified**
**the concerns regarding our sorting strategy in the revised manuscript and in our responses below.**

Major weaknesses:

**Comment #1**

- Author's don't connect their findings to previously published populations of neutrophil and
monocytic progenitors

- What are the overlapping populations relative to previous studies?

- Monocytic and granulocytic progenitor populations have been described in detail with scRNA-seq
experiments, combined with cell surface protein experiments to identify subpopulations in these
trajectories. The authors make no attempt to reference these prior studies or integrate the labels of
previously identified populations to their new scheme.

The authors made an effort to map their identified bulk expression profiles to single cell RNA-seq
experiments that include myeloid differentiation. However, little to no effort was made to try to
understand the overlap of current isolation protocols. They claim that differences in the markers used
and the gating strategies prevent this understanding. If all markers are included in the same panel,
then it would be possible to see the overlap of the identified populations.

**Response referee #1 comment #1:**

**The aim of this work was to characterize molecular events (TF binding, epigenetic configurations)**
**during myeloid differentiation and to that end we needed a sorting strategy that could resolve especially**
**the granulocytic differentiation trajectory into snapshots representing the gradual acquirement of**
**characteristics of neutrophil granulocytes.**

In the revised MS we have now made efforts to compare our populations with that of two
comprehensive complementary protocols for sorting of murine BM populations (derived from sc
analyses) representing successive early and late stages of granulocytic and monocytic differentiation
(Kwok et al., Immunity, 2020; Evrad et al., Immunity, 2018). We have also compared our populations
with another seminal study applying scRNA-seq to identify the transcriptional landscape of granulocytic
differentiation including distinct curated molecular signatures of sorted BM populations representing
successive developmental stages (Xie et al., Nature Immunology, 2020).

We like to acknowledge that the published sorting protocols (Kwok et al., Immunity, 2020; Evrad et al.,
Immunity, 2018) and ours differ with respect to the immunophenotypes of the sorted populations due
the application of different markers and most importantly gating levels for shared markers. As a
consequence, the protocols differ with respect to the total numbers of sorted populations representing
successive developmental stages.

The protocol reported by Xie et al initial gates away lymphoid cells, eosinophils and erythroid
progenitors before resolving the granulocytic hierarchy into five populations using a combinations of
KIT and Ly6G, which bare some resemblance to our work but with less markers to distinguish
especially granulocytic differentiation (see below). In contrast, the protocols by Kwok et al. and Evrad
et al. used several CD markers not applied in our sorting protocol (Kwoc: 106/81; Evrad CXCR2+4,
SiglecF) as well as overlapping markers (Kwoc: SCA-1, KIT, 115, 16/32, 11b, Ly6G; Evrad: SCA-1,
KIT, 115, 16/32, 11b, Ly6G). However, our BM populations differed from those of Kwok et al. and
Evrad et al.. as they were sorted based on 4 levels of KIT expression (i.e. Kit
high/intermediate/low/negative Fig 1B) and 3 levels of Ly6G expression (hi/int/neg) and 2 different SSC
levels rather than 2 expression levels for KIT and Ly6G and no discrimination of SSC levels. We like to
notice that this aspect of our gating strategy is important as it allowed to discriminate between GP (Lin-
neg/115-neg/KIT-hi/34pos/16+32-pos/11b-neg/Ly6G-neg) and PM (Lin-neg/115-neg/KIT-
int/34pos/16+32-pos/11b-neg/Ly6G-neg), and MY1 (Lin-neg/115-neg/KIT-lo/34neg/16+32-pos/11b-
pos/ly6G-neg) and MY2 (Lin-neg/115-neg/KIT-lo/34neg/16+32-pos/11b-pos/ly6G-int). Moreover, the
post-mitotic more mature cells could be discriminated as MM (Lin-neg/115-neg/KIT-neg/34neg/16+32-
pos/11b-pos/ly6G-int), BC(Lin-neg/115-neg/KIT-neg/34neg/16+32-pos/11b-pos/ly6G-hi/SSC-lo) and
GR (Lin-neg/115-neg/KIT-neg/34neg/16+32-pos/11b-pos/ly6G-hi/SSC-hi). Hence, we applied a sorting
strategy where we sorted populations that stepwise downregulated a marker of early hematopoietic
differentiation and proliferation (4 levels of KIT expression) in combination with the upregulation of a
marker of granulocytic differentiation (3 levels of Ly6G) and most terminal maturation
(SSC=granularity). We would like to note that the distinct level of granularity (i.e. SSC level) between
BC and GR is a specific cellular characteristic of terminal granulocytic maturation that cannot be
detected by surface markers.

To compare our populations with previously reported populations and scRNA-seq based signatures of
developmental stages, we generated UMAPs and assessed the correlations of gene expression
signatures for populations and scRNA-seq signatures defining specific developmental stages. As
depicted in the new Fig S3 our populations revealed a substantial transcriptomic overlap with published
gene expression signatures representing successive developmental stages, albeit at a somewhat
higher resolution due to the fact that we report a slightly higher number of developmental stages along
the monocytic and granulocytic differentiation trajectories.

We therefore like to argue that our purified BM populations are well suited for the aim of our study, i.e.
to investigate the sequential transcription factor-driven coordination of cell cycle exit and lineage-
specification during granulocytic differentiation.

Line 181-184 - we have added the following text including a new Figure S3.

*Additional validation demonstrated that the gene expression profiles of sorted BM populations were*
*similar to those of previously reported BM populations and scRNA-seq bulk signatures representing*
*successive developmental stages of granulocytic and monocytic differentiation (Figure S3) ²²⁻²⁴.*

**Comment #2**

- It would be fairly straightforward to integrate the labels provided for new populations by doing a
correlation of RNA-seq vectors between the pseudobulks and bulk RNA-seq of the populations the
authors identify. Of course, if the populations studied actually correspond to more than one cluster
from single cell studies then this would fail.

- Expression of cell cycle genes at different stages of neutrophil progenitor populations has previously
been explored with scRNA-seq and should be explored/referenced

The authors took the bulk expression profiles from their sorted populations and mapped these to
scRNA-seq for cells undergoing myeloid differentiation. This is fine, but these assays do not strongly
suggest that the sorted BM populations are highly homogenous, as the authors state. It is not possible
to tell if the sorted populations are homogenous because they are captured in bulk. Because it is in
bulk, the data that the authors have collected are averages of cell populations. It does not consider
whether there are multiple populations or smaller contaminating populations that exist inside the
sorted gates. Based on previously published CITE-seq data, the populations that the authors have
sorted are not transcriptionally homogenous.

**Response referee #1 comment #2:**

We thank the referee for this very important comment!! We completely agree with the referee that
we cannot state that the sorted BM populations are homogenous or pure with respect to the biological
state of their individual cells, but rather reflect bulk populations of individual cells that share a specific
immunophenotype and generate average transcriptomic signature of its individual cells! By habit, we
used the terms homogenous and pure populations in the context of their specific immunophenotype
and not biological/transcriptional state, which we now realize is incorrect and misleading.

We have corrected the text accordingly in the MS and removed our statements that the sorted
populations are homogenous or pure with respect to the biological state of their individual cells.

Line 185-186 - we have omitted the text highlighted in yellow:

*"The latter highlights that the sorted BM populations are highly homogeneous at the single cell level*
*and faithfully recapitulate the developmental trajectories of granulocytic and monocytic differentiation."*

The passage now reads:
*This highlights that the sorted BM populations faithfully recapitulate myeloid developmental trajectories*
*by representing successive stages of granulocytic and monocytic differentiation.*

Line 454-457 - we have omitted the text highlighted in yellow:

*Indeed, the latter was corroborated by annotation of single cell RNA-seq profiles to bulk RNA-seq*
*profiles of sorted BM population, which confirmed that the sorted BM populations are highly*
*homogeneous at the single cell level and faithfully recapitulate the developmental trajectories of early*
*and late granulocytic differentiation.*

*Legend of Supplementary Figure S4: BM populations are homogenous at the single cell level and*
*exhibit sequential and almost exclusive expression of Cebpa/CEBPA and Cebpe/CEBPE at the RNA*
*and protein level during granulocytic differentiation.*

The reviewer states that based on available CITE-seq data our populations are not transcriptionally
homogenous and, as noted above, we do acknowledge that we cannot state that bulk sorted
populations are homogenous. It is not clear to us which study is being referred to, but we took a look at
the web server (<http://www.altanalyze.org/ICGS/Neutrophil/index.php>) derived from Muench et al
(2020) to assess the correlation between our markers and the CITE-seq derived data. Specifically, four
of our markers are represented with CITE-seq abs (CD115, CD11b, Ly6G, KIT) in this work (Muench et
al, 2020). Except for Kit, we see a good concordance with the expression profiles of these markers and
our populations. However, the CITE-seq data fail to support the co-expression of KIT and Ly6G (within
the CD34- compartment) which we use to distinguish MY1 (Ly6G neg) and MY2 (Ly6G int) from each
other. However, our FACS data clearly support the existence of these populations, specifically a KIT-
dim/int population which can be divided based on Ly6G (see Fig 1b). The reason for this discrepancy is
not clear, but we do note that the *Kit* mRNA is expressed in cells with intermediate Ly6G-ADT
expression (preNeu-2) despite them being apparently negative for the CD117-ADT. We also note that
the CD117-ADT is not particular bright in this data set, not even in the most immature populations
which should express high levels of KIT. Finally, it represents a formal possibility that conventional flow
cytometry may display a higher dynamic range for some markers.

**Comment #3**

- It is not clear whether there is heterogeneity within the identified populations, for which single cell
assays are required to validate.

- There is nowhere near enough data to justify that these populations are transcriptionally distinct and
pure relative to single-cell heterogeneity

The authors response to this comment is absolutely not sufficient. Single cell assays are required to
show that the single-cells in the proposed sorted gates are homogenous. Bulk assays will average
together the signals from multiple populations. Those averages will easily map to distinct positions in
cellular differentiation trajectories. This does not mean that the sorted populations are homogenous. It
simply means that the averaged signal from the bulk assay represents a single point in the high

dimensional expression space. This emphasizes the need for the authors to look at more markers
within the panel they are using for sorting.

**Response referee #1 comment #3:**

As stated in our response to comment #2, we completely agree with the referee that our sorted
populations are NOT homogeneous or pure with respect to biological state of their single cells (i.e.
their transcriptome, etc.) but merely reflect an average biological state of cells with a specific
immunophenotype. We also agree that these averages will map to distinct positions in the sc
transcriptome differentiation trajectories. As demonstrated in Supplementary Figure S3A single cells
annotated to the bulk transcriptome of specific sorted BM populations maintained the hierarchical
order of monocytic and granulocytic differentiation trajectories based on similarity of gene expression
profiles. These novel analyses strongly suggest that the “average” transcriptome states of our sorted
populations, albeit they are not homogeneous populations, represents relevant snap shots of
successive stages of granulocytic and monocytic differentiation. Moreover, as the purpose of our
sorting was to isolate enough cells for downstream molecular analyses, impurities in the analyzed
populations would be averaged out.

As stated above in our response to comment #2, we have corrected the text accordingly in the MS and
removed our statements that the sorted BM populations are homogenous or pure with respect to the
biological state (i.e. transcriptome, etc.) of their individual cells.

**Comment #4**

- - The separation of populations is based on only 8 markers (Lin, Kit, Sca, CD115, CD16/32, CD34, Ly6G,
CD11b)
- Again, need to respond to work that was previously published in this area
- Other groups are using assays that capture signals from >20 markers (CYTOF and CITE-seq). Can the
authors comment on the purpose of using such tools if 8 markers is sufficient to isolate these
populations of interest?

Here, the authors state that including more markers could further improve the resolution of sorted BM
populations, yet above, they said that they had isolated pure populations with their panel. Both
statements can't be true, if you can improve the resolution with more antibodies, then the populations
you are identifying are not pure. The authors should remove any statements from the manuscript
claiming that the BM populations are homogenous.

**Response referee #1 comment #4:**

As stated in our response to comment #2 and #3, we completely agree with the referee that our sorted
populations are NOT homogeneous or pure with respect to biological state of their single cells (i.e.

their transcriptome, etc.) but merely reflect an average biological state of cells with a specific
immunophenotype.

We have corrected the text accordingly in the MS and removed our statements that the sorted BM
populations are homogenous or pure with respect to the biological state (i.e. transcriptome, etc.) of
their individual cells (for details see response to comment #2).

**Comment #5**

- The authors claim that unipotent granulocytic (GP) and monocytic (MP) progenitor populations exist,
yet they don't validate these predictions with colony forming assays or lineage tracing experiments, or
reference previous literature to validate that these populations are unipotent progenitors for the
specified trajectories.

To respond to this point, the authors reference a prior study in which they observed ~80% of lineage
restricted progenitors in their gates that are unipotent granulocytic and monocytic. It is confusing that
the authors do not try to repeat this with their proposed new "homogenous" populations, as surely
they should improve the accuracy to which the unipotent progenitors give rise to lineage restricted
colonies. Again, either the authors should adjust the language of the text such that they are not
claiming that their populations are homogenous, or they should redo this assay and show that they can
improve the accuracy.

**Response referee #1 comment #5:**

We are sorry for the confusion here. The isolated GPs and MPs in this study are essentially identical to
those in our previous work (Pundhir et al, 2018) and were based on the protocol by Olsson et al (2016),
who was the first to report the existence of these populations. Essentially, the conventional GMP
population is resolved based on CD115. But again, we completely agree with the referee and have
corrected the text accordingly in the MS and removed our statements that the sorted BM populations
are homogenous or pure with respect to the biological state (i.e. transcriptome, etc.) of their individual
cells (for details see response to comment #2).

**Comment #6**

- The authors claim that Cebpa and Cebpe show mutually exclusive expression profiles, but this is
wrong. Figure 2B is certainly insufficient to claim this. First of all, these are bulk RNA-seq samples, so it
is not clear if single cells simultaneously express Cebpa and Cebpe. Previously published scRNA-seq
data from multiple papers shows that there are many cells that co-express these genes. This claim is
wrong and no data presented by the authors provides conclusive evidence to the contrary.

The authors further investigated this point and discovered that a population of cells do in fact co-
express Cebpa and Cebpe, referred to as the "PM" population. It is good that they changed the
language of the manuscript so that false statements are not being made.

**Response referee #1 comment #6:**

We thank the reviewer for this encouraging comment.

**Comment #7**

- The effect of CEBPE from enhancers and promoters at this point is only an association of the binding
of CEBPE to the change in gene expression at different stages. No causal role for CEBPE can be made
without functional validation of genomic loci on target gene expression.

- For example, the authors could mutate CEBPE binding sites in promoters and enhancers and test for
changes in target gene expression

- Alternatively, CRISPRi assays could be performed to silence the chromatin regions for functional
validation

The authors validated some examples of targets of CEBPE effectively.

**Response referee #1 comment #7:**

Again, we thank the reviewer for this encouraging comment.

**Minor Comment #1**

- Gene clusters identified in 2A are overlapping in areas

- Eg. what is the difference between: LSK vs preGM? GP vs PM? MY1 vs MY2? MO1 vs MO2?

- If these are unique populations, they should certainly have unique gene expression patterns...

The authors clarified that there are unique gene expression patterns in the given clusters. It is clear
from the PCA analysis in Figure 1D that the bulk transcriptomes reflect unique expression patterns.

However, it is still not clear if the sorted populations are homogenous.

**Response referee #1 minor comment #1:**

We refer the reviewer to our comments to points 1-5

**Minor Comment #2**

- The authors state:

"Overall, these findings demonstrate that Cebpa expression correlates with myeloid commitment of
HSCs (i.e. LSKs) toward lineage-restricted progenitors (i.e. GPs/PMs) and expression of early

granulocytic differentiation genes (i.e. primary granule proteins)."

- However, the word commitment should be replaced with specification (moving from a multipotent
state towards a single lineage, but not fully committed). Commitment should be a word reserved for
not being able to produce other lineages. However, the authors do not show that this cluster is a
commitment stage, and previous literature would suggest that it is myeloid specification.

The authors agreed and changed the manuscript to fix this issue.

**Response referee #1 minor comment #2:**

We are happy that the reviewer agrees

**Minor Comment #3**

- The authors proposed enhancers for genes lack a clear genomics + informatics approach to link
enhancers to those genes. The assessment of genomic loci as enhancers >1,000 bp from the
transcription start site is ambiguous

- Would the authors consider a loci 1,000,000 bp from a TSS to be an enhancer of the gene?

- Promoter capture Hi-C or co-accessibility of genomic regions accross a single-cell experiment would
provide better definition of enhancer-gene connections (again, if the authors had read prior literature,
they would see that some of these datasets already exist)

The authors clarified this in the text effectively.

**Response referee #1 minor comment #3:**

We are happy for the reviewers assessment

**Minor Comment #4**

- Information about the 3D chromatin shape is lacking. Is it the case that there are interactions
provoked by CEBPE bound enhancer that help to explain the changes in gene expression?

The authors claim that studying the 3D chromatin shape is outside the scope of the paper, which is
acceptable after they provided more support for the functional role of Cebpe later in the text.

**Response referee #1 minor comment #4:**

We are happy that the reviewer agrees

**Minor Comment #5**

- Given the large number of motifs in the databases used ("Known TF-binding motifs were downloaded
from MEME, Jaspar, Uniprobe, Hocomoco and Jolma et al."), the authors should include the motif
scores for all of the tested motifs, at least in the supplemental information.

The authors provided the requested information.

**Response referee #1 minor comment #5:**

We are happy that the reviewer agrees

**Minor Comment #6**

- Ranked importance of TFs seems heavily biased towards the available ChIP-seq data. The authors
should comment on efforts to control for the importance of other TFs.

The authors clarified this.

**Response referee #1 minor comment #6:**

Thank you

**Minor Comment #7**

- Can co-immunoprecipitation validate the complex of CEBPE and E2F in the indicated populations?

The authors provided EMSA experiments and updated the manuscript to provide validation for this
mechanism.

**Response referee #1 minor comment #7:**

We are happy that the reviewer agrees

**Minor Comment #8**

- Can 3D chromatin conformation confirm chromatin looping of these enhancer-promoter
interactions? Alternatively, can the authors comment on how the regulation of this gene via this
enhancer is occurring?

Again, the 3D chromatin structure is proposed to be outside the scope of the paper.

Response referee #1 minor comment #8:

We are happy that the reviewer agrees

Minor Comment #9

- Can the authors comment on the structural differences of CEBPA and CEBPE that may confer differential cell cycle activity (interaction with E2F complex)?

The authors showed in vitro that CEBPA can induce granulocytic differentiation and argue that the structural differences do not confer the differential activity observed.

Response referee #1 minor comment #9:

Thank you

Minor Comment #10

- The authors should supply a supplementary file with each of the CEBPE motif positions that they find interesting in describing the target genes of interest, such that they can provide genomic coordinates for the regulatory activity of CEBPE.

The authors provided this information.

Response referee #1 minor comment #10:

Thank you

Minor Comment #11

- Is there a reason the authors don't plot ChIP-seq profiles for CEBPA and CEBPE simultaneously for the same populations (Figure 5E/F)?

- Do CEBPA and CEBPE ever bind to the same location in the same cell population?

- I would really like to see the dynamics of CEBPA and CEBPE binding across all of the populations

The authors provide additional ChIP-seq data which helps to understand the specification event.

**Response referee #1 minor comment #11:**

Thank you

**Minor Comment #12**

- Can the authors functionally validate the MYC/MAX activating sites of the Cebpa promoter?

- Would mutation at these MYC/MAX sites inhibit Cebpa promoter activity? (This could easily be
answered with a Cebpa reporter, cloning in the enhancer upstream)

The authors added this experiment to the paper.

**Response referee #1 minor comment #12:**

Thank you

**Minor Comment #13**

- Why are CEBPA and CEBPE written in all caps sometimes? Aren't all of the data generated in mice,
and should take on the convention of Cebpa and Cebpb with the first letter alone capitalized?

The authors clarified this misunderstanding.

**Response referee #1 minor comment #13:**

Thank you

**Minor Comment #14**

- Shouldn't the title use identity instead of identify

- Eg. Transcription factor-driven coordination of cell cycle exit and specification of cell identity during
granulocytic differentiation

The authors changed this.

**Response referee #1 minor comment #14:**

Thank you

**Reviewer #2 (Remarks to the Author):**

The authors have addressed all of my comments and suggestions. Particularly, I am very happy to see
that the authors performed a quantitative MS analysis of CEBPA and CEBPE proteins to directly
compare their expression levels. Overall, the quality of the paper has been significantly improved.

**Response referee #2**

We thank the reviewer for this encouraging comment

**Reviewer #3 (Remarks to the Author):**

The authors have performed a number of new experiments that significantly improve the manuscript
and start to address some of the functional predictions derived from their RNA-seq and CHIP-seq
experiments. While the study remains largely descriptive, the new functional experiments and
extended analyses produce a study that will be of interest to the field and is performed about as well
as possible given current technical limitations.

**Response referee #3**

We thank the reviewer for this encouraging comment

REVIEWERS' COMMENTS

Reviewer #1 (Remarks to the Author):

In the revised manuscript, Theilgaard-Mönch et al. identify subsets of cells in the murine bone marrow that in average follow a trajectory for monocytic and neutrophil differentiation. The authors highlight the importance of two transcription factors, CEBPA and CEBPE, which are shown to play roles in the induction of granulocytic genes and silencing of cell cycle genes as uncommitted progenitors undergo specification and commitment to more mature myeloid progenitors. The revisions, particularly the clarification of what is being sorted and how these populations compare to what has been done in the past, fully address concerns that the data were not consistent with claims. This is a valuable study that elucidates parts of the regulatory networks that lead to the production of myeloid cells.

Point-to-point response to the referees' comments:

Reviewer #1 (final remark to the Author):

In the revised manuscript, Theilgaard-Mönch et al. identify subsets of cells in the murine bone marrow that in average follow a trajectory for monocytic and neutrophil differentiation. The authors highlight the importance of two transcription factors, CEBPA and CEBPE, which are shown to play roles in the induction of granulocytic genes and silencing of cell cycle genes as uncommitted progenitors undergo specification and commitment to more mature myeloid progenitors. The revisions, particularly the clarification of what is being sorted and how these populations compare to what has been done in the past, fully address concerns that the data were not consistent with claims. This is a valuable study that elucidates parts of the regulatory networks that lead to the production of myeloid cells

Response referee #1:

We thank the reviewer for this very positive remark.